# Interfacial assembly of binary atomic metal-$N_x$ sites for high-performance energy devices

Zhe Jiang [1,2,3,9], Xuerui Liu[4,9], Xiao-Zhi Liu[5,9], Shuang Huang[6], Ying Liu[1], Ze-Cheng Yao [1,3], Yun Zhang[1], Qing-Hua Zhang[5], Lin Gu [5], Li-Rong Zheng[7], Li Li[6], Jianan Zhang[8], Youjun Fan [2] ✉, Tang Tang [1,3] ✉, Zhongbin Zhuang [4] & Jin-Song Hu [1,3] ✉

Anion-exchange membrane fuel cells and Zn–air batteries based on non-Pt group metal catalysts typically suffer from sluggish cathodic oxygen reduction. Designing advanced catalyst architectures to improve the catalyst's oxygen reduction activity and boosting the accessible site density by increasing metal loading and site utilization are potential ways to achieve high device performances. Herein, we report an interfacial assembly strategy to achieve binary single-atomic Fe/Co-$N_x$ with high mass loadings through constructing a nanocage structure and concentrating high-density accessible binary single-atomic Fe/Co–$N_x$ sites in a porous shell. The prepared FeCo-NCH features metal loading with a single-atomic distribution as high as 7.9 wt% and an accessible site density of around $7.6 \times 10^{19}$ sites $g^{-1}$, surpassing most reported M–$N_x$ catalysts. In anion exchange membrane fuel cells and zinc–air batteries, the FeCo-NCH material delivers peak power densities of 569.0 or 414.5 mW cm$^{-2}$, 3.4 or 2.8 times higher than control devices assembled with FeCo-NC. These results suggest that the present strategy for promoting catalytic site utilization offers new possibilities for exploring efficient low-cost electrocatalysts to boost the performance of various energy devices.

Clean energy conversion and storage devices such as low-temperature membrane-based hydrogen fuel cells and metal-air batteries have been attracting intensive research interest[1–3]. However, the commercialization of these electrocatalytic energy techniques is still hindered by the sluggish cathodic oxygen reduction reaction (ORR)[4,5], which requires high-performance electrocatalysts such as platinum group metal (PGM) materials for accelerating the ORR kinetics[6,7]. The high cost and scarcity of PGMs have driven numerous efforts to reduce their usage[8].

Compared with the proton-exchange membrane fuel cells (PEMFCs) which rely on the extensive use of PGMs[9], anion-exchange membrane fuel cells (AEMFCs) and aqueous Zn–air batteries (ZABs) have received increasing attention for their potentials of using earth-abundant non-PGM materials as the catalysts[10–13]. Given the high-loading PGM catalysts are still required for competitive output in the state-of-the-art AEMFC and ZABs[14], it is necessary to develop efficient and durable non-PGM ORR electrocatalysts[15,16]. Due to the intrinsically sluggish kinetics

[1]Beijing National Laboratory for Molecular Sciences (BNLMS), Institute of Chemistry, Chinese Academy of Sciences, Beijing 100190, China. [2]Guangxi Key Laboratory of Low Carbon Energy Materials, School of Chemistry and Pharmaceutical Sciences, Guangxi Normal University, Guilin 541004, China. [3]University of Chinese Academy of Sciences, Beijing 100049, China. [4]Beijing Advanced Innovation Center for Soft Matter Science and Engineering, Beijing University of Chemical Technology, Beijing 100029, China. [5]Beijing National Laboratory for Condensed Matter Physics, Institute of Physics, Chinese Academy of Sciences, Beijing 100190, China. [6]Chongqing Key Laboratory of Chemical Process for Clean Energy and Resource Utilization, School of Chemistry and Chemical Engineering, Chongqing University, Chongqing 400044, China. [7]Beijing Synchrotron Radiation Facility, Institute of High Energy Physics, Chinese Academy of Sciences, Beijing 100049, China. [8]College of Materials Science and Engineering, Zhengzhou University, Zhengzhou 450001, China. [9]These authors contributed equally: Zhe Jiang, Xuerui Liu, Xiao-Zhi Liu. ✉e-mail: youjunfan@mailbox.gxnu.edu.cn; tangtang@iccas.ac.cn; hujs@iccas.ac.cn

of non-PGM ORR electrocatalysts compared to PGM ones, constructing high-density accessible active sites is thus indispensable for their use at the device level although it is still challenging.

Among various non-PGM alternatives, single-atomically dispersed M−N$_x$ (M−N−C, M = Fe, Co, Ni, etc.) catalysts have been regarded as the most promising candidates to replace PGM catalysts for ORR due to their relatively suitable adsorption/desorption of ORR intermediates such as *OH and *OOH[17–20]. To further improve the intrinsic activity, binary dual-atomic M−N$_x$ sites have been recently developed. They not only inherit the advantages of single-atom catalysts (SACs), such as maximal atomic utilization, well-defined, and adjustable coordination configuration but also offer the possibility to improve catalytic performance by harnessing synergistic effects[15,21]. The interaction between multiple active centers modifies the electronic structure and geometric configuration of the active site, which optimizes the adsorption and desorption of intermediates on the active site to approach the apex of the activity volcano plot[19,22].

Moreover, although a variety of M−N−C catalysts have shown impressive ORR activities in rotating disk electrode (RDE) tests, few of them have demonstrated comparable performance to PGM catalysts in fuel cells or ZABs[16,23]. It is probably due to the insufficient amount of available M−N$_x$ sites in the catalyst layer[8,24,25]. Only the M−N$_x$ accessible to the tri-phase interface are able to participate in ORR, leaving those inside catalysts inactive[26]. For example, Kucernak et al. reported that only 4.5% of Fe−N$_x$ sites functioned in their Fe−N−C SACs[18]. Since the power density of the device is proportional to the accessible site density (ASD) and cannot be amplified by simply increasing catalyst loading and thickening the catalyst layer due to mass transfer issues[27,28], a feasible way is to construct high-density highly-active catalytic sites while maximizing their utilization by making them accessible to reactants[29]. The current efforts on atomic M−N−C catalysts mainly focused on how to increase metal loading. More attention is needed on how to simultaneously maximize the site utilization to achieve high ASD to overcome the bottleneck of their use at the device level[8,30].

Inspired by these requirements for device-level application, we herein reported an interfacial assembly strategy for constructing binary single-atomic Fe/Co-N$_x$ sites with a high ASD (denoted as FeCo-NCH) as an efficient ORR electrocatalyst for high-performance AEMFC and ZAB. The binary metal incorporation promoted both the intrinsic electrocatalytic activity for ORR and the metal loading of single atomic sites to 7.9 wt%. More importantly, the stress-induced interfacial assembly was able to construct ultra-thin nanocages from the nano-polyhedron precursors. The atomic sites were thus concentrated and exposed to the tri-phase interface at both outside and inside of nanocages, giving an ultrahigh ASD of around $7.6 \times 10^{19}$ sites g$^{-1}$ with a site utilization of 9.3%. In contrast, the solid nano-polyhedron counterpart with a similar shape and metal loading gave only around a quarter of ASD. As a result, when such FeCo-NCH ORR catalysts were assembled as O$_2$ cathodes in AEMFC or ZAB, the device power density was lifted to 569.0 mW cm$^{-2}$ or 414.5 mW cm$^{-2}$, 3.4 or 2.8 times higher than those with solid catalysts, respectively. Such performance was also comparable to the devices with commercial Pt/C (20 wt% Pt), suggesting the present strategy for increasing the site utilization is efficient for boosting the energy device performance, thus opening up opportunities for replacing PGM catalysts with low-cost earth-abundant ones.

## Results and Discussion
### Synthesis and structural characterizations of FeCo-NCH

The atomically dispersed Fe/Co-N$_x$ catalytic sites were synthesized through the coordination-assisted pyrolysis of ZIF precursors and their interfacial assembly on ultra-thin hollow N-doped carbon nanocages (FeCo-NCH) was achieved by stress-induced orientation contraction during the pyrolysis. As illustrated in Fig. 1, the ZIF-8 polyhedral

nanocrystals were first prepared as the sacrificial matrix. A thin layer of Co-containing ZIF-67 was then epitaxially grown onto these ZIF-8 nanocrystals (ZIF-8@ZIF-67) given they shared similar topological structures and cell parameters[31]. To achieve binary metal sites, Fe adsorption was further carried out on ZIF-8@ZIF-67 via a facile impregnation (ZIF-8@ZIF-67@Fe) so that both Co and Fe species exist in the ZIF-67 layer. The morphology and structure of the ZIF-8 matrix were well maintained during the two processes, as evidenced by scanning electron microscopic (SEM) and transmission electron microscopic (TEM) images as well as X-ray diffraction (XRD) experiments (Supplementary Figs. 1 and 2). Subsequently, ZIF-8@ZIF-67@Fe was coated with an ultra-thin polydopamine (PDA) shell. During the pyrolysis, the relatively rigid carbonized PDA shell (Supplementary Fig. 3) formed at the very early stage acted as a framework to restrain the contraction of the inner ZIF component and induced the accumulation of pyrolyzed species on the shell, resulting in the final hollow structure. The Fe, Co-containing ZIF-67 layer was accordingly transformed to the carbon shell with neighboring Fe-N and Co-N moieties. Such formation process was validated by a series of TEM images taken from the various decomposing stages (see details in Supplementary Fig. 4). Without the assistance of the PDA coating, the pyrolyzed product ended up with a solid nano-polyhedral morphology with a concaved surface (Supplementary Fig. 5).

TEM images (Fig. 2a and inset, Supplementary Fig. 6) show that the precursor has a morphology of solid polyhedrons in a size of around 100 nm with a coating layer of ~3 nm. The pyrolyzed product FeCo-NCH inherits the polyhedral morphology of its precursor as evidenced by the SEM image (Fig. 2b) while the hollow interior is clearly revealed by high-angle annular darkfield scanning transmission electron microscopic (HAADF-STEM) images (Fig. 2c, Supplementary Fig. 7). The clear contrast difference between the carbon shell and hollow interior indicates that the shell thickness of the nanocage is about 10 nm (Fig. 2d, e). No aggregated metal-based nanoparticles are found in the shell or inside the nanocage, implying the metal species should be atomically distributed in the shell. XRD pattern (Supplementary Fig. 8) further corroborates that no diffraction peaks from crystalline species are detected except for two typical broad peaks at ~ 24.2° and ~ 43.4° from carbon[32]. The atomic dispersion of metal species is confirmed by aberration-corrected HAADF-STEM imaging. The discrete brighter dots in a single atomic size circled in red in Fig. 2f could be safely identified as Fe or Co atoms. Energy-dispersive spectroscopic (EDS) elemental mapping images (Fig. 2g) suggest the uniform distribution of Fe, Co, N, and C elements in the nanocage shells, consistent with linear scanning results (Supplementary Fig. 9). This result implies the N-doped carbon shells and the coexistence of Fe/Co-N$_x$ sites.

The surface chemical composition and elemental state of the prepared FeCo-NCH were investigated via X-ray photoelectron spectroscopy (XPS). The XPS survey shows the coexists of the Fe, Co, N, O, and C in the catalyst (Supplementary Fig. 10). The surface Fe and Co content examined by XPS is 0.5 at.% (2.1 wt%) and 1.12 at.% (4.6 wt%), respectively, giving a total surface metal content of 6.7 wt% (Supplementary Table 3). The N, O, and C content is 9.23, 9.34, and 79.81 at.%, respectively (Supplementary Figs. 10 and 11). The surface oxygen in the FeCo-NCH originates from the carbon-oxygen functionalities, which are commonly observed in carbon-supported materials[33,34]. As shown in (Supplementary Fig. 11a), the high-resolution C 1$s$ XPS spectrum exhibits three peaks at the binding energies of 284.4, 285.7, 286.9, and 289.2 eV, corresponding to C−C, C−N, C−O and C=O bonds, respectively[35]. The high surface N content (9.23 at.%) contributes to the formation of M-N$_x$ active sites. The high-resolution N 1$s$ spectrum (Supplementary Fig. 11b) can be deconvoluted into four typical peaks at 398.3 eV, 399.6 eV, 400.8 eV, and 401.1 eV, referring to pyridinic-N, Fe/Co−N, pyrrolic-N, and graphitic-N, respectively. These results suggest the presence of Fe/Co−N moieties in FeCo-NCH[36]. For metal

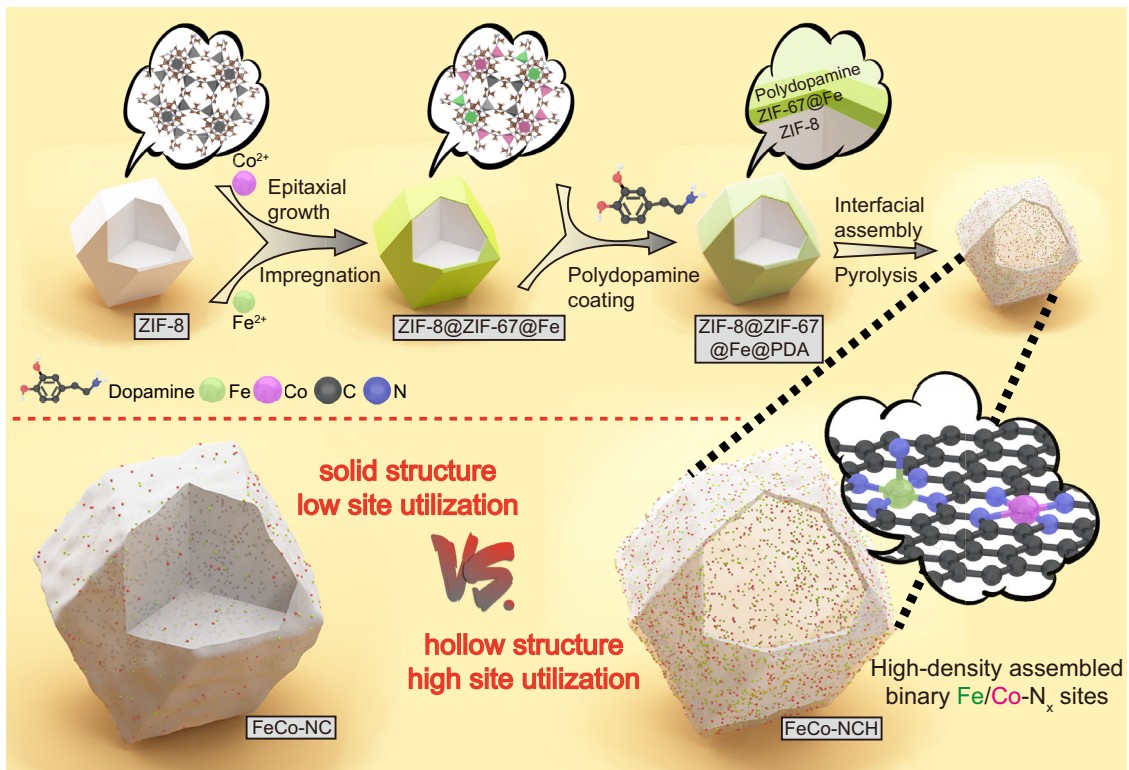

**Fig. 1 | Schematic illustration of FeCo-NCH.** Illustration of the typical synthesis of FeCo-NCH with atomically dispersed binary Fe/Co-Nₓ sites in the shell.

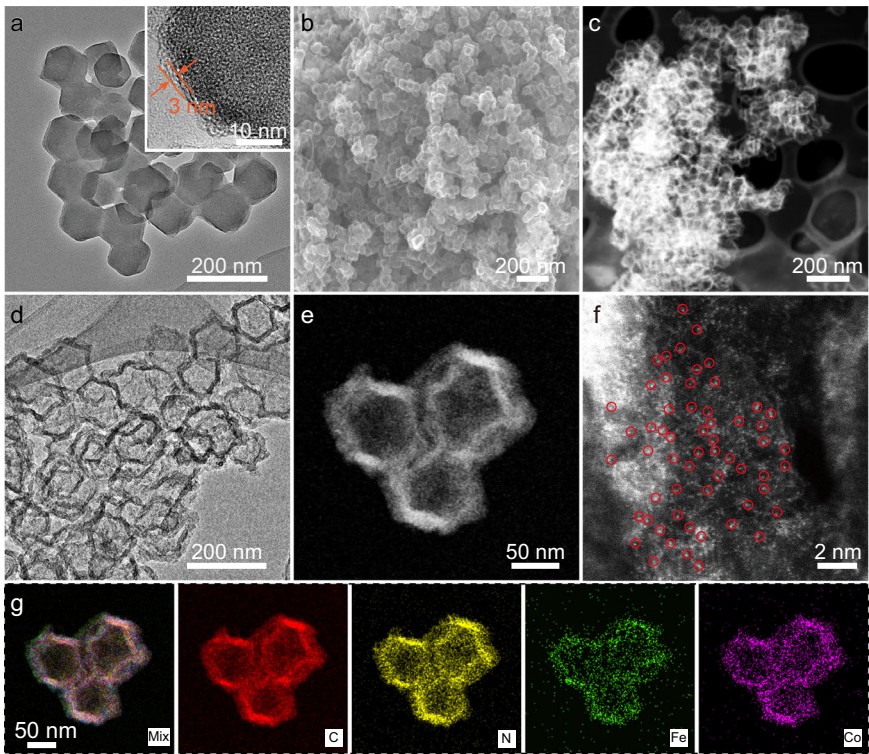

**Fig. 2 | Structural characterizations of FeCo-NCH. a** TEM image of the ZIF-8@ZIF-67@Fe@PDA (inset: HRTEM image), **b** SEM image, **c, d** TEM images, **e, f** HAADF-STEM images, and **g** elemental mappings of the FeCo-NCH.

species, the peaks at 709.2 and 711.1 eV in the Fe 2*p* spectrum can be well indexed to Fe species in an oxidative state and the peaks at 780.5 eV in Co 2*p* spectrum are safely assigned to bivalent Co species (Supplementary Fig. 11d, e), corroborating the formation of Fe/Co–N

coordination[37]. It is noted that no metallic Fe or Co species are detected, ruling out the formation of metal particles in FeCo-NCH. These results support the formation of Fe-Nₓ and Co-Nₓ sites which are favorable active sites for ORR.

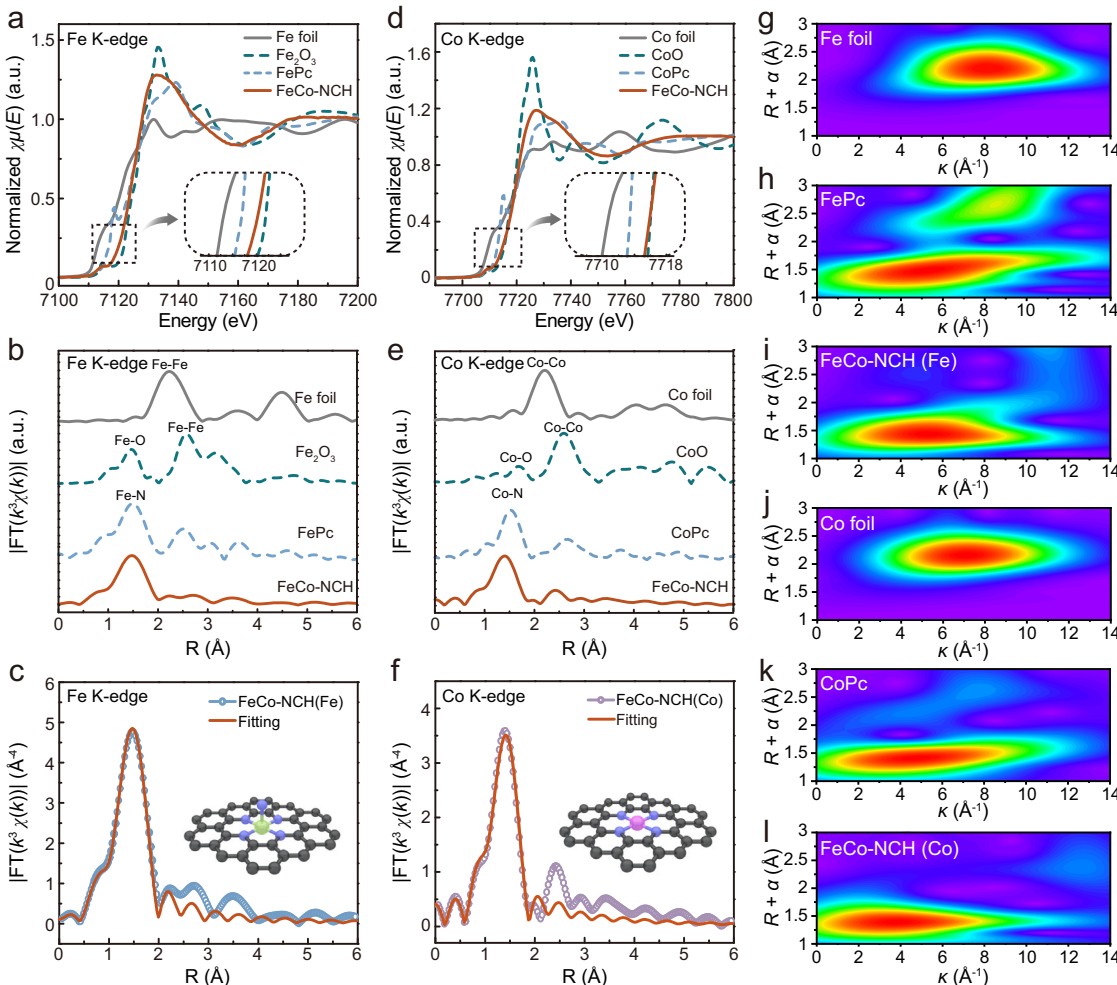

**Fig. 3 | Local structural characterizations of FeCo-NCH. a** Fe K-edge XANES spectra and **b** Fourier-transform (FT) EXAFS spectra of FeCo-NCH and reference samples. **c** The fitting of the FT R-space Fe K-edge EXAFS of FeCo-NCH. **d** Co K-edge XANES spectra and **e** FT EXAFS spectra of FeCo-NCH and reference samples. **f** The fitting of the FT R-space Co K-edge EXAFS of FeCo-NCH. **g**–**l** Wavelet transform for the $k^3$-weighted EXAFS of FeCo-NCH and reference samples (Fe foil, FePc, Co foil, and CoPc).

To further investigate the coordination environments of the catalytic active Fe and Co species, the X-ray absorption near-edge structures (XANES) and the extended X-ray absorption fine structures (EXAFS) were investigated for the FeCo-NCH and the reference materials. According to the Fe K-edge XANES spectra (Fig. 3a), the position of the absorption threshold for FeCo-NCH locates between iron phthalocyanine (FePc) and $Fe_2O_3$ but close to $Fe_2O_3$. The zoom-in pre-edge profiles are presented as an inset in Fig. 3a. It indicates that the average oxidation state of Fe species is close to +3[8]. This can be ascribed to the neighboring N atoms partially depleting Fe-free electrons through the valence bond[38], which is consistent with XPS results. The coordination environments are further analyzed by the $k^3$-weighted Fourier transforms (FT) of the extended X-ray absorption fine structure (FT-EXAFS) at the Fe K-edge. The main peak at ~1.47 Å (Fig. 3b) is similar to that in FePc, which is typically assigned to the Fe−N coordination at the first shell[39]. Distinct from the reference materials, no obvious peaks for longer backscattering paths such as Fe-Fe bonding are observed in the FT-EXAFS spectrum of FeCo-NCH, corroborating the single-atomic dispersion of Fe species without long-range order in the matrix[40]. The EXAFS fitting indicates that the first shell of the Fe exhibits an average coordination number of 4.9 for Fe-N (Fig. 3c, Supplementary Fig. 12a, details see Supplementary Table 1,), suggesting each Fe atom is bonded with five N atoms at 2.01 Å.

Meanwhile, the Co K-edge XANES spectra (Fig. 3d) show that the position of the absorption threshold of FeCo-NCH is close to the cobalt phthalocyanine (CoPc) and CoO. The zoom-in pre-edge profiles clarify that the valence state of Co species in FeCo-NCH is close to +2[41]. The main peak at ~1.40 Å in the Co K-edge FT-EXAFS spectrum of FeCo-NCH (Fig. 3e) can be assigned to the Co−N coordination at the first shell. This is similar to CoPc and distinguished from the Co-O coordination[42]. No obvious peaks for longer backscattering paths such as Co-Co are detected, indicating Co species in FeCo-NCH exist mainly in a single atomic state. The EXAFS fitting curve supports that the first shell of the Co atom has an average coordination number of 4.1 for Co−N, indicating that each Co atom should be bonded with 4 N atoms at 1.93 Å (Fig. 3f, Supplementary Fig. 12b, details see Supplementary Table 1).

Wavelet transform (WT) analysis was further carried out to investigate the Fe and Co K-edge EXAFS oscillations of FeCo-NCH and the reference materials. As shown in Fig. 3g-l, WT analysis of Fe K-edge displays only one intensity maximum at about 5.4 Å⁻¹ for FeCo-NCH, which is very close to FePc (5.5 Å⁻¹) but distinct from Fe foil (8.5 Å⁻¹)[43]. Similarly, the only one intensity maximum at 4.1 Å⁻¹ in Co K-edge WT contour plots for FeCo-NCH is very close to the reference CoPc (4.3 Å⁻¹) but distinct from Co foil (7.3 Å⁻¹)[44]. This results together with the above HAADF-STEM, EXANES, EXAFS, and WT analysis results

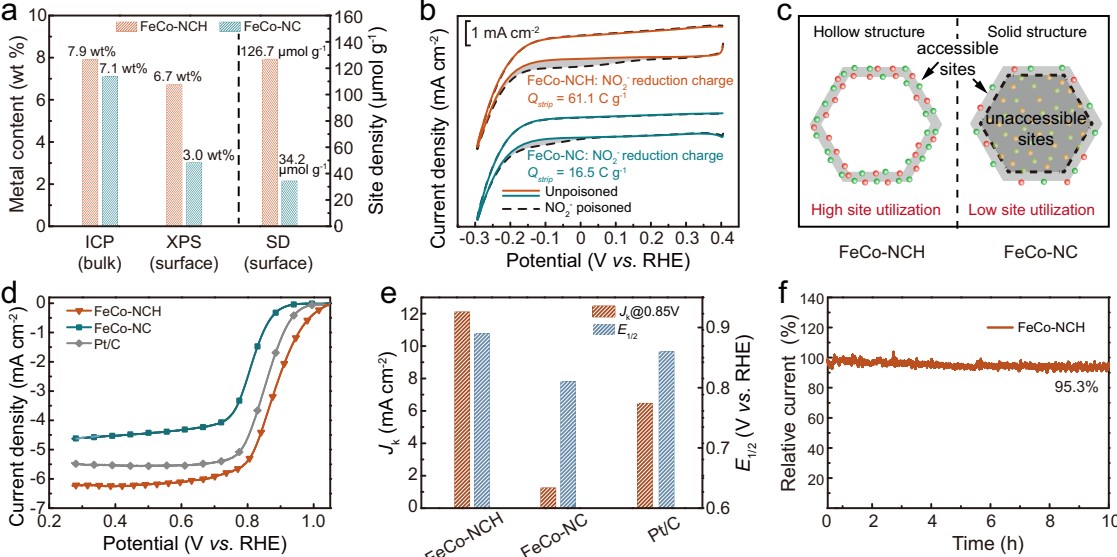

**Fig. 4 | Evaluation of the site utilization and electrocatalytic performance of prepared catalysts for ORR. a** The comparison of metal sites in FeCo-NCH and FeCo-NC, measured by XPS, ICP, and electrocatalytic measurements. **b** CV curves for FeCo-NCH and FeCo-NC under unpoisoned or $NO_2^-$ poisoned conditions. **c** Scheme showing the hollow structure with high site utilization and the solid structure with low site utilization. **d** Steady-state ORR polarization curves of FeCo-NCH, FeCo-NC, and Pt/C in $O_2$-saturated 0.1 M KOH under a rotating rate of 1600 rpm. **e** Kinetic current density ($J_k$) at 0.85 V and half-wave potential ($E_{1/2}$) for these catalysts. **f** Normalized chronoamperometric curve for the FeCo-NCH.

consistently confirmed the co-existence of single-atomically dispersed Fe-N$_5$ structure and Co-N$_4$ structure in FeCo-NCH.

To demonstrate the advantage of the present interfacial assembly strategy in the construction of high-density accessible binary M-N$_x$ sites, a couple of control samples were delicately designed and synthesized in parallel. As detailed in the Supplementary Information, the two control samples containing only Fe or Co were synthesized via a similar procedure and denoted as Fe-NCH or Co-NCH, respectively (details see Experimental procedures, Supplementary). Systematic characterizations suggest that Fe-NCH or Co-NCH shares similar nanocage morphology to FeCo-NCH but contains single-atomically dispersed Fe-N$_x$ or Co-N$_x$ sites only in the nanocage shell (Supplementary Figs. 13 and 14). XANES and EXAFS results corroborate that they have single-atomically dispersed Fe-N$_x$ or Co-N$_x$ sites similar to FeCo-NCH (Supplementary Figs. 15 and 16). Another control catalyst with a similar shape to FeCo-NCH but a solid interior (denoted as FeCo-NC) was prepared by similar procedures but without PDA coating outside ZIF cores. SEM and TEM images indicate that it shows the morphology of solid nanopolyhedron with a uniform elemental distribution of Co and Fe throughout the whole polyhedron (Supplementary Fig. 17).

The mass loadings of Fe and/or Co in these samples were precisely determined by the inductively coupled plasma-optical emission spectroscopy (ICP-OES). The control Fe-NCH has 2.7 wt% Fe and Co-NCH has 5.2 wt% Co. Further increasing metal loading results in the aggregation of metal atoms in Fe-NCH or Co-NCH. In contrast, FeCo-NCH can simultaneously hold 2.4 wt% Fe and 5.5 wt% Co, giving a total metal loading as high as 7.9 wt% which ranks among the highest for atomically dispersed bimetal electrocatalysts derived from MOF precursors (Supplementary Table 2)[45]. This can be attributed to our ingenious synthesis strategy, the introduction of Fe was carried out under the optimal load of Co, and the nitrogen in the PDA-derived NC layer will capture Fe atoms at high temperature to form Fe-N$_x$, leading to the increased total metal content. This result indicates that the current coordination-assisted adsorption with PDA-assisted pyrolysis is an effective strategy for synthesizing high-density single-atomic metal-N$_x$ sites. Moreover, the solid control sample FeCo-NC shows a similar total metal loading of 7.1 wt%.

## Determination of the site density

Since previous reports have demonstrated that not all loaded M-N$_x$ sites in the catalysts can participate in electrochemical reactions[46], both physical characterizations and electrochemical measurements were thus conducted to evaluate the ASD and site utilization in hollow FeCo-NCH and solid FeCo-NC. In view of the electrocatalytic reaction only taking place at the catalyst surface, the hollow FeCo-NCH gives a much higher surface metal content than solid FeCo-NC (6.9 vs. 3.0 wt %) as examined by the surface-sensitive XPS (Fig. 4a). The metal loadings for all catalysts were listed in Supplementary Table 3. The ASD was further assessed by in-situ electrochemical nitrite stripping experiments (Supplementary Fig. 18). Since the nitrite ions ($NO_2^-$) are specifically adsorbed on the metal-N$_x$ sites and could be stripped via cyclic voltammetry (CV), this experiment is regarded as a reliable method for quantifying the electrochemically ASD in electrocatalysts[8,18,47]. A much larger $NO2^-$ reduction peak is observed for FeCo-NCH than FeCo-NC in the CV stripping curves (Fig. 4b). The total charge amount associated with the $NO_2^-$ reduction is determined to be 61.1 C g$^{-1}$ for FeCo-NCH, which is about 3.7 times greater than that for FeCo-NC (16.5 C g$^{-1}$). The ASD is thus calculated to be 126.7 µmol g$^{-1}$ (i.e., $7.6 \times 10^{19}$ sites g$^{-1}$) for FeCo-NCH, corresponding to a site utilization of 9.3%. In contrast, the solid FeCo-NC exhibits a much smaller ASD of 34.2 µmol g$^{-1}$ (i.e., $2.1 \times 10^{19}$ sites g$^{-1}$), corresponding to a low site utilization of only 2.5%.

The N$_2$ adsorption-desorption experiments were further performed to analyze the surface area and pore distribution to further understand the difference in site utilization for FeCo-NCH and FeCo-NC (Supplementary Figs. 19 and 20). The results show that FeCo-NCH has a Brunauer−Emmett−Teller (BET) surface area of 416 m$^2$ g$^{-1}$. The pore size distribution plot indicates that plenty of micropores in a size of less than 1 nm together with mesopores in a couple of nanometers exist in the nanocage shell. Although the control FeCo-NC has a slightly larger BET area of 491 m$^2$ g$^{-1}$, it holds only micropores in <1 nm while lack mesopores, leading to most of the loaded active sites being inaccessible[48,49]. Since the electrocatalytic reactions such as ORR involve both gas and electrolyte transfer to and from catalytic sites, it is reasonably believed that such a porous shell structure with a hollow interior in tens of nanometers in FeCo-NCH ensures much more sites

accessible and thus effectively multiplies the site utilization, as schemed in Fig. 4c[20,23,50].

## Electrochemical performance

The ORR activities of FeCo-NCH and control catalysts were subsequently evaluated through rotating disk electrode (RDE) measurements in an $O_2$-saturated 0.1 M KOH solution. All potentials are reported versus reversible hydrogen electrodes (vs. RHE). As shown in the linear sweep voltammetry (LSV) curves (Fig. 4d), the superior ORR activity is achieved on FeCo-NCH with a half-wave potential ($E_{1/2}$) of 0.889 V and a large limiting current density of 6.2 mA cm$^{-2}$, ranking among the most active state-of-the-art nonprecious metal ORR catalysts (Supplementary Table 4)[51,52]. This performance is significantly better than the control catalysts of Co-NCH ($E_{1/2}$: 0.86 V; limiting current density: 5.6 mA cm$^{-2}$) and Fe-NCH ($E_{1/2}$: 0.82 V; limiting current density: 4.2 mA cm$^{-2}$) (Supplementary Fig. 21), suggesting that the binary Fe/Co-$N_x$ sites are favorable for ORR due to the synergistic electronic modulation of the active site and the increase of site density[19]. The synergistic effect on binary Fe/Co-$N_x$ sites for ORR was also evaluated by the density functional theory (DFT) calculations. Three models (i.e., $FeN_5$, $CoN_4$, and $FeN_5$-$CoN_4$) were constructed based on the EXAFS results (Supplementary Fig. 22). The downhill free energy pathways at $U = 0$ V on both Co and Fe sites in three models indicate the ORR is spontaneous exothermal (Supplementary Figs. 23–26)[53]. The calculation results indicate that compared with Co or Fe-only site, the introduction of the Fe site promotes the desorption of *OH on the Co site (Supplementary Figs. 23, 24) while the introduction of the Co site significantly reduces the *OOH formation energy barrier on the Fe site (Supplementary Figs. 25, 26). In either case, the ORR pathway is promoted, suggesting the synergistic effect for $FeN_5$ and $CoN_4$ moieties and justifying the enhanced ORR activity for FeCo-NCH compared with single-metal Fe-NCH or Co-NCH. The detailed discussion can be found in Supplementary Information[22,43]. These results confirm that the constructing neighboring $FeN_5$ and $CoN_4$ moieties can generate a synergistic effect therebetween and contributes to enhanced ORR performance. Notably, the DFT calculation also predicts the higher intrinsic ORR activity of Co-$N_4$ to the Fe-$N_5$. Together with more active site loading, the Co-NCH exhibits better performance than the Fe-NCH (detailed discussion in Supplementary Fig. 26). Besides, the turnover frequency (TOF) of the FeCo-NCH was calculated based on the ASD. The TOF value of the FeCo-NCH is calculated to be 0.6 e$^-$ s$^{-1}$ site$^{-1}$ at 0.90 V, which is higher than most of previously reported ORR catalysts (Supplementary Table 5), indicating a fast ORR kinetic[54]. The high TOF together with the high ASD contributes to the enhanced ORR performance for FeCo-NCH.

Notably, the FeCo-NCH is also substantially superior to the solid FeCo-NC with the same kind of active sites ($E_{1/2}$: 0.81 V; limiting current density: 4.5 mA cm$^{-2}$). This result unambiguously proves that promoting site utilization by constructing a nanocage structure with a porous shell and hollow interior is an effective strategy for boosting electrocatalytic activity. Moreover, FeCo-NCH outperforms the commercial benchmark Pt/C (20 wt% Pt) as well ($E_{1/2}$: 0.86 V; limiting current density: 5.5 mA cm$^{-2}$). The kinetic current density ($J_k$) at 0.85 V and $E_{1/2}$ of these catalysts were extracted and summarized in Fig. 4e. FeCo-NCH delivers the highest $J_k$ of 12.1 mA cm$^{-2}$, 1.87 and 9.68 times higher than Pt/C (6.46 mA cm$^{-2}$) and FeCo-NC (1.25 mA cm$^{-2}$), respectively. The Tafel plots (Supplementary Fig. 27) of these catalysts are further deduced to get insights into their ORR kinetics. As expected, FeCo-NCH also exhibits enhanced reaction kinetics in terms of the lowest Tafel slope of 74.6 mV dec$^{-1}$, smaller than Pt/C (79.5 mV dec$^{-1}$) and FeCo-NC (93.9 mV dec$^{-1}$), corroborating the superior ORR performance of FeCo-NCH.

The rotating ring-disk electrode (RRDE) measurements were subsequently conducted to evaluate the electron transfer number ($n$)

and HO$_2^-$ yield during the ORR (Supplementary Figs. 28 and 29). FeCo-NCH shows an $n$ value ranging from 3.76 to 3.96 and produces only ~5% HO$_2^-$ in the potential range of 0.4–0.8 V. The slightly higher HO$_2^-$ yield in FeCo-NCH should come from the Co-$N_4$ sites, which is common for Co SACs[55,56]. This demonstrates that the ORR proceeds on the FeCo-NCH mainly via the 4-electron pathway, with is essential for fuel cells. The side 2-electron pathway with the formation of HO$_2^-$ reduces the fuel cell efficiency and degrades the catalysts and membranes, which is detrimental in fuel cells[56]. The genuine catalytic sites in FeCo-NCH for ORR are clarified by the typical SCN$^-$ poisoning experiment[57]. The results indicate that the superior ORR activity should be ascribed to the Fe/Co−$N_x$ sites (Supplementary Fig. 30). Moreover, to evaluate the methanol tolerance, the chronoamperometric responses of FeCo-NCH and Pt/C for ORR at 0.6 V were recorded upon the addition of methanol (Supplementary Fig. 31). Compared with the sharp decrease of current on commercial Pt/C, FeCo-NCH shows tiny current change, suggesting its superior tolerance to methanol crossover effects. Furthermore, the chronoamperometric measurements were carried out to assess the long-term operation durability of the catalysts. As displayed in Fig. 4f, FeCo-NCH retains 95.3% of initial current over 10 h of continuous operation in an $O_2$-saturated electrolyte, suggesting its promising ORR durability. This is corroborated by the CV scanning tests where FeCo-NCH displays only 8 mV decay in $E_{1/2}$ after 10,000 continuous CV cycles (Supplementary Fig. 32).

## AEMFCs and ZABs performance

To demonstrate the significance of boosting the accessible catalytic site density and utilization in achieving high-performance energy devices, the FeCo-NCH, FeCo-NC, and commercial Pt/C were used as cathodic catalysts to assemble AEMFC devices as displayed in Supplementary Fig. 33. The membrane electrode assembly (MEA) was first measured under 2.0 bar $H_2/O_2$ conditions at 80 °C. The polarization and power density curves were shown in Fig. 5a. FeCo-NCH-based MEA delivers a current density of ~1800 mA cm$^{-2}$, significantly higher than FeCo-NC-based one (~500 mA cm$^{-2}$). As an important indicator for fuel cell performance, the maximal peak power density ($P_{max}$) was then measured[8,27]. The corresponding power density curve of FeCo-NCH MEA derived from the $J$-V curve shows a $P_{max}$ of 569.0 mW cm$^{-2}$ at 0.5 V, which is 3.4 times higher than that of FeCo-NC MEA (168.1 mW cm$^{-2}$) at 0.4 V. Although the $P_{max}$ is slightly lower than Pt-based MEA, the FeCo-NCH MEA still shows almost the same performance in the working current density of 1000 mA cm$^{-2}$ for automobile applications. The performance for FeCo-NCH MEA ranks among the highest for non-PGM-based AEMFC cathode catalysts (Supplementary Fig. 34 and Supplementary Table 4)[34,58–61]. The importance of constructing advanced catalyst architecture was also demonstrated by Kumar et al.[62] By constructing three-dimensional (3D) carbon nanotube network with superior electron and mass transfer properties, they reported a FeCoN-MWCNT catalyst with a high $P_{max}$ of 692 mW cm$^{-2}$. Besides, to demonstrate the performance under practical conditions, the FeCo-NCH MEA was also measured under 2.0 bar $H_2$-air (CO$_2$-free) conditions under 80 °C. A current density of ~1000 mA cm$^{-2}$ and a $P_{max}$ of 299.3 mW cm$^{-2}$ were achieved on FeCo-NCH MEA (Fig. 5b), while FeCo-NC MEA shows a much smaller current density of ~270 mA cm$^{-2}$ and a $P_{max}$ of 86.1 mW cm$^{-2}$, these values are also comparable to the reported non-PGM-based cathode catalysts, as summarized in Supplementary Table 4. The performance variation between $H_2$-air and $H_2$-$O_2$ conditions was due to the lower $O_2$ content in air, leading to sluggish mass transport behavior. The design of catalyst architecture to provide greater active site accessibility thus accelerating mass transfer properties is more vital under $H_2$-air conditions[46]. Notably, the FeCo-NCH MEA also demonstrated nearly the same current density as Pt-based MEA in the kinetic region (>0.75 V), effectively illustrating the promoted mass transport properties in FeCo-NCH by the interfacial assembly strategy. Lilloja and Kumar et al. also demonstrated a high-

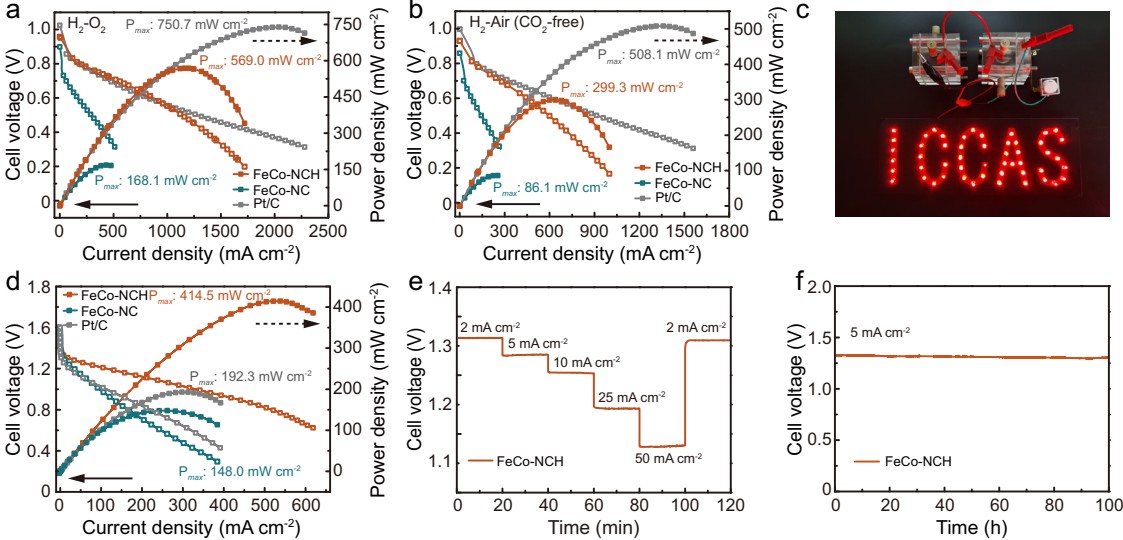

**Fig. 5 | Performance of AEMFCs and Zn − air batteries. a** $H_2$-$O_2$ AEMFC performance of FeCo-NCH and control samples. **b** $H_2$-Air ($CO_2$ free) AEMFC performance of FeCo-NCH and control samples. **c** Optical image of an LED light array powered by two Zn−air batteries in series using FeCo-NCH as the air cathode. **d** Discharge polarization curves and corresponding power density curves of FeCo-NCH-based Zn−air battery and control samples. **e** Discharge curves of the FeCo-NCH-based Zn−air battery at current densities of 2, 5, 10, 25, 50, and 2 mA cm$^{-2}$ for 120 min with each step being 20 min. **f** Discharge curves of FeCo-NCH-based Zn−air battery at 5 mA cm$^{-2}$ for 100 h.

performance $H_2$-air fuel cell by constructing a 3D carbon nanotube network to promote mass transfer (Supplementary Table 4)[56,62]. These results conclude that FeCo-NCH-based AEMFC delivers significantly superior performance to FeCo-NC-based one, which could be safely attributed to the designed nanocage structure of FeCo-NCH with porous shell and hollow interior effectively enabling a much more synergistic Fe/Co-N$_x$ sites in catalyst layer to participate in ORR[8,63].

Besides the fuel cells, aqueous ZAB was selected as the other energy device to demonstrate the merits of the present strategy for promoting site utilization, given it is considered to be an intriguing next-generation energy storage system due to its low cost, high safety, and high energy density[3,64,65]. The FeCo-NCH, FeCo-NC, and conventional Pt/C were used as the cathode catalysts to assemble ZABs with 6 M KOH/0.2 M Zn(AC)$_2$ as the electrolyte and zinc plate as the anode, as shown in (Supplementary Fig. 35). An open-circuit voltage (OCV) of 1.45 V is achieved on the FeCo-NCH-based ZAB (Supplementary Fig. 36), higher than Pt/C ZAB (1.42 V) and FeCo-NC ZAB (1.40 V). Such an OCV enables two FeCo-NCH batteries in series to steadily lighten an LED light array (Fig. 5c). The specific capacity and energy storage capacity of the assembled batteries were then measured at a discharge current density of 20 mA cm$^{-2}$. The FeCo-NCH ZAB exhibits a large specific capacity of 809.2 mAh g$_{Zn}^{-1}$ (Supplementary Fig. 37), corresponding to a gravimetric energy density of 1009.1 Wh kg$_{Zn}^{-1}$, close to the theoretical Zn energy density of 1353 Wh kg$_{Zn}^{-1}$. The discharge performance was subsequently evaluated. FeCo-NCH battery displays a $P_{max}$ of 414.5 mW cm$^{-2}$ (Fig. 5d), which is 2.8 times higher than the FeCo-NC battery ($P_{max}$: 148.0 mW cm$^{-2}$) and 2.16 times higher than Pt/C battery ($P_{max}$: 192.3 mW cm$^{-2}$). Such performance ranks among the highest for state-of-the-art non-PGM ZAB catalysts (Supplementary Fig. 38 and Supplementary Table 6)[66,67]. Furthermore, the FeCo-NCH battery presents impressive discharge rate performance (Fig. 5e). The discharge potential remains stable at each current density from 2 to 50 mA cm$^{-2}$ in the galvanostatic discharge measurements and can be fully resumed when switching the current density from 50 mA cm$^{-2}$ to 2 mA cm$^{-2}$, implying the superior reversibility of the FeCo-NCH battery. The long-term durability was then measured as shown in Fig. 5f. At a discharge current density of 5 mA cm$^{-2}$, the potential of the FeCo-NCH battery remains steady for 100 h, indicating its

impressive durability. The above results demonstrate that both AEMFCs and ZABs with FeCo-NCH as cathodic catalysts show significantly superior performance to control devices.

In summary, we investigated the critical influence of accessible catalytic site density on the performance of electrocatalysts and electrochemical energy devices. Coordination-assisted pyrolysis of binary metal doped ZIF precursors was adopted to produce high-density single atomic Fe/Co-N$_x$ sites. An interfacial assembly strategy was developed to create a nanocage-like catalyst structure with a porous shell and hollow interior so as to bring as many Fe/Co-N$_x$ sites as possible to the interface of electrolyte and catalyst. The single-atomically dispersion and the configuration of the Fe/Co-N$_x$ sites in the prepared FeCo-NCH were confirmed by HAADF-STEM and EXAFS techniques. The total metal loading reached 7.9 wt%. The electrochemical accessible site density was determined to be $7.6 \times 10^{19}$ sites g$^{-1}$ Fe/Co-N$_x$ sites participating in catalyzing ORR, corresponding to a site utilization of 9.3%. This is 3.7 times higher than control FeCo-NC with solid structure ($2.1 \times 10^{19}$ sites g$^{-1}$). Benefiting from such high-density accessible binary synergistic Fe/Co-N$_x$ sites, the FeCo-NCH delivered a superior ORR activity in terms of a half-wave potential of 0.889 V and a limiting current density of 6.2 mA cm$^{-2}$. More importantly, featuring an ultrahigh accessible site density, the Fe/Co-NCH cathode enables AEMFC and ZAB devices to output promising peak power densities of 569.0 mW cm$^{-2}$ and 414.5 mW cm$^{-2}$, respectively. These performances are comparable to the devices based on commercial Pt/C and rank among the best devices based on state-of-the-art non-PGM catalysts. These results demonstrate that the present strategy for boosting the accessible site density via suitable structural design is effective for boosting the performance of AEMFC and ZAB devices, opening up opportunities for developing high-performance energy devices with earth-abundant non-PGM catalysts.

## Methods
### Chemicals and materials
Cobalt (II) nitrate hexahydrate (98%), iron (II) sulfate heptahydrate (98%), zinc nitrate hexahydrate (99%), 2-methylimidazole (99%) zinc acetate dihydrate (98%), Pt/C (20%), dopamine hydrochloride (99%), 3-tris (hydorxymethyl) aminomethane (99.8%-100.1%), sodium thiocyanate (98%), potassium hydroxide (99.98%), sodium acetate

trihydrate (99%), glacial acetic acid (99.9985%), and zinc foil (99.994%) were purchased from Alfa Aesar. Methanol and ethanol were received from Beijing Chemical Work Co. in analytic grade (AR). All chemicals were used as received without further purification. Nafion® solution (5 wt%, DuPont) was obtained from commercial suppliers. Milli-Q ultrapure water (resistance of 18.2 MΩ·cm at 25 °C) was used for all experiments.

## Synthesis of ZIF-8@ZIF-67@Fe@PDA and control samples

In a typical synthesis, the ZIF-8 nanocrystals were first prepared. $Zn(NO_3)_2 \cdot 6H_2O$ (2 g, 6.7 mmol) and 2-methylimidazole (7 g, 0.085 mol) were dissolved in 100 and 50 mL of methanol to form two clear solutions, respectively. The solution of 2-methylimidazole was subsequently poured into the solution of $Zn(NO_3)_2$ with a white precipitate formed immediately. After the mixture was stirred at room temperature for 24 h, the white precipitates were centrifuged and washed with methanol several times, followed by drying overnight at 60 °C. To prepare the ZIF-8@ZIF-67, the achieved ZIF-8 powder (150 mg) was then dispersed in 50 ml of methanol and ultrasonicated for 10 min. The two methanol solutions, containing $Co(NO_3)_2 \cdot 6H_2O$ (0.175 g, 0.6 mmol) and 2-methylimidazole (0.6 g, 7.3 mmol), respectively, were then poured into the ZIF-8 methanol dispersion under vigorous stirring. After stirring at room temperature for 15 min, the Morandi purple precipitates were centrifuged and washed with methanol several times and then dried overnight at 60 °C. To achieve the ZIF-8@ZIF-67@Fe, the prepared ZIF-8@ZIF-67 powder (150 mg) was dispersed in 25 mL of ethanol and ultrasonicated for 10 min of $FeSO_4 \cdot 7H_2O$ (0.05 g, 0.18 mmol) was then added to the dispersion under vigorous stirring, followed by stirring at room temperature for 10 min to adsorb Fe species in the ZIF-67 layer. The Morandi purple precipitates were centrifuged and washed with methanol several times, then dried overnight at 60 °C. Finally, the obtained ZIF-8@ZIF-67@Fe was coated with polydopamine through a typical polymerization process. 200 mg ZIF-8@ZIF-67@Fe was dispersed in a mixed solution of 30 mL water and 80 mL of ethanol under magnetic stirring for 30 min. 40 mL of 100 mM tris (hydorxymethyl) aminomethane solution was then added to the suspension under vigorous stirring. Afterward, 30 mL water containing 60 mg of hydroxytyramine hydrochloride (0.06 g, 0.3 mmol) was added to the suspension and gently stirred for 60 min. The product was centrifuged and washed with water and ethanol, followed by drying overnight at 60 °C. The ZIF-8@ZIF-67@PDA, and ZIF-8@Fe@PDA samples were also prepared in parallel via a similar protocol. A Co, Fe co-doped ZIF-8 was also prepared by a similar protocol with the same metal salt feedstock.

## Synthesis of FeCo-NCH and control samples

The prepared ZIF-8@ZIF-67@Fe@PDA composites were further calcined in a temperature-programmable tube furnace under an Ar atmosphere to obtain the carbonized products. The composites were firstly heated to 200 °C with a heating rate of 2 °C min$^{-1}$ and maintained at the same temperature for 2 h. After that, the temperature was increased to 800 °C at a heating rate of 2 °C min$^{-1}$ and kept for another 2 h. The final product was denoted as FeCo-NCH. For comparison, the Co-NCH, Fe-NCH, and FeCo-NC samples were also prepared following a similar protocol by using the ZIF-8@ZIF-67@PDA, ZIF-8@Fe@PDA, and Co, Fe co-doped ZIF-8 as the precursor, respectively.

## Catalysts characterizations

A Rigaku D/max 2500 with a Cu Kα1 radiation source (λ = 1.54056 Å) was used to collect the X-ray diffraction (XRD) data. The morphologies of as-synthesized materials were investigated via a scanning electron microscope (SEM, HITACHI SU-8020) at an accelerating voltage of 15 kV and a transmission electron microscope (TEM, JEOL, JEM-2100F) at an acceleration voltage of 200 kV. Energy-dispersive X-ray spectra (EDS) were collected on an Oxford Materials Analysis System equipped

on the TEM. The high-angle annular dark-field scanning transmission electron microscopy (HAADF−STEM) was carried out on a JEOL ARM200F (JEOL, Japan) STEM operated at 200 kV with a cold-filled emission gun and double hexapole Cs correctors (CEOS GmbH, Germany). The surface chemical bonding states of the samples were analyzed via X-ray photoelectron spectroscopy (XPS) using a Thermo Scientific ESCALab 250Xi with 300 W monochromatic Mg Kα radiation. $N_2$ adsorption/desorption isotherms of the materials were determined using a Micromeritics ASAP 2460 instrument to obtain Brunauer−Emmett−Teller (BET) specific surface area and pore size distribution. The specific surface area was calculated by the BET method. The elemental compositions were measured by inductively coupled plasma-optical emission spectroscopy (ICP-OES) on ICPE-9000 (Shimadzu).

## XAFS measurement and analysis

XAS experiments were conducted at the Fe K-edge (7112 eV) and Co K-edge (7709 eV) on the beamline 1W1B station of Beijing Synchrotron Radiation Facility (BSRF, operated at 2.5 GeV with a maximum current of 250 mA). A water-cooled Si (111) double-crystal monochromator (DCM) was utilized to monochromatize the X-ray beam and the detuning was done by 20% to remove harmonics. The data were acquired in fluorescence excitation mode using a Lytle detector. Spectra were acquired under room temperature, non-vacuum state, and with no special sample tank. XAS data were processed and analyzed using the Demeter software package. A linear function was subtracted from the pre-edge region, then the edge jump was normalized using Athena software. The χ(k) data were isolated by subtracting a smooth, three-term polynomial approximating the absorption background of an isolated atom. The $k^3$ weighted χ(k) data were Fourier transformed after applying a Hanning window function in a k range of 0−14 Å$^{-1}$. For wavelet transform analysis, the χ(k) exported from Athena was imported into the Hama Fortran code. The parameters were listed as follows: R range, 1−3 Å, k range, 0−14 Å$^{-1}$; k weight, 3; and Morel function with κ = 10, σ = 1 was used as the mother wavelet to provide the overall distribution. The global amplitude EXAFS coordination number, bond length, Debye-Waller factor, and $E_0$ shift (CN, R, σ$^2$, ΔE$_0$), were obtained by nonlinear fitting, with least-squares refinement, of the EXAFS equation to the Fourier-transformed data in R-space, using Artemis (version 0.9.25) software.

## Electrochemical measurements

The electrochemical measurements were conducted on an RRDE-3A (ALS, Japan) and an Autolab PGSTAT302N (Metrohm, Netherlands) electrochemical workstation at room temperature. The catalysts were evaluated in 0.1 M KOH aqueous solution using a conventional three-electrode configuration. The carbon rod was used as the counter electrode and Ag/AgCl electrode as the reference electrode. The working electrode was prepared by drop-casting 12 μL of catalyst ink (2 mg FeCo-NCH/500 μL ethanol) and 0.5 μL of 0.5 wt% Nafion® solution onto a glassy carbon rotating disk electrode (RDE, area: 0.1256 cm$^2$), giving a catalyst loading of 0.33 mg cm$^{-2}$. ORR LSV polarization curves were recorded at a scan rate of 5 mV s$^{-1}$ without iR-compensation. The rotating ring-disk electrode (RRDE) (4 mm in diameter) measurements were also tested in $O_2$-saturated 0.1 M KOH with the studying catalysts as the working electrode. The ring electrode is Pt and the ring potential was kept at 1.26 V. Prior to using, the RRDE was polished with 0.3 and 0.05 μm alumina slurries and then cleaned by subsequently sonicated in Milli-Q water and 2-propanol to a mirror finish state. The Pt ring was cleaned with CV scans from 0.05 to 1.2 V (vs. RHE) for 100 cycles at a scan rate of 50 mV s$^{-1}$. The Chronoamperometric (CA) curve was recorded at a constant potential of 0.56 V (vs. RHE) for ORR. The LSV curves were recorded without iR-compensation. All potential reported in this work were referred to the

RHE, which was converted according to the Nernst equation as follows:

$$E_{RHE} = E_{Ag/AgCl} + 0.198 + 0.059 \times pH \qquad (1)$$

Where $E_{Ag/AgCl}$ is the measured potential.

The Koutecky-Levich (K-L) equations were used to analyze kinetic parameters as follows:

$$\frac{1}{J} = \frac{1}{J_K} + \frac{1}{J_L} = \frac{1}{J_K} + \frac{1}{B\,\omega^{-1/2}} \qquad (2)$$

$$B = 0.2nFC_0 D_0^{2/3} \nu^{-1/6} \qquad (3)$$

Where $J$ represents the measured current density; $\omega$ is the electrode rotating rate; $F$ is the Faraday constant (96485 C mol⁻¹); $J_k$ is the kinetic current density; $C_O$ is the bulk concentration of $O_2$ 0.1 M KOH ($1.2 \times 10^{-6}$ mol cm⁻³); $D_O$ is the oxygen diffusion coefficient ($1.9 \times 10^{5}$ cm² s⁻¹) of 0.1 M KOH, and $v$ is the kinematic viscosity of 0.1 M KOH (0.01 cm² s⁻¹).

The following formula was applied to calculate the $HO_2^-$ yield ($HO_2^-$%) and the electron transfers number ($n$), respectively:

$$HO_2^-\% = \frac{\frac{2I_r}{N}}{\frac{I_r}{N} + I_d} \times 100\% \qquad (4)$$

$$n = \frac{4I_d}{\frac{I_r}{N} + I_d} \qquad (5)$$

where $I_d$ and $I_r$ represent disk and ring current, respectively, and $N$ is the ring collecting efficiency (0.37).

## Quantification of the active sites

The site density (SD) was obtained according to the method presented by Kucernak et al.[18] Briefly, extensive electrochemical cycling was performed on the catalyst alternatively in $O_2$ and $N_2$ in acetate buffer (pH 5.2) to give consistent cyclic voltammetry curves in $N_2$. The catalyst was then poisoned by $NaNO_2$. The ORR performance was recorded before, during, and after the nitrite adsorption. Nitrite stripping was conducted in the potential region of 0.35 to −0.35 V vs. RHE. The excess in cathodic charge ($Q_{strip}$) is proportional to the SD.

$$SD[mol\,g^{-1}] = \frac{Q_{strip}[C\,g^{-1}]}{n_{strip}F[C\,mol^{-1}]} \qquad (6)$$

$$\text{Utilization of sites}\,(\%) = SD[mol\,g^{-1}] \times \frac{M[g\,mol^{-1}]}{W} \qquad (7)$$

The turnover frequency (TOF) was calculated by the following equation:

$$TOF[e^-\,s^{-1}site^{-1}] = \frac{J_k(mA\,cm^{-2}) \times N_e}{W \times C_{cat} \times N_A/M} \qquad (8)$$

where $n_{strip}$ (=5) is the number of electrons associated with the reduction of one nitrite per site; $F$ is the Faraday constant (96485 C mol⁻¹); $J_k$ is the kinetic current density, $N_e$ is the electron number per Coulomb ($6.24 \times 10^{18}$), $W$ is the metal contents of FeCo-NCH or FeCo-NC. $C_{cat}$ is the catalyst loading on the electrode. $N_A$ is the Avogadro constant ($6.022 \times 10^{23}$), $M$ is the average atomic mass of Fe and Co for FeCo-NCH or FeCo-NC (based on the ratio of Fe and Co).

## Density functional theory (DFT) calculations

All DFT calculations were performed by using Vienna ab initio Simulation Package (VASP)[68]. The generalized gradient approximation (GGA) with the Perdew-Burke-Ernzerh (PBE)[69] of the exchange-correlation functional within the projector augmented wave method[70] was utilized to model the electron-ion interaction. An energy cutoff of 450 eV for the plane-wave basis set was used. The convergence threshold was set to $10^{-5}$ eV in energy and 0.02 eV/Å in force, respectively. To prevent the interaction between two neighboring images, the vacuum layer thickness was set to 15 Å. A semi-empirical van der Waals (vdW) correction proposed by Grimme (DFT-D3)[71] was included to account for the dispersion interactions. A $2 \times 2 \times 1$ Monkhorst-Pack grid of k-points was used to sample the first Brillouin zones of the surfaces for structural optimizations. All atoms are fully relaxed during structural optimizations.

The free energy of each elementary step in the proton-coupled electron transfer (PCET) reactions was computed using the computational hydrogen electrode (CHE) model for oxygen reduction reaction (ORR). Considering the $O_2$ molecule was not broken before reduction, the associative 4e- reduction pathway was evaluated to be most feasible for ORR in this work, as follows:

$$* + O_2 + H_2O + e^- \rightarrow *OOH + OH^- \qquad (9)$$

$$*OOH + e^- \rightarrow *O + OH^- \qquad (10)$$

$$*O + H_2O + e^- \rightarrow *OH + OH^- \qquad (11)$$

$$*OH + e^- \rightarrow * + OH^- \qquad (12)$$

where * represents the active site on the corresponding surface.

The free energy of the reactants and each intermediate state at an applied electrode potential U are computed by the equations, $\mu = E + E_{ZPE} - TS$, $\Delta G = \Delta E + \Delta E_{ZPE} - T\Delta S$, then the $\Delta G_{*O}$, $\Delta G_{*OH}$ and $\Delta G_{*OOH}$ could be calculated as follows:

$$\Delta G_O^* = \mu_O^* - \mu^* - (\mu_{H_2O} - \mu_{H_2}) \qquad (13)$$

$$\Delta G_{OH}^* = \mu_{OH}^* - \mu^* - \left(\mu_{H_2O} - \frac{1}{2}\mu_{H_2}\right) \qquad (14)$$

$$\Delta G_{OOH}^* = \mu_{OOH}^* - \mu^* - \left(2\mu_{H_2O} - \frac{3}{2}\mu_{H_2}\right) \qquad (15)$$

The Gibbs free energy of step 1-4 reaction could be expressed as:

$$\Delta G_1 = \Delta G_{OOH}^* - 4.92\,eV + eU \qquad (16)$$

$$\Delta G_2 = \Delta G_O^* - \Delta G_{OOH}^* + eU \qquad (17)$$

$$\Delta G_3 = \Delta G_{OH}^* - \Delta G_O^* + eU \qquad (18)$$

$$\Delta G_4 = -\Delta G_{OH}^* + eU \qquad (19)$$

## Membrane-electrode-assembly (MEA) and fuel cell tests

The AEMFC performance with FeCo-NCH cathode was evaluated in an 850e fuel cell test system (Scribner Associates). The PtRu/C (40 wt% Pt and 20 wt% Ru on Vulcan XC-72R, HiSpec 10000) were used as the anode catalyst. The hydrogen exchange membrane and ionomer (PAP-TP-x, x is the molar ratio between N-methyl-4-piperidone and terphenyl monomers) was synthesized as reported by Yan et al.[10,72] FeCo-NCH was used as the cathode catalyst with a loading of 1 mg cm⁻². The

catalysts (with 1 mg Vulcan XC-72R per 10 mg catalyst) and ionomer (PAP-TP-100, 5 wt% in ethanol for performance test, the dry weight of the ionomer accounted for 45% of the catalyst mass) were dispersed in the mixture of water and isopropanol (1:20 v/v) via sonication for 2 h in an ice water bath. The catalyst ink was then sprayed on the PAP-TP-85 membrane (Cl$^-$ form, $20 \pm 5$ μm in the dry state), forming a catalyst-coated membrane (CCM) with an electrode area of 5 cm$^2$. The metal loading of PtRu in the anode was 0.2 mg cm$^{-2}$. Next, the prepared CCM was soaked in 3 M KOH for 24 h at 60 °C to exchange Cl$^-$ with OH$^-$, and washed with distilled water before fuel cell tests to remove the excess KOH. The resulting CCM was positioned between two pieces of Teflon-treated carbon paper (SIGRACET, 29BC) to make the membrane electrode assembly. H$_2$/O$_2$ single-cell AEMFCs were tested at 80 °C. H$_2$ and O$_2$ were fully humidified at 80 °C (100% RH) and fed with a flow rate of 1000 mL min$^{-1}$ and a backpressure of 200 kPa symmetrically on both sides. H$_2$/Air single-cell AEMFCs were tested at 80 °C. H$_2$ and air were fully humidified at 80 °C (100% RH) and fed with a flow rate of 1000 mL min$^{-1}$ and a backpressure of 100 kPa symmetrically on both sides.

**Zn-air battery tests.** Zn-air battery tests were performed in a home-made Zn-air battery. A polished Zn foil was used as the anode and the gas diffusion electrode (GDE) coated with the catalysts was used as the air cathode. A mixed solution of 0.2 M Zn(AC)$_2$ and 6 M KOH was used as the electrolyte. The catalyst ink was prepared by mixing 1 mg catalysts with 40 μL 0.5% Nafion® solution in 800 μL ethanol, followed by a 30 min ultrasonic treatment. It was then drop-casting onto a Teflon-coated carbon fiber paper with a catalyst loading of 1.0 mg cm$^{-2}$ as the GDE. The achieved GDE was dried in a vacuum oven at 60 °C overnight. A control sample was also prepared by using 1 mg cm$^{-2}$ commercial Pt/C catalyst. The discharge curves were measured on the Autolab PGSTAT302N (Metrohm, Netherlands) electrochemical workstation, the durable tests were carried out on the LAND CT2001A instrument. The current densities were normalized to the effective surface area of the air electrode.

## Data availability

The data supporting the findings of this work are available within the article and its Supplementary Information files. Source Data are provided with this paper, and all the data reported in this work are available from the authors on request. Source data are provided with this paper.

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

## Acknowledgements

The authors acknowledge the financial support from the National Key Research and Development Program of China (2020YFB1505801), the National Natural Science Foundation of China (22025208, 22075300, 22162006, and 22202212), the China National Postdoctoral Program for Innovative Talents (BX2021319), the DNL Cooperation Fund, CAS (DNL202008), Guangxi Natural Science Foundation of China (2019GXNSFGA245003), and the Chinese Academy of Sciences. We also thank Dr. Zhi-Juan Zhao, Xiao-Yu Zhang, and Bao-Long Qu for the XPS analysis; Yang Sun for the XRD analysis; Dr. Bo Guan, Yong-Xin Cheng, and Ji-Ling Yue for SEM and TEM support.

## Author contributions

Z.J., T.T., and J.S.H. conceived the project. Z.J. carried out the synthesis, most of the structural characterizations, and electrochemical tests. X.R.L. and Z.B.Z. performed the fuel cell measurements. X.Z.L., Q.H.Z., and L.G. performed the HAADF–STEM characterizations. Z.J. and L.R.Z. analyzed the EXAFS and XANES data. S. H. and L. L. performed the DFT calculations. Y.L., Z.C.Y., Y.Z., J.N.Z., and Y.J.F. discussed the results and commented on the manuscript. Z.J., T.T., and J.S.H. analyzed the data and co-wrote the paper. J.S.H. supervised the project.

## Competing interests

The authors declare no competing interests.
