## [Peer Review File · Nature Communications]

Interfacial Assembly of Binary Atomic Metal-N_x Sites for High-Performance Energy DevicesREVIEWER COMMENTS

Reviewer #1 (Remarks to the Author):

This paper reports the preparation of Fe-N-C catalysts via the pyrolysis of ZIF-8-based multiple core@shell structures for use as the cathodic oxygen reduction reaction catalysts for anion exchange membrane fuel cells (AEMFCs) and zinc-air batteries. The catalysts were prepared by coating of pre-synthesized ZIF-8 in sequence with co-based ZIF-67, Fe²⁺ ion, and polydopamine (ZIF-8@ZIF-67@Fe@PDA) and pyrolysis at high temperature. The resulting FeCo-NCH catalyst had a hollow interior and contained high metal (Fe+Co) contents of 7.9 wt%. Comparatively, the same syntheses without the addition of ZIF-67 or Fe²⁺ resulted in the Fe-NCH and Co-NCH catalysts, respectively. The synthesis in the absence of PDA coating afforded the FeCo-NC catalyst with a solid interior. Fe and Co-codoped FeCo-NCH showed higher ORR activity than other control samples and commercial Pt/C catalyst. Also, the AEMFC and Zn-air battery employing FeCo-NCH as the cathode catalyst exhibited superior performance compared to other control sample-based devices.

While this work is well-organized work in its own sake, both the synthesis strategy and catalytic performances in this work do not represent significant advances, expected for a paper in Nature Comm. A materials- or catalysis-specialized journal appears more suitable forum for this work.

- 1) Why were hollow structures generated from the pyrolysis of PDA-coated composites? Mechanistic consideration on the formation of hollow structures should be discussed, including the role of PDA coating.
- 2) The formation of porous or hollow architectures via the pyrolysis of ZIF-8-based structures have already been reported, with notable examples by the groups of G. Wu and J. Shui.
- 3) In the FeCo-NCH catalyst, Fe- and Co-containing carbon layers were formed sequentially but not simultaneously; that is, they can be hardly present in proximity. Then, can the synergistic effects between the Fe-N_x and Co-N_x species evolve in this catalyst?
- 4) The half-wave potential of FeCo-NCH was 0.89 V (vs. RHE) in alkaline media. This value is comparatively better than other control catalysts. However, this is far from being impressive. Indeed, many catalysts having half-wave potentials even higher than 0.9 V are reported, some of which are included in the Table in SI.
- 5) TOFs of the catalysts should be quantified, and comparatively discussed with those of previous works.

Reviewer #2 (Remarks to the Author):

Recently, many such single atom Fe-N-C catalysts have been reported for the oxygen reduction reaction in the alkaline media for fuel cells or metal-air batteries. One of the most impressive works is in Nature Energy: <https://www.nature.com/articles/s41560-021-00878-7>.

Compared to the above work, the manuscript reports binary atomic metal-N_x sites containing Fe and Co with enhanced catalytic activity. However, based on current characterization and spectroscopy, it is unclear what are the interactions between Fe and Co atoms in the catalysts. Therefore, the possible

promotional mechanisms remain unknown yet. Furthermore, the reporting performance in anion exchange member fuel cells is not compelling yet, especially compared to the work reported in Nature Energy by Adabi et al. Also, the synthesis methods to combine two types of single atoms sites in catalysts derived from ZIF-8 and ZIF-67 do not demonstrate sufficient innovation and creativity. Overall, this is routine work to develop a single atom catalyst for fuel cells and metal-air batteries, which cannot justify its qualification to be published in the journal.

Reviewer #3 (Remarks to the Author):

In this manuscript, the authors developed an interfacial assembly strategy to boost the exposure of catalytically active sites for enhancing the device performance. The high-density accessible atomically dispersed metal-nitrogen active sites for oxygen reduction reaction were achieved by introducing binary Fe/Co-N_x sites and confining them in the ultrathin porous shell structures. Compared with the solid counterparts, the electrochemically accessible active sites were augmented by 3.7 times, resulting in the 3.4 times increase in the peak power densities for fuel cells and 2.8 times increase for Zn-air batteries. These results are impressive and will inspire more attention on the design of catalyst structure for boosting the effective active sites to enhance the device performance. The manuscript was well-organized and presented comprehensive data to support the main conclusions. The findings may also shed lights on the design of other catalysts with highly exposed active sites for various applications. Based on the above facts, the reviewer would recommend publishing this study in this journal after addressing the following comments.

1. It is mentioned that the formation of the hollow structure was caused by the stress-induced strategy due to the difference in the pyrolysis conditions between PDA and ZIF. The author should provide detailed information about the pyrolysis process and the related data for further discussion of this phenomena.
2. According to the manuscript, the formation of the ZIF-PDA core-shell structure is important for the formation of mesoporous in the final hollow structured catalysts. More discussion is needed for the role of PDA in the formation of such structure.
3. Why did author choose PDA as the coating layer? Are there other materials which can achieve the similar structure?
4. In this manuscript, the author mentioned that the Fe/Co-N_x binary atom catalyst synthesized by this strategy achieved a 7.9 wt% of metal loading. Considering that polydopamine also contains the N element, therefore, does the extra N species from the pyrolysis of polydopamine contribute to the increase in metal loading of the catalyst?
5. The author introduced both Fe and Co sources to achieve binary atomic Fe/Co-N_x sites. What are the merits for the binary metal-nitrogen structure? Can the author achieve the similar metal loading with single metal source?
6. The author mentioned that the number of the metal-N_x sites on the surface of the catalyst can be assessed by in-situ electrochemical nitrite stripping. Can the author provide the relevant fundamental

principle involved in the method?

7. The detailed preparation process of the GDL for the gas diffusion layer (GDE) and the membrane-electrode-assembly (MEA) for the fuel cell should be provided in the manuscript.
8. Some typos and spelling errors should be corrected in this manuscript, for example, the extra "32" mark in the Y-axis of Figure 3b; "0.5 μ L of 0.5 Nafion solution" in the Method section, etc.
9. Some of recent progress in the field should be supplemented.

Reviewer #4 (Remarks to the Author):

Developing ORR catalysts with high intrinsic sites and dense accessible site density is critical for boosting the performance of practical energy devices but remains a great challenge. This manuscript reported a stress-induced interfacial assembly strategy for synthesizing a dual atomic catalyst with a highly accessible site density of 7.6×10^{19} sites g^{-1} . The 3-fold improvement of the accessible site density for as-obtained FeCo-NCH catalyst is very impressive. Detailed physical characterizations and electrochemical measurements were conducted and the synthesis process was fully elaborated. The practical devices employing FeCo-NCH cathode exhibited remarkably improved power densities of 569 $mW\ cm^{-2}$ for AEMFC and 414.5 $mW\ cm^{-2}$ for ZAB compared with the control ones, which are also among the state-of-the-art non-PGM ORR catalysts. These results prove the feasibility of the interfacial assembly strategy to improve the performance of SACs in practical energy conversion devices. Based on these facts, this reviewer would like to recommend its publication in Nat. Commun. after addressing the following comments.

1. The author proposed a stress-induced interfacial assembly strategy to boost the accessible site density, the stress-induced effect should be explained in the manuscript.
2. To validate the advantages of constructing interfacial assembly active sites, the metal contents of the single-metal-based control samples should be also included in Supplementary Table 3.
3. Some catalysts listed in Supplementary Table 2 are not derived from MOF-based precursors. The authors should check this table again.
4. The as-prepared catalyst was denoted as FeCo-NCH (line 75). What does H stand for?
5. The XRD pattern of ZIF-8@Fe was given in Supplementary Figure 2. However, ZIF-8@Fe was not mentioned elsewhere throughout the manuscript. The authors should revise Supplementary Figure 2 accordingly.
6. To prepare the ZIF-8@ZIF-67@Fe, the ZIF-8@ZIF-67 and Fe^{2+} ions were dispersed in EtOH and the mixture was stirred at RT for 10 min. In this process, Fe^{2+} ions should be adsorbed on the surface instead of entering the ZIF skeleton. Therefore, the statement of "Fe substitution" is inaccurate.
7. The name of the same precursor should be unified throughout the manuscript and also the supplementary material (ZIF-8@ZIF-67, rather than ZIF-8@ZIF67).
8. The authors should correct some mistakes in the manuscript, including: the ordinate of Figure 3b; the caption of Supplementary Figure 4; "adsorption" and "desorption" in Supplementary Figure 16a and 17a; "0.1256 cm^{-2} " (line 407).

Reviewer #5 (Remarks to the Author):

Manuscript: NCOMMS-22-33399

“Boosting the Accessible Site Density by Interfacial Assembly of Binary Atomic Metal–Nx for High-Performance Energy Devices”, by Zhe Jiang, Xuerui Liu, Xiao-Zhi Liu, Ying Liu, Ze-Cheng Yao, Yun Zhang, Qing-Hua Zhang, Lin Gu, Li-Rong Zheng, Youjun Fan, Tang Tang, Zhongbin Zhuang, Jin-Song Hu

In this work the electrocatalyst materials with dual-metal M-Nx active sites are prepared and tested for ORR performance in alkaline media using the rotating (ring)-disk electrode methods. These catalysts are employed as cathode materials in anion exchange membrane fuel cells and Zn-air batteries. The results obtained are important in the field of ORR electrocatalysis on non-PGM catalysts. The catalyst materials are thoroughly characterized by various physico-chemical methods. A discussion about the electrochemical properties of the studied catalysts needs further elaboration. The authors should determine the turnover frequency values and compare these with literature data.

Overall. The paper is suitable for publication in this Journal. Recommendation: major revision

Comments to the authors:

- 1) Page 2, line 59. Refs. 17 and 18 do not contain the information presented, thus appropriate references are needed for ORR electrocatalysis on dual-atom M-Nx sites. Also, a discussion about the benefits of this type of bimetallic catalyst materials should be further elaborated.
- 2) Page 2, line 35. “Such FeCo-NCH cathode electrocatalyst enables AEMFC or ZAB to deliver outstanding peak power densities of 569.0 or 414.5 mW cm⁻²,” This reviewer does not think that 569 mW cm⁻² is an outstanding peak power density for AEMFCs. Please re-phrase.
- 3) Page 2, line 50. Important literature is missing. See, for example, recent reviews by Sarapuu et al. DOI: 10.1039/c7ta08690c and Hossen et al. <https://doi.org/10.1016/j.apcatb.2022.121733>
- 4) Page 2, lines 60-62. Reference 19 is not relevant, there are plenty of review articles available.
- 5) Page 2, line 62. “It is due to not only the insufficient intrinsic activity of M–Nx sites...” These sites are considered to be the most active ones in alkaline media. This statement is not justified.
- 6) XPS survey spectra of materials should be included in the manuscript as well and discussed.
- 7) Page 6, lines 196-198. These findings should be discussed more in depth. Why in bimetallic material the overall metal and just Co loading can be higher without agglomerate formation?
- 8) Page 9. The authors should determine the turnover frequency (TOF) values of the catalysts for ORR at a specific potential (0.9 or 0.85 V vs. RHE) and compare these with literature data. The TOF value shows the intrinsic ORR activity of the M-Nx sites and is therefore highly relevant.
- 9) Page 9, lines 268-271. “The smaller Tafel slope means the faster ORR process.⁵¹ As expected,...” This claim is not justified. The authors should recall the meaning of Tafel slope in electrochemical kinetics (see, for example, any basic textbook of electrochemical kinetics).
- 10) Page 9, 2nd paragraph. The mechanistic aspects of the ORR on dual-metal M-Nx sites need to be properly discussed. At present it remains unclear why these materials are highly ORR active.
- 11) Page 9, 3rd paragraph. By comparing the AEMFC results with literature data, the author should make

reference to previous studies in which FeCo-N-C materials were used as cathode catalyst.

12) The RRDE measurements. The experimental details of RRDE measurements are lacking, e.g. (i) which ring electrode was used? (ii) at which potential the ring electrode was kept; (iii) was there any pretreatment of the ring electrode used prior to the ORR measurements? These details of the RRDE measurements need to be added. Also, the obtained ring currents are unreadable in Figure S20, which needs to be fixed. Use a proper scale for the ring current values.

13) The SCN⁻ anion poisoning test was conducted in acidic media. Why not in alkaline solution?

14) Comparison with H₂/air AEMFC results from the literature should be added.

15) The Co-NCH catalyst seems to have rather similar ORR electrocatalytic activity to FeCo-NCH, but the performance of Fe-NCH is significantly worse. What could be the reason for that?

16) Figures 5a,d,g. These Figures are not informative and should be removed from the manuscript.

17) Page 2, line 44. A recent review by Hussain et al. dealing with ORR electrocatalysis on Pt-based catalysts is missing (<https://doi.org/10.1016/j.ijhydene.2020.08.215>). It needs to be cited.

18) In Supplementary Tables S2, S4 and S5 the references should be replaced with number and the full list of references should be given in the end of the Supplementary Information.

Minor remarks:

Page 2, line 24. "M-N-C catalyst" should be the last keyword.

Page 2, line 36. "...3.4 or 2.8 times higher than control devices." As the control devices are not specified, then this part needs re-phrasing or removing.

Page 2, line 60. "...shown excellent ORR activities on rotating disk electrodes (RDE)," should be re-phrased.

In alkaline media HO₂⁻ forms instead of H₂O₂. This should be corrected throughout the manuscript.

Page 8, line 236. "AEMFC discharge performance" should be "AEMFC performance"

Page 8, line 334. "and energy devices" should be "and electrochemical energy devices"

Page 10. The origin and purity of used chemicals is missing, this information should be added.

Page 10, line 315. "high than" should be replaced with "higher than"

Page 11, line 407. "0.5 Nafion solution" What is meant here? Also, it should be "glassy carbon", not "glass carbon". The unit of surface area is also wrong. Please correct.

Page 11, line 413. "The K-L equations" should be "The Koutecky-Levich (K-L) equation"

Experimental details of H₂/air AEMFC are missing.

Tables S4 and S5. There are some errors, so the authors should check the data in these Tables and correct it (e.g. Fe-N-Gra was tested at 60°C not 80°C; CoFe-N-CDC should be CoFe-N-CDC/CNT).

The value of the electrode rotation rate used is missing from Figure S15, S20 and S22 captions.

Responses to the Reviewers' Comments

Responses to the Reviewer #1's comments

Reviewer #1:

This paper reports the preparation of Fe-N-C catalysts via the pyrolysis of ZIF-8-based multiple core@shell structures for use as the cathodic oxygen reduction reaction catalysts for anion exchange membrane fuel cells (AEMFCs) and zinc-air batteries. The catalysts were prepared by coating of pre-synthesized ZIF-8 in sequence with co-based ZIF-67, Fe²⁺ ion, and polydopamine (ZIF-8@ZIF-67@Fe@PDA) and pyrolysis at high temperature. The resulting FeCo-NCH catalyst had a hollow interior and contained high metal (Fe+Co) contents of 7.9 wt%. Comparatively, the same syntheses without the addition of ZIF-67 or Fe²⁺ resulted in the Fe-NCH and Co-NCH catalysts, respectively. The synthesis in the absence of PDA coating afforded the FeCo-NC catalyst with a solid interior. Fe and Co-codoped FeCo-NCH showed higher ORR activity than other control samples and commercial Pt/C catalyst. Also, the AEMFC and Zn-air battery employing FeCo-NCH as the cathode catalyst exhibited superior performance compared to other control sample-based devices.

While this work is well-organized work in its own sake, both the synthesis strategy and catalytic performances in this work do not represent significant advances, expected for a paper in Nature Comm. A materials-or catalysis-specialized journal appears more suitable forum for this work.

Brief response:

We appreciate your valuable time and comments below, which have helped us to improve the quality of our manuscript.

In view of mass transfer issues, boosting the density of accessible catalytic sites is essential for enhancing the performance of energy devices featuring the assembled catalyst layer structure. Such requirement is not limited to the fuel cells and metal-air batteries, but also to other devices such as membrane electrode-based water electrolyzers and CO₂ electrolyzers. Compared with the intensive efforts on improving the intrinsic activity of electrocatalysts, this requirement received much less attention and is still a challenge.

By presenting a dual atomic catalyst with a highly accessible site density of 7.6×10^{19} sites g⁻¹ for ORR and comparing it with control catalysts, we reported here an effective strategy for substantially boosting the accessible site density via interfacial assembly synthesis and demonstrated its significance in promoting the device performance for fuel cells and Zn-air batteries. We believe that such a strategy and the findings are generally applicable and helpful for the design of other energy-related electrocatalysts, which would be appealing to the community and a broad range of readerships for a comprehensive journal like *Nat. Commun.*. By addressing the comments you raised with supplemented data and evidence, we hope you could support the publication of this manuscript.

Reproduced Comments and Responses:

Comment 1:

Why were hollow structures generated from the pyrolysis of PDA-coated composites? Mechanistic consideration on the formation of hollow structures should be discussed, including the role of PDA coating.

Response 1:

We appreciate this constructive suggestion. We accordingly carried out a series of experiments to investigate this process. With the following data, we believe that the PDA coating plays a role like a soft template during pyrolysis. Without the PDA coating layer, the precursor (ZIF-8@ZIF-67@Fe as shown in Supplementary Fig. 1c,f) tends to decompose and shrinks into concave nanoparticles with a solid structure in a smaller size (Supplementary Fig. 5). The supplemented thermogravimetric analysis (newly added Supplementary Fig. 3) shows that the PDA starts to decompose at the very early stage of the pyrolysis, and formed a relatively rigid carbon shell at the initial stage. The decomposition of the inner ZIF component begins at a temperature of around 400 °C. Given the interaction between the partly carbonized PDA shell and the ZIF-8 phase, the outer relatively rigid carbon shell restrains the contraction of the inner ZIF component during the pyrolysis. As the decomposition of the ZIF component goes with the temperature rising, the stresses from the shell induce the accumulation of pyrolyzed species on the shell, thus resulting in the hollow structure after the pyrolysis.

Such formation process was validated by a series of TEM images taken from the various decomposing stages (newly added Supplementary Fig. 4).

Supplementary Figure 1. SEM and TEM images of (a,d) ZIF-8, (b,e) ZIF-8@ZIF-67, and (c,f) ZIF-8@ZIF-67@Fe.

Supplementary Figure 5. (a-b) SEM and (c-d) TEM images of the control sample FeCo-NC prepared without PDA coating.

The corresponding discussion has been supplemented in the revision and the detailed analyses were supplied in the **Supplementary Information**.

In the revision (highlighted in red, page 4, first paragraph).

“During the pyrolysis, the relatively rigid carbonized PDA shell (Supplementary Fig. 3) formed at the very early stage acted as a framework to restrain the contraction of the inner ZIF component and induced the accumulation of pyrolyzed species on the shell, resulting in the final hollow structure. The Fe, Co-containing ZIF-67 layer was accordingly transformed to the carbon shell with neighboring Fe-N and Co-N moieties. Such formation process was validated by a series of TEM images taken from the various decomposing stages (see details in Supplementary Fig. 4). Without the assistance of the PDA coating, the pyrolyzed product ended up with a solid nano-polyhedral morphology with a concaved surface (Supplementary Fig. 5).”

The following **Supplementary Figures 3 and 4** and the discussion have been added in the revised **Supplementary Information** (highlighted in red, page 5).

The PDA starts to decompose at the very early stage of pyrolysis and formed a relatively rigid carbon structure at the initial stage. The decomposition of the inner ZIF component begins at a temperature over 440 °C. Given the interaction between the partly carbonized PDA shell and the ZIF-8 phase, the outer relatively rigid carbon shell restrains the contraction of the inner ZIF component during the pyrolysis. As the decomposition of the ZIF component goes with the temperature rising, the stresses from the shell induce the accumulation of pyrolyzed species on the shell, thus resulting in the hollow structure after the pyrolysis.

Supplementary Figure 3. TGA plots for ZIF8, PDA, and ZIF-8@ZIF-67@Fe@PDA composites. The tests were conducted in an N₂ atmosphere with a heating rate of 10 °C min⁻¹.

Supplementary Figure 4. TEM images collected at various temperatures during the pyrolysis for the preparation of FeCo-NCH. (a) 300 °C, (b) 400 °C, (c) 450 °C, (d) 500 °C, (e) 550 °C, (f) 650 °C, (g)

750 °C, and (h) 800 °C.

The morphology evolution of the FeCo-NCH during the pyrolysis was investigated in detail. It is found the composite surface becomes rougher but the inner remains in a solid structure at the low temperature of 300 °C, consistent with the TGA results that PDA decomposes before the ZIF component. As the temperature goes up to 400 °C at which ZIF starts to decompose, a small hollow appears, which grows as the temperature rises to 450 and 500 °C. The hollow structure formed at the temperature of 550 °C. The dark spots can be attributed to Zn-containing species from ZIF-8. As the temperature keeps rising, the carbon shell becomes thinner and the Zn species evaporate from the product. The final carbonized hollow structure was maintained at 800 °C.

Comment 2:

The formation of porous or hollow architectures via the pyrolysis of ZIF-8-based structures have already been reported, with notable examples by the groups of G. Wu and J. Shui.

Response 2:

Thanks for the information. The ZIF structure bridges metal atoms and ligands into 3D-ordered crystal frameworks with rich micropores and high surface areas. Such unique features make ZIF structures widely serving as templates to prepare high-performance ORR catalysts. Although many efforts have been devoted to improving the ZIF-derived catalysts, the current ORR performance is still unsatisfactory at the device level. The research on this topic is still important and necessary.

Although some of the studies reported the construction of porous or hollow structures, the synthetic process usually involves hard templates like SiO₂, which requires multi-step complicated processing, such as the structures reported by Prof. Wu and Prof. Shui et. al. Moreover, the metal loadings in the state-of-the-art ZIF-based hollow structures are still relatively low (for instance: ~ 3 wt% Fe loading in TPI@Z8(SiO₂)-650-C, reported by Shui et al. Nat. Catal. 2, 259-268 (2019)). Compared with those reports, the synthetic method in this manuscript requires only one-step pyrolysis with no need for the hard template and its removal via acid leaching. This is more suitable for the scale-up synthesis. More importantly, the coordination-assisted adsorption together with PDA-assisted pyrolysis can achieve high metal loading (i.e. 7.9 wt% for Fe+Co in this case), implying the advance of our strategy in preparing high-loading single atomic catalysts. Meantime, the present coordination-assisted adsorption enables a combination of two or more types of single-atomic catalytic sites for further improving the intrinsic activity of catalysts. We believe this synthetic strategy and new insights into boosting the accessible catalytic site density presented in this manuscript would advance the design of efficient electrocatalysts for high-performance energy devices.

Comment 3:

In the FeCo-NCH catalyst, Fe- and Co-containing carbon layers were formed sequentially but not simultaneously; that is, they can be hardly present in proximity. Then, can the synergistic effects between the Fe-N_x and Co-N_x species evolve in this catalyst?

Response 3:

Please see our corresponding responses to these two comments below.

- a) In the FeCo-NCH catalyst, Fe- and Co-containing carbon layers were formed sequentially but not simultaneously; that is, they can be hardly present in proximity.

The Fe²⁺ ions were introduced into the outer ZIF-67 layer in ZIF-8@ZIF-67 during the impregnation, giving the Fe-doped ZIF-67 layer. The Co and Fe species exist in the same ZIF-67 layer of the precursors. During pyrolysis, the Fe, Co-containing ZIF-67 layer was simultaneously transformed to the corresponding Fe, Co-containing species in the same thin carbon shell. It can be reasonably believed that there is a large probability that the Fe- and Co-species exist in proximity. To avoid any confusion, we have revised the corresponding description in the main text and supplemented the explanation in the Methods section.

In the revised main text (highlighted in red, page 3, last paragraph):

“To achieve binary metal sites, Fe adsorption was further carried out on ZIF-8@ZIF-67 via a facile impregnation (ZIF-8@ZIF-67@Fe) so that both Co and Fe species exist in the ZIF-67 layer”

and in the revised main text (highlighted in red, page 4, first paragraph):

“The Fe, Co-containing ZIF-67 layer was accordingly transformed to the carbon shell with neighboring Fe-N and Co-N moieties.”

In the revised **Methods** section (highlighted in red, page 11):

“To achieve the ZIF-8@ZIF-67@Fe, the prepared ZIF-8@ZIF-67 powder (150 mg) was dispersed in 25 mL of ethanol and ultrasonicated for 10 minutes. 50 mg of FeSO₄·7H₂O was then added to the dispersion under vigorous stirring, followed by stirring at room temperature for 10 min to adsorb Fe species in the ZIF-67 layer.”

- b) Then, can the synergistic effects between the Fe-N_x and Co-N_x species evolve in this catalyst?

We appreciate the constructive comment. To understand the synergistic effects between the Fe-N_x and Co-N_x species, we have performed the density functional theory (DFT) calculations to reveal the origin of the improved catalytic performance for ORR.

On the basis of the characterization results of XPS and EXAFS, we first constructed three catalyst models: FeN₅, CoN₄, and FeN₅-CoN₄ model, respectively (Supplementary Fig. 22). The ORR process on the Co sites and Fe sites were discussed respectively. The adsorption models of various intermediate states on three models with different sites during the ORR process were optimized as shown in Supplementary Fig. 23 (Co sites) and 25 (Fe sites). At the equilibrium potential of 0 V, all the elementary reaction steps on Co sites and Fe sites are exothermic, implying the ORR on the FeN₅-CoN₄, CoN₄, and FeN₅ is a spontaneous exothermal process.

For Co sites (Supplementary Fig. 24), the rate-determining step (RDS) for the CoN₄ model is the desorption of adsorbed *OH with the largest Gibbs free energy change ΔG₄ of 0.42 eV at U = 1.23 V. In contrast, the corresponding ΔG₄ for FeN₅-CoN₄ model is much small (0.25 eV). The RDS for the FeN₅-CoN₄ model is the adsorption of the *OOH from the first electron transfer step with a ΔG₁ of

0.29 eV. The limiting reaction energy barrier of ORR on FeN₅-CoN₄ (0.29 eV) is thus much lower than that on the CoN₄ model (0.42 eV). These results indicate the introduction of Fe sites promotes the desorption of *OH on the Co site, optimizing the ORR pathway.

For Fe sites (Supplementary Fig. 26, the RDS for both the FeN₅-CoN₄ model and FeN₅ model is the adsorption of the *OOH at U = 1.23 V. The RDS energy barrier on the FeN₅-CoN₄ model is 0.83 eV, also lower than that on the FeN₅ model (0.92 eV). These results suggest that the introduction of the Co site reduces the *OOH formation energy barrier in RDS on Fe sites.

These DFT calculation results clearly support that the combination of FeN₅ and CoN₄ moieties in FeN₅-CoN₄ generates a synergistic effect therebetween, justifying the enhanced ORR performance for FeCo-NCH compared with single-metal Fe-NCH or Co-NCH catalysts.

The following brief discussion and calculation details have been supplemented in the revised main text and **Methods** section, respectively (highlighted in red, page 9, first paragraph). The corresponding Supplementary Figures and detailed discussion have been provided in the Supplementary Information.

“The synergistic effect on binary Fe/Co-N_x sites for ORR was also evaluated by the density functional theory (DFT) calculations. Three models (i.e. FeN₅, CoN₄, and FeN₅-CoN₄) were constructed based on the EXAFS results (Supplementary Fig. 22). The downhill free energy pathways at U = 0 V on both Co and Fe sites in three models indicate the ORR is spontaneous exothermic (Supplementary Fig. 23-26).⁵⁰ The calculation results indicate that compared with Co or Fe-only site, the introduction of the Fe site promotes the desorption of *OH on the Co site (Supplementary Fig. 23, 24) while the introduction of the Co site significantly reduces the *OOH formation energy barrier on the Fe site (Supplementary Fig. 25, 26). In either case, the ORR pathway is promoted, suggesting the synergistic effect for FeN₅ and CoN₄ moieties and justifying the enhanced ORR activity for FeCo-NCH compared with single-metal Fe-NCH or Co-NCH. The detailed discussion can be found in Supplementary Information.^{21, 40} These results confirm that the constructing neighboring FeN₅ and CoN₄ moieties can generate a synergistic effect therebetween and contributes to enhanced ORR performance.”

In the **Methods** section: (highlighted in red, pages 12-13)

“**Density functional theory (DFT) calculations.** All DFT calculations were performed by using Vienna ab initio Simulation Package (VASP).⁶² The generalized gradient approximation (GGA) with the Perdew-Burke-Ernzerh (PBE)⁶³ of the exchange correlation functional within the projector augmented wave method⁶⁴ was utilized to model the electron-ion interaction. An energy cutoff of 450 eV for the plane-wave basis set was used. The convergence threshold was set to 10⁻⁵ eV in energy and 0.02 eV/Å in force, respectively. To prevent the interaction between two neighboring images, the vacuum layer thickness was set to 15 Å. A semi-empirical van der Waals (vdW) correction proposed by Grimme (DFT-D3)⁶⁵ was included to account for the dispersion interactions. A 2×2×1 Monkhorst-Pack grid of k-points was used to sample the first Brillouin zones of the surfaces for structural optimizations. All atoms are fully relaxed during structural optimizations.

The free energy of each elementary step in the proton-coupled electron transfer (PECT) reactions was computed using the computational hydrogen electrode (CHE) model for oxygen reduction reaction (ORR). Considering the O₂ molecule was not broken before reduction, the associative 4e- reduction

pathway was evaluated to be most feasible for ORR in this work, as follows:

where * represents the active site on the corresponding surface.

The free energy of the reactants and each intermediate state at an applied electrode potential U were computed by the equations, $\mu = E + E_{ZPE} - TS$, $\Delta G = \Delta E + \Delta E_{ZPE} - T\Delta S$, then the ΔG^*_{*O} , ΔG^*_{*OH} and ΔG^*_{*OOH} could be calculated as follows:

$$\begin{aligned} \Delta G^*_{*O} &= \mu^*_{*O} - \mu^* - (\mu_{H_2O} - \mu_{H_2}) \\ \Delta G^*_{*OH} &= \mu^*_{*OH} - \mu^* - (\mu_{H_2O} - \frac{1}{2}\mu_{H_2}) \\ \Delta G^*_{*OOH} &= \mu^*_{*OOH} - \mu^* - (2\mu_{H_2O} - \frac{3}{2}\mu_{H_2}) \end{aligned}$$

The Gibbs free energy of step 1-4 reaction could be expressed as:

$$\begin{aligned} \Delta G_1 &= \Delta G^*_{*OOH} - 4.92 \text{ eV} + eU \\ \Delta G_2 &= \Delta G^*_{*O} - \Delta G^*_{*OOH} + eU \\ \Delta G_3 &= \Delta G^*_{*OH} - \Delta G^*_{*O} + eU \\ \Delta G_4 &= -\Delta G^*_{*OH} + eU \end{aligned}$$

The following Supplementary Figures and detailed discussion have been supplemented in the revised **Supplementary Information** (highlighted in red, pages 23-27).

Supplementary Figure 22. Illustrations of optimized atomic configurations of (a) FeN₅, (b) CoN₄, and (c) FeN₅-CoN₄ model. (C: gray, N: blue, Fe: green, Co: pink, O: red, H: white).

Supplementary Figure 23. (a-c) Configurations of corresponding adsorbed intermediates (*OOH, *O, and *OH) on CoN₄ and (d-f) FeN₅-CoN₄ models (C: gray, N: blue, Fe: green, Co: pink, O: red, H: white).

Supplementary Figure 24. ORR free energy diagrams for the Co site in CoN₄ (green line) and in FeN₅-CoN₄ models (orange line) at U = 0 V, 0.9 V, and 1.23 V.

For Co sites, all the elementary reaction steps on both FeN₅-CoN₄ and CoN₄ models present a consistent downhill tendency at the U = 0 V, implying a spontaneous exothermal process. Upon increasing the thermodynamic equilibrium potential to 0.9 V, the ORR process on the FeN₅-CoN₄ model is still spontaneous, while the CoN₄ possesses a slight endothermic process from the desorption of *OH in the fourth step, suggesting that the external force is needed to drive this process. At U = 1.23 V, the rate-determining step (RDS) in the CoN₄ model is the desorption of adsorbed *OH with the largest Gibbs free energy change ΔG_4 of 0.42 eV. In contrast, the corresponding ΔG_4 in the FeN₅-CoN₄ model is much small (0.25 eV). The RDS for the FeN₅-CoN₄ model is the adsorption of the *OOH from the first electron transfer step with a ΔG_1 of 0.29 eV. The limiting reaction energy barrier of ORR on FeN₅-CoN₄ (0.29 eV) is lower compared with the CoN₄ model (0.42 eV). These results indicate that the introduction of the Fe site promotes the desorption of *OH on the Co site and optimizes the ORR pathway.

Supplementary Figure 25. (a-c) Configurations of corresponding adsorbed intermediates ($*\text{OOH}$, $*\text{O}$, and $*\text{OH}$) on FeN_5 and (d-f) $\text{FeN}_5\text{-CoN}_4$ models (C: gray, N: blue, Fe: green, Co: pink, O: red, H: white).

Supplementary Figure 26. ORR free energy diagrams for the Fe site in FeN₅ (orange line) and FeN₅-CoN₄ models (blue line) at U=0 V, 0.9 V, and 1.23 V.

As for Fe sites, when U = 0 V, all the elementary reaction steps on FeN₅-CoN₄ and FeN₅ models present a consistent downhill tendency, implying a spontaneous exothermic process. Upon increasing the thermodynamic equilibrium potential to 0.9 V, both the FeN₅-CoN₄ and FeN₅ present the uphill free energy in the first electron transfer step ($O_2 + H_2O + e^- \rightarrow *OOH + OH^-$), which is the RDS for the two models. It is noted that the FeN₅-CoN₄ model gives a lower energy barrier of 0.49 eV than that of the FeN₅ model (0.59 eV). Upon increasing the potential to 1.23 V, the first electron transfer step remains to be the RDS. The limiting reaction energy barrier of ORR on the FeN₅-CoN₄ (0.83 eV) model is still lower than that on the FeN₅ (0.92 eV) model, implying the ORR on the Fe site in FeN₅-CoN₄ is energetically favorable to that in the FeN₅. These results suggest that the introduction of the Co site significantly reduces the *OOH formation energy barrier on the Fe site and optimizes the ORR pathway.

Comment 4:

The half-wave potential of FeCo-NCH was 0.89 V (vs. RHE) in alkaline media. This value is comparatively better than other control catalysts. However, this is far from being impressive. Indeed, many catalysts having half-wave potentials even higher than 0.9 V are reported, some of which are included in the Table in SI.

Response 4:

RDE measurements evaluate the intrinsic activity of a catalyst in ideal conditions, which may or may not be consistent with the device's performance. The device performance is determined by the intrinsic activity as well as the catalyst architecture in terms of accessible active sites and mass transport etc. Similar RDE performances may give very different MEA performances due to the differences in accessible active sites and mass transport properties in fuel cells, which are determined by the porosity structure, surface properties, catalyst architecture, etc. For example, Prof. Shui's group reported a PEMFC with a peak power density of 1.18 W cm⁻² using an ORR electrocatalyst with a half-wave potential of only 0.823 V (*Nat. Catal.* **2**, 259–268 (2019)). Herein, exploring the catalysts with suitable architecture for high-performance devices and understanding the underlying fundamental science from the mechanistic level to the device level will advance the field and the exploration of the catalysts for practical devices (*Science* **356**, 599-604 (2017), etc.).

As mentioned above, this manuscript reported an interfacial assembly strategy to achieve a high accessible site density of 7.6×10^{19} sites g⁻¹, enabling the AEMFC or ZAB to deliver a high peak power densities of 569.0 or 414.5 mW cm⁻². Although the ORR performance in the RDE test is not the best, it is still ranking among the top non-PGM catalysts. We noted that some of the reported catalysts showed a higher half-wave potential ($E_{1/2}$) than ours but the poor power densities of fuel cells (taking some catalysts listed in Table S1 as examples: Fe/N/C nanotubes, $E_{1/2}$: 0.93 V, AEMFC P_{max} : 485 mW cm⁻² (*ACS Catal.* **7**, 6485-6492 (2017)); Fe/Ce-NCNW, $E_{1/2}$: 0.915 V, AEMFC P_{max} : 500 mW cm⁻² (*ACS Catal.* **10**, 2452-2458 (2020), etc.).

Again, we think our manuscript presented an effective strategy for substantially boosting the

accessible site density via interfacial assembly synthesis and demonstrated its significance in promoting the device performance for fuel cells and Zn-air batteries. We believe that such a strategy and the findings are generally applicable and helpful for the design of a wide range of energy-related electrocatalysts, which would be appealing to the community and a broad range of readerships.

Comment 5:

TOFs of the catalysts should be quantified, and comparatively discussed with those of previous works.

Response 5:

Thank you for the constructive comment. The TOF value shows the intrinsic ORR activity of the M-N_x sites and is highly related to the catalyst performance. Since the accessible site density (ASD) of the catalysts has been accessed by the nitrite stripping experiments, we estimated the TOFs from the decrease in ORR kinetic current on nitrite poisoning divided by ASD according to the literature (Kucernak et al. In situ electrochemical quantification of active sites in Fe–N/C non-precious metal catalysts. Nat. Commun. 7, 13285 (2016)). This method has been widely used to determine ASD and TOF as can be found in Nat. Catal. 2, 259–268 (2019); Nat. Catal. 5, 311–323 (2022); J. Am. Chem. Soc. 144, 13487–13498 (2022). In this method, TOFs are commonly reported at 0.80 V. The TOF value of the FeCo-NCH catalyst is calculated to be 2.2 e⁻¹ s⁻¹ site⁻¹, which is higher than most of the previously reported ORR catalysts (**Supplementary Table 5**). It indicates a fast ORR kinetic on the binary single-atomic Fe/Co-N_x sites. The high TOF together with the high ASD contributes to the enhanced ORR performance for FeCo-NCH.

The following discussion has been added in the revision (highlighted in red, page 9):

“The TOF value of the FeCo-NCH is calculated to be 2.2 e⁻¹ s⁻¹ site⁻¹ at 0.80 V, which is higher than most of the previously reported ORR catalysts (Supplementary Table 5), indicating a fast ORR kinetic.⁵¹ The high TOF together with the high ASD contributes to the enhanced ORR performance for FeCo-NCH.”

The corresponding TOF calculation method has been supplied in the **Methods** section (highlighted in red, page 12):

“The turnover frequency (TOF) was calculated by the following equation:

$$\text{TOF [e}^{-1} \text{ s}^{-1} \text{ site}^{-1}] = n_{\text{strip}} \times j_k \text{ (mA cm}^{-2}\text{)} / Q_{\text{strip}} \text{ [C g}^{-1}\text{]} \times L_c \text{ (mg cm}^{-2}\text{)}$$

Where n_{strip} (= 5) is the number of electrons associated with the reduction of one nitrite per site; F is the Faraday constant (96485 C mol⁻¹); j_k is the kinetic current density, L_c is the catalyst loading (0.27 mg cm⁻²). M is the average atomic mass of Fe and Co for FeCo-NCH or FeCo-NC (based on the ratio of Fe and Co); W is the metal contents of FeCo-NCH or FeCo-NC.”

The following Table S5 has been supplemented in the revised **Supplementary Information** (highlighted in red, page 45)

“Supplementary Table 5. Comparison of the site density (SD) and TOF performances for our electrocatalysts and reported PGM-free cathode electrocatalysts.”

Catalysts	SD (site g ⁻¹)	TOF (e ⁻¹ s ⁻¹ site ⁻¹)	Ref.
FeCo-NCH	7.6×10 ¹⁹	2.2 @ 0.80 V	This work
FeCo-NC	2.1×10 ¹⁹	1.8 @ 0.80 V	This work
TPI@Z8(SiO ₂)-650-C	~	1.63 @ 0.80 V	48
Fe-N/C	7.2×10 ¹⁸	1.6 @ 0.80 V	49
Fe-NC ^{A-DCDA}	4.69×10 ¹⁹	0.13 @ 0.85 V	50
*Fe _{0.5} NC-800	3.99×10 ¹⁹	0.46 @ 0.9 V	51
PAJ	~2×10 ¹⁹	~0.7 @ 0.80 V	52
*PANI-Fe	~1×10 ²⁰	0.23 @ 0.90 V	53
CNRS	~6×10 ¹⁹	~0.2 @ 0.80 V	52
FeNC-CVD-750	1.92×10 ²⁰	0.8 @ 0.80 V	54

The * marks represent SD and TOF were calculated based on CO sorption and desorption while the others are based on the ASD measured by the nitrite stripping experiments.

Responses to the Reviewer #2's comments

Recently, many such single atom Fe-N-C catalysts have been reported for the oxygen reduction reaction in the alkaline media for fuel cells or metal-air batteries. One of the most impressive works is in Nature Energy: <https://www.nature.com/articles/s41560-021-00878-7>.

Compared to the above work, the manuscript reports binary atomic metal – N_x sites containing Fe and Co with enhanced catalytic activity. However, based on current characterization and spectroscopy, it is unclear what are the interactions between Fe and Co atoms in the catalysts. Therefore, the possible promotional mechanisms remain unknown yet. Furthermore, the reporting performance in anion exchange membrane fuel cells is not compelling yet, especially compared to the work reported in Nature Energy by Adabi et al. Also, the synthesis methods to combine two types of single atoms sites in catalysts derived from ZIF-8 and ZIF-67 do not demonstrate sufficient innovation and creativity. Overall, this is routine work to develop a single atom catalyst for fuel cells and metal-air batteries, which cannot justify its qualification to be published in the journal.

Response:

We appreciate your valuable time and comments, which have helped us to improve the quality of our manuscript.

As you mentioned, many single atomic Fe-N-C catalysts were reported in recent years. However, The overwhelming majority of these reports are devoted to improving the ORR intrinsic activity and stability. Few research focused on understanding how the accessible site density affects the device performance and how to improve it by modulating catalyst structures. This work is trying to provide new insights into these scientific questions.

The device performance is determined not only by the intrinsic activity but also by the catalyst architecture in terms of accessible active sites and mass transport etc. as well as the MEA design. Adabi et al. work focused on the optimization of the porosity and hydrophilicity of commercial Fe-N-C catalysts to significantly improve the transport of liquid water in the catalyst layer. They used SiO₂ as hard templates to increase the average pore size and surface area of the Fe-N-C material and increase its graphitization to decrease the surface hydrophilicity for better water management. Together with the state-of-the-art AEMFC electrode design and a PtRu/C anode with a 0.6 mg cm⁻² PtRu loading, they achieved an impressively high power density in Nature Energy paper. Although Adabi's Fe-N-C material showed a half-wave potential of 0.846 V (slightly lower than ours, 0.889V), their fuel cells achieved extraordinary device performance, demonstrating the significance of catalyst porosity and surface properties as well as the electrode design. Adabi's work is very helpful for the further optimization of our catalysts and has been supplemented in the introduction part of the revision as reference 10, the AEMFC performance is also added in Table S4 for comparison (Ref. 25 in the **Supplementary Information**).

In the Reference section:

"10. Adabi, H., et al. High-performing commercial Fe-N-C cathode electrocatalyst for anion-exchange membrane fuel cells. *Nat. Energy* **6**, 834-843 (2021)."

And in **Supplementary Information**

“Supplementary Table 4. Comparison of the RDE and AEMFC performances for our electrocatalysts and various reported PGM-free cathode electrocatalysts (H₂/O₂ and H₂/Air).”

Catalysts	Ionomer	Membrane	$E_{1/2}$ (V vs. RHE)	P_{max} (H ₂ -O ₂) (mW cm ⁻²)	P_{max} (H ₂ -Air) (mW cm ⁻²)	Operating conditions	Ref.
Fe-N-C	ETFE-RG	HDPE-AEM	0.84	2050	~1000	80 °C, 200 kPa	25

Our work focuses on understanding how the accessible site density affects the catalyst and device performance and developing facile strategies to improve it. By presenting an interfacial assembly approach to boost accessible site density from 2.1×10^{19} to 7.6×10^{19} sites g⁻¹ and site utilization from 2.5% to 9.3% and comparing with a series of control catalysts, we reported here an effective and facile strategy (one-step pyrolysis with no need of hard template and template removal) for substantially enhancing the accessible site density and demonstrated its significance in promoting the device performance for fuel cells and Zn-air batteries.

With coordination-assisted adsorption, we effectively combine two types of single-atom sites in the catalyst. Such a combination holds two merits: 1) **significantly increasing metal loading**. The metal loadings in the state-of-the-art ZIF-based SACs are still relatively low (for instance: ~ 3 wt% Fe loading). In our case, we prepared the two control catalysts with similar morphology but one type of metal. Under the condition without metal clusters, the Fe loading in Fe-NCH SAC can only reach ~2.7 wt% and the Co loading in Co-NCH SAC can only reach ~5.2% wt%. However, bimetallic FeCo-NCH SAC gives a total metal loading of 7.9 wt% (2.4 wt% for Fe and 5.5 wt% for Co). 2) **enhancing intrinsic activity**. As explained and discussed below, the electronic interaction between Fe and Co lowers the energy barrier of the rate-limiting step in ORR on either the Fe or Co site, thus improving its intrinsic activity.

As for the device performance, without the optimization of the catalyst porosity and surface properties as well as the anode and MEA assembling (Again, these are not the focusing points of this manuscript), the fuel cell performance with our catalysts as cathode and a commercial PtRu catalyst in a relatively noble metal loading of 0.2 mg cm⁻² still ranks among the top (Supplementary Table 4). The zinc-air battery performance with our catalysts ranks among the top catalysts (Supplementary Table 5). Compared with control devices, these performances have well supported the significance of boosting accessible site density and site utilization.

In brief, we presented an effective and facile strategy for boosting accessible site density and site utilization and demonstrated its importance in enhancing device performance. We believe that the findings are generally applicable and helpful for the design of a wide range of energy-related electrocatalysts, which would be appealing to the community and a broad range of readerships.

To unravel the interactions between Fe and Co atoms in the catalysts and their influence on catalytic

performance, we have performed the density functional theory (DFT) calculations. On the basis of the characterization results of XPS and EXAFS, we first constructed three catalyst models: FeN₅, CoN₄, and FeN₅-CoN₄ model, respectively (Supplementary Fig. 22). The ORR process on the Co sites and Fe sites were discussed respectively. The adsorption models of various intermediate states on three models with different sites during the ORR process were optimized as shown in Supplementary Fig. 23 (Co sites) and 25 (Fe sites). At the equilibrium potential of 0 V, all the elementary reaction steps on Co sites and Fe sites are exothermic, implying the ORR on the FeN₅-CoN₄, CoN₄, and FeN₅ is a spontaneous exothermal process.

For Co sites (Supplementary Fig. 24), the rate-determining step (RDS) for the CoN₄ model is the desorption of adsorbed *OH with the largest Gibbs free energy change ΔG_4 of 0.42 eV at $U = 1.23$ V. In contrast, the corresponding ΔG_4 for FeN₅-CoN₄ model is much small (0.25 eV). The RDS for the FeN₅-CoN₄ model is the adsorption of the *OOH from the first electron transfer step with a ΔG_1 of 0.29 eV. The limiting reaction energy barrier of ORR on FeN₅-CoN₄ (0.29 eV) is thus much lower than that on the CoN₄ model (0.42 eV). These results indicate the introduction of Fe sites promotes the desorption of *OH on the Co site, optimizing the ORR pathway.

For Fe sites (Supplementary Fig. 26), the RDS for both the FeN₅-CoN₄ model and FeN₅ model is the adsorption of the *OOH at $U = 1.23$ V. The RDS energy barrier on the FeN₅-CoN₄ model is 0.83 eV, also lower than that on the FeN₅ model (0.92 eV). These results suggest that the introduction of the Co site reduces the *OOH formation energy barrier in RDS on Fe sites.

These DFT calculation results clearly support that the combination of FeN₅ and CoN₄ moieties in FeN₅-CoN₄ generates a synergistic effect therebetween, justifying the enhanced ORR performance for FeCo-NCH compared with single-metal Fe-NCH or Co-NCH catalysts.

The following brief discussion and calculation details have been supplemented in the revised main text and **Methods** section, respectively (highlighted in red, page 9, first paragraph). The corresponding Supplementary Figures and detailed discussion have been provided in the Supplementary Information.

“The synergistic effect on binary Fe/Co-N_x sites for ORR was also evaluated by the density functional theory (DFT) calculations. Three models (i.e. FeN₅, CoN₄, and FeN₅-CoN₄) were constructed based on the EXAFS results (Supplementary Fig. 22). The downhill free energy pathways at $U = 0$ V on both Co and Fe sites in three models indicate the ORR is spontaneous exothermal (Supplementary Fig. 23-26).⁵⁰ The calculation results indicate that compared with Co or Fe-only site, the introduction of the Fe site promotes the desorption of *OH on the Co site (Supplementary Fig. 23, 24) while the introduction of the Co site significantly reduces the *OOH formation energy barrier on the Fe site (Supplementary Fig. 25, 26). In either case, the ORR pathway is promoted, suggesting the synergistic effect for FeN₅ and CoN₄ moieties and justifying the enhanced ORR activity for FeCo-NCH compared with single-metal Fe-NCH or Co-NCH. The detailed discussion can be found in Supplementary Information.^{21, 40} These results confirm that the constructing neighboring FeN₅ and CoN₄ moieties can generate a synergistic effect therebetween and contributes to enhanced ORR performance.”

In the **Methods** section: (highlighted in red, pages 12-13)

“Density functional theory (DFT) calculations. All DFT calculations were performed by using

Vienna ab initio Simulation Package (VASP).⁶² The generalized gradient approximation (GGA) with the Perdew-Burke-Ernzerh (PBE)⁶³ of the exchange correlation functional within the projector augmented wave method⁶⁴ was utilized to model the electron-ion interaction. An energy cutoff of 450 eV for the plane-wave basis set was used. The convergence threshold was set to 10^{-5} eV in energy and 0.02 eV/Å in force, respectively. To prevent the interaction between two neighboring images, the vacuum layer thickness was set to 15 Å. A semi-empirical van der Waals (vdW) correction proposed by Grimme (DFT-D3)⁶⁵ was included to account for the dispersion interactions. A $2 \times 2 \times 1$ Monkhorst-Pack grid of k-points was used to sample the first Brillouin zones of the surfaces for structural optimizations. All atoms are fully relaxed during structural optimizations.

The free energy of each elementary step in the proton-coupled electron transfer (PECT) reactions was computed using the computational hydrogen electrode (CHE) model for oxygen reduction reaction (ORR). Considering the O_2 molecule was not broken before reduction, the associative 4e- reduction pathway was evaluated to be most feasible for ORR in this work, as follows:

where \ast represents the active site on the corresponding surface.

The free energy of the reactants and each intermediate state at an applied electrode potential U were computed by the equations, $\mu = E + E_{ZPE} - TS$, $\Delta G = \Delta E + \Delta E_{ZPE} - T\Delta S$, then the $\Delta G_{\ast O}$, $\Delta G_{\ast OH}$ and $\Delta G_{\ast OOH}$ could be calculated as follows:

$$\Delta G_{\ast O} = \mu_{\ast O} - \mu_{\ast} - (\mu_{H_2O} - \mu_{H_2})$$

$$\Delta G_{\ast OH} = \mu_{\ast OH} - \mu_{\ast} - (\mu_{H_2O} - \frac{1}{2}\mu_{H_2})$$

$$\Delta G_{\ast OOH} = \mu_{\ast OOH} - \mu_{\ast} - (2\mu_{H_2O} - \frac{3}{2}\mu_{H_2})$$

The Gibbs free energy of step 1-4 reaction could be expressed as:

$$\Delta G_1 = \Delta G_{\ast OOH} - 4.92 \text{ eV} + eU$$

$$\Delta G_2 = \Delta G_{\ast O} - \Delta G_{\ast OOH} + eU$$

$$\Delta G_3 = \Delta G_{\ast OH} - \Delta G_{\ast O} + eU$$

$$\Delta G_4 = -\Delta G_{\ast OH} + eU$$

The following Supplementary Figures and detailed discussion have been supplemented in the revised **Supplementary Information** (highlighted in red, pages 23-27).

Supplementary Figure 22. Illustrations of optimized atomic configurations of (a) FeN₅, (b) CoN₄, and (c) FeN₅-CoN₄ model. (C: gray, N: blue, Fe: green, Co: pink, O: red, H: white).

Supplementary Figure 23. (a-c) Configurations of corresponding adsorbed intermediates (*OOH, *O, and *OH) on CoN₄ and (d-f) FeN₅-CoN₄ models (C: gray, N: blue, Fe: green, Co: pink, O: red, H: white).

Supplementary Figure 24. ORR free energy diagrams for the Co site in CoN_4 (green line) and in FeN_5-CoN_4 models (orange line) at $U = 0$ V, 0.9 V, and 1.23 V.

For Co sites, all the elementary reaction steps on both FeN_5-CoN_4 and CoN_4 models present a consistent downhill tendency at the $U = 0$ V, implying a spontaneous exothermal process. Upon increasing the thermodynamic equilibrium potential to 0.9 V, the ORR process on the FeN_5-CoN_4 model is still spontaneous, while the CoN_4 possesses a slight endothermic process from the desorption of $*OH$ in the fourth step, suggesting that the external force is needed to drive this process. At $U = 1.23$ V, the rate-determining step (RDS) in the CoN_4 model is the desorption of adsorbed $*OH$ with the largest Gibbs free energy change ΔG_4 of 0.42 eV. In contrast, the corresponding ΔG_4 in the FeN_5-CoN_4 model is much small (0.25 eV). The RDS for the FeN_5-CoN_4 model is the adsorption of the $*OOH$ from the first electron transfer step with a ΔG_1 of 0.29 eV. The limiting reaction energy barrier of ORR on FeN_5-CoN_4 (0.29 eV) is lower compared with the CoN_4 model (0.42 eV). These results indicate that the introduction of the Fe site promotes the desorption of $*OH$ on the Co site and optimizes the ORR pathway.

Supplementary Figure 25. (a-c) Configurations of corresponding adsorbed intermediates ($*\text{OOH}$, $*\text{O}$, and $*\text{OH}$) on FeN_5 and (d-f) $\text{FeN}_5\text{-CoN}_4$ models (C: gray, N: blue, Fe: green, Co: pink, O: red, H: white).

Supplementary Figure 26. ORR free energy diagrams for the Fe site in FeN₅ (orange line) and FeN₅-CoN₄ models (blue line) at U=0 V, 0.9 V, and 1.23 V.

As for Fe sites, when U = 0 V, all the elementary reaction steps on FeN₅-CoN₄ and FeN₅ models present a consistent downhill tendency, implying a spontaneous exothermal process. Upon increasing the thermodynamic equilibrium potential to 0.9 V, both the FeN₅-CoN₄ and FeN₅ present the uphill free energy in the first electron transfer step ($O_2 + H_2O + e^- \rightarrow *OOH + OH^-$), which is the RDS for the two models. It is noted that the FeN₅-CoN₄ model gives a lower energy barrier of 0.49 eV than that of the FeN₅ model (0.59 eV). Upon increasing the potential to 1.23 V, the first electron transfer step remains to be the RDS. The limiting reaction energy barrier of ORR on the FeN₅-CoN₄ (0.83 eV) model is still lower than that on the FeN₅ (0.92 eV) model, implying the ORR on the Fe site in FeN₅-CoN₄ is energetically favorable to that in the FeN₅. These results suggest that the introduction of the Co site significantly reduces the *OOH formation energy barrier on the Fe site and optimizes the ORR pathway.

Responses to the Reviewer 3's comments

In this manuscript, the authors developed an interfacial assembly strategy to boost the exposure of catalytically active sites for enhancing the device's performance. The high-density accessible atomically dispersed metal-nitrogen active sites for oxygen reduction reaction were achieved by introducing binary Fe/Co-N_x sites and confining them in the ultrathin porous shell structures. Compared with the solid counterparts, the electrochemically accessible active sites were augmented by 3.7 times, resulting in the 3.4 times increase in the peak power densities for fuel cells and 2.8 times increase for Zn-air batteries. These results are impressive and will inspire more attention on the design of catalyst structure for boosting the effective active sites to enhance the device performance. The manuscript was well-organized and presented comprehensive data to support the main conclusions. The findings may also shed light on the design of other catalysts with highly exposed active sites for various applications. Based on the above facts, the reviewer would recommend publishing this study in this journal after addressing the following comments.

We appreciate your positive assessment and the valuable comments, which have helped us to improve the quality of our manuscript.

Reproduced specific comments and responses:

Comment 1:

It is mentioned that the formation of the hollow structure was caused by the stress-induced strategy due to the difference in the pyrolysis conditions between PDA and ZIF. The author should provide detailed information about the pyrolysis process and the related data for further discussion of this phenomena.

Response 1:

Thank you for the constructive suggestion. We accordingly carried out a series of experiments to investigate this process. With the following data, we believe that the PDA coating plays a role like a soft template during pyrolysis. Without the PDA coating layer, the precursor (ZIF-8@ZIF-67@Fe as shown in Supplementary Fig. 1c,f) tends to decompose and shrinks into concave nanoparticles with a solid structure in a smaller size (Supplementary Fig. 5). The supplemented thermogravimetric analysis (newly added Supplementary Fig. 3) shows that the PDA starts to decompose at the very early stage of the pyrolysis, and formed a relatively rigid carbon shell at the initial stage. The decomposition of the inner ZIF component begins at a temperature of around 400 °C. Given the interaction between the partly carbonized PDA shell and the ZIF-8 phase, the outer relatively rigid carbon shell restrains the contraction of the inner ZIF component during the pyrolysis. As the decomposition of the ZIF component goes with the temperature rising, the stresses from the shell induce the accumulation of pyrolyzed species on the shell, thus resulting in the hollow structure after the pyrolysis.

Such formation process was validated by a series of TEM images taken from the various decomposing stages (newly added Supplementary Fig. 4).

Supplementary Figure 1. SEM and TEM images of (a,d) ZIF-8, (b,e) ZIF-8@ZIF-67, and (c,f) ZIF-8@ZIF-67@Fe.

Supplementary Figure 5. (a-b) SEM and (c-d) TEM images of the control sample FeCo-NC prepared without PDA coating.

The corresponding discussion has been supplemented in the revision and the detailed analyses were supplied in the **Supplementary Information**.

In the revision (highlighted in red, page 4, first paragraph).

“During the pyrolysis, the relatively rigid carbonized PDA shell (Supplementary Fig. 3) formed at the very early stage acted as a framework to restrain the contraction of the inner ZIF component and induced the accumulation of pyrolyzed species on the shell, resulting in the final hollow structure. The Fe, Co-containing ZIF-67 layer was accordingly transformed to the carbon shell with neighboring Fe-N and Co-N moieties. Such formation process was validated by a series of TEM images taken from the various decomposing stages (see details in Supplementary Fig. 4). Without the assistance of the PDA coating, the pyrolyzed product ended up with a solid nano-polyhedral morphology with a concaved surface (Supplementary Fig. 5).”

The following **Supplementary Figures 3 and 4** and the discussion have been added in the revised **Supplementary Information** (highlighted in red, page 5).

The PDA starts to decompose at the very early stage of pyrolysis and formed a relatively rigid carbon structure at the initial stage. The decomposition of the inner ZIF component begins at a temperature over 440 °C. Given the interaction between the partly carbonized PDA shell and the ZIF-8 phase, the outer relatively rigid carbon shell restrains the contraction of the inner ZIF component during the pyrolysis. As the decomposition of the ZIF component goes with the temperature rising, the stresses from the shell induce the accumulation of pyrolyzed species on the shell, thus resulting in the hollow structure after the pyrolysis.

Supplementary Figure 3. TGA plots for ZIF8, PDA, and ZIF-8@ZIF-67@Fe@PDA composites. The tests were conducted in an N₂ atmosphere with a heating rate of 10 °C min⁻¹.

Supplementary Figure 4. TEM images collected at various temperatures during the pyrolysis for the preparation of FeCo-NCH. (a) 300 °C, (b) 400 °C, (c) 450 °C, (d) 500 °C, (e) 550 °C, (f) 650 °C, (g) 750 °C, and (h) 800 °C.

The morphology evolution of the FeCo-NCH during the pyrolysis was investigated in detail. It is found the composite surface becomes rougher but the inner remains in a solid structure at the low temperature of 300 °C, consistent with the TGA results that PDA decomposes before the ZIF component. As the temperature goes up to 400 °C at which ZIF starts to decompose, a small hollow appears, which grows as the temperature rises to 450 and 500 °C. The hollow structure formed at the temperature of 550 °C. The dark spots can be attributed to Zn-containing species from ZIF-8. As the temperature keeps rising, the carbon shell becomes thinner and the Zn species evaporate from the product. The final carbonized hollow structure was maintained at 800 °C.

Comment 2:

According to the manuscript, the formation of the ZIF-PDA core-shell structure is important for the formation of mesoporous in the final hollow structured catalysts. More discussion is needed for the

role of PDA in the formation of such structure.

Response 2:

Thank you very much for the comment. In brief, the PDA coating serves as a soft templating during the pyrolysis. Without the PDA coating layer, the ZIF-8@ZIF-67@Fe precursor (Supplementary Fig. 1c,f) tends to decompose and shrink into concave nanoparticles with a smaller size (Supplementary Fig. 5). The corresponding discussion about the role of the PDA in the formation of the hollow structure have been integrated into the above discussion of the structure formation process in response to the comment 1.

Comment 3:

Why did author choose PDA as the coating layer? Are there other materials which can achieve the similar structure?

Response 3:

PDA is widely used as a precursor for fabricating a porous carbon layer with a controllable thickness. Messersmith et al. (*Science*, 318, 426-430 (2007)) found that dopamine (DA) could easily polymerize and form polydopamine (PDA) nanofilms on a broad range of solid surfaces when exposed to air under weakly alkaline conditions (pH 8.5). This property allows researchers to coat various substrates with the PDA layer in a well-controllable way. After the pyrolysis, a well-defined porous carbon layer can be obtained.

In our case, the PDA can be controllably coated on ZIF-based precursors via the strong interaction of the PDA's catechol structures and ZIFs. We believe that the PDA coating plays a role like the soft template to induce the formation of the hollow structure during pyrolysis. Without the PDA coating layer, the precursor (ZIF-8@ZIF-67@Fe as shown in Supplementary Fig. 1c,f) tends to decompose and shrinks into concave nanoparticles with a solid structure in a smaller size (Supplementary Fig. 5). The supplemented thermogravimetric analysis (Supplementary Fig. 3) shows that the PDA starts to decompose at the very early stage of the pyrolysis, and formed a relatively rigid carbon shell at the initial stage. The decomposition of the inner ZIF component begins at a temperature of around 400 °C. Given the interaction between the partly carbonized PDA shell and the ZIF-8 phase, the outer relatively rigid carbon shell restrains the contraction of the inner ZIF component during the pyrolysis. As the decomposition of the ZIF component goes with the temperature rising, the stresses from the shell induce the accumulation of pyrolyzed species on the shell, thus resulting in the hollow structure after the pyrolysis. In addition, the catechol structure is also a strong ligand of multivalent metal ions, which can help anchor metal atoms and prevent them from aggregation to some extent during the pyrolysis, contributing to achieving high loading of single atomic sites.

Although both PANI and polypyrrole can form nitrogen-doped carbon layers on the surface of

materials, they are not as effective as polydopamine in stabilizing metal atomic-level dispersion and the coating thickness cannot be well controlled. Both aniline and pyrrole monomers are self-polymerized by chemical oxidation polymerization. In this process, they follow the cationic polymerization mechanism with characteristics of fast initiation, slow growth, and difficult termination. The chain length of the polymer is easy to increase rapidly, and the end of the chain is difficult to terminate, so the final coating thickness is not easy to control. As for the polymerization of dopamine, it follows the free radical polymerization mechanism with the characteristics of slow initiation, fast growth, and quick termination. The initial polymerization is a slow process, the concentration of the dimer is low, the reaction is mild, and with the increase of chain length, the final chain end termination is relatively easy. Therefore, the reaction process of dopamine polymerization is more controllable. Considering this, we herein chose the PDA as the coating layer in our case.

Comment 4:

In this manuscript, the author mentioned that the Fe/Co-N_x binary atom catalyst synthesized by this strategy achieved a 7.9 wt% of metal loading. Considering that polydopamine also contains the N element, therefore, does the extra N species from the pyrolysis of polydopamine contribute to the increase in metal loading of the catalyst?

Response 4:

Thanks for the insightful thoughts. Yes, as mentioned above, the catechol structure in the PDA layer is a strong ligand of multivalent metal ions. In our synthesis process of FeCo-NCH, the Fe ions were introduced to the surface of ZIF-8@ZIF-67 by impregnation adsorption, we believe the catechol structure contained in the in-situ coated polydopamine layer could coordinate well with Fe and Co species in the Fe doped ZIF-67 layer. In the meanwhile, the amidogen groups in the PDA layer provide extra anchoring sites during the pyrolysis, contributing to the increase in metal loading of the FeCo-NCH.

Comment 5:

The author introduced both Fe and Co sources to achieve binary atomic Fe/Co-N_x sites. What are the merits for the binary metal-nitrogen structure? Can the author achieve the similar metal loading with single metal source?

Response 5:

Thank you for the valuable comment.

These two merits for using Fe and Co sources to achieve binary atomic Fe/Co-N_x sites. 1) **significantly increasing metal loading**. The metal loadings in the state-of-the-art ZIF-based SACs

is still relatively low (for instance: ~ 3 wt% Fe loading). In our case, we prepared the two control catalysts with similar morphology but one type of metal. Under the condition without metal clusters, the Fe loading in Fe-NCH SAC can only reach ~ 2.7 wt% and the Co loading in Co-NCH SAC can only reach ~ 5.2 wt%. However, bimetallic FeCo-NCH SAC gives a total metal loading of 7.9 wt% (2.4 wt% for Fe and 5.5 wt% for Co). 2) **enhancing intrinsic activity**. As explained and discussed below, the electronic interaction between Fe and Co lowers the energy barrier of the rate-limiting step in ORR on either the Fe or Co site, thus improving its intrinsic activity.

To unravel the interactions between Fe and Co atoms in the catalysts and their influence on catalytic performance, we have supplemented the density functional theory (DFT) calculations. On the basis of the characterization results of XPS and EXAFS, we first constructed three catalyst models: FeN₅, CoN₄, and FeN₅-CoN₄ model, respectively (Supplementary Fig. 22). The ORR process on the Co sites and Fe sites were discussed respectively. The adsorption models of various intermediate states on three models with different sites during the ORR process were optimized as shown in Supplementary Fig. 23 (Co sites) and 25 (Fe sites). At the equilibrium potential of 0 V, all the elementary reaction steps on Co sites and Fe sites are exothermic, implying the ORR on the FeN₅-CoN₄, CoN₄, and FeN₅ is a spontaneous exothermal process.

For Co sites (Supplementary Fig. 24), the rate-determining step (RDS) for the CoN₄ model is the desorption of adsorbed *OH with the largest Gibbs free energy change ΔG_4 of 0.42 eV at $U = 1.23$ V. In contrast, the corresponding ΔG_4 for FeN₅-CoN₄ model is much small (0.25 eV). The RDS for the FeN₅-CoN₄ model is the adsorption of the *OOH from the first electron transfer step with a ΔG_1 of 0.29 eV. The limiting reaction energy barrier of ORR on FeN₅-CoN₄ (0.29 eV) is thus much lower than that on the CoN₄ model (0.42 eV). These results indicate the introduction of Fe sites promotes the desorption of *OH on the Co site, optimizing the ORR pathway.

For Fe sites (Supplementary Fig. 26, the RDS for both the FeN₅-CoN₄ model and FeN₅ model is the adsorption of the *OOH at $U = 1.23$ V. The RDS energy barrier on the FeN₅-CoN₄ model is 0.83 eV, also lower than that on the FeN₅ model (0.92 eV). These results suggest that the introduction of the Co site reduces the *OOH formation energy barrier in RDS on Fe sites.

These DFT calculation results clearly support that the combination of FeN₅ and CoN₄ moieties in FeN₅-CoN₄ generates a synergistic effect therebetween, justifying the enhanced ORR performance for FeCo-NCH compared with single-metal Fe-NCH or Co-NCH catalysts.

In the meanwhile, as mentioned above, we cannot achieve a similar metal loading with a single metal source. In this work, we have optimized the content of Co and Fe as much as possible. For the Fe-only catalyst (Fe-NCH SAC), the Fe loading can only reach ~ 2.7 wt%. For the Co-only catalyst (Co-NCH SAC), the Co loading can only reach ~ 5.2 wt%. If we further increase the number of metal sources, the aggregated metal particles will inevitably appear in the catalyst, as shown in Figure R1.

Figure. R1. TEM images of the (a) Co-NCH and (b) Fe-NCH with ~8 wt % metal loading.

The following brief discussion and calculation details have been supplemented in the revised main text and **Methods** section, respectively (highlighted in red, page 9, first paragraph). The corresponding Supplementary Figures and detailed discussion have been provided in the Supplementary Information.

“The synergistic effect on binary Fe/Co-N_x sites for ORR was also evaluated by the density functional theory (DFT) calculations. Three models (i.e. FeN₅, CoN₄, and FeN₅-CoN₄) were constructed based on the EXAFS results (Supplementary Fig. 22). The downhill free energy pathways at $U = 0$ V on both Co and Fe sites in three models indicate the ORR is spontaneous exothermic (Supplementary Fig. 23-26).⁵⁰ The calculation results indicate that compared with Co or Fe-only site, the introduction of the Fe site promotes the desorption of *OH on the Co site (Supplementary Fig. 23, 24) while the introduction of the Co site significantly reduces the *OOH formation energy barrier on the Fe site (Supplementary Fig. 25, 26). In either case, the ORR pathway is promoted, suggesting the synergistic effect for FeN₅ and CoN₄ moieties and justifying the enhanced ORR activity for FeCo-NCH compared with single-metal Fe-NCH or Co-NCH. The detailed discussion can be found in Supplementary Information.^{21, 40} These results confirm that the constructing neighboring FeN₅ and CoN₄ moieties can generate a synergistic effect therebetween and contributes to enhanced ORR performance.”

In the **Methods** section: (highlighted in red, pages 12-13)

“**Density functional theory (DFT) calculations.** All DFT calculations were performed by using Vienna ab initio Simulation Package (VASP).⁶² The generalized gradient approximation (GGA) with the Perdew-Burke-Ernzerh (PBE)⁶³ of the exchange correlation functional within the projector augmented wave method⁶⁴ was utilized to model the electron-ion interaction. An energy cutoff of 450 eV for the plane-wave basis set was used. The convergence threshold was set to 10^{-5} eV in energy and 0.02 eV/Å in force, respectively. To prevent the interaction between two neighboring images, the vacuum layer thickness was set to 15 Å. A semi-empirical van der Waals (vdW) correction proposed by Grimme (DFT-D3)⁶⁵ was included to account for the dispersion interactions. A $2 \times 2 \times 1$ Monkhorst-

Pack grid of k-points was used to sample the first Brillouin zones of the surfaces for structural optimizations. All atoms are fully relaxed during structural optimizations.

The free energy of each elementary step in the proton-coupled electron transfer (PECT) reactions was computed using the computational hydrogen electrode (CHE) model for oxygen reduction reaction (ORR). Considering the O₂ molecule was not broken before reduction, the associative 4e⁻ reduction pathway was evaluated to be most feasible for ORR in this work, as follows:

where * represents the active site on the corresponding surface.

The free energy of the reactants and each intermediate state at an applied electrode potential U were computed by the equations, $\mu = E + E_{ZPE} - TS$, $\Delta G = \Delta E + \Delta E_{ZPE} - T\Delta S$, then the ΔG^*_{*O} , ΔG^*_{*OH} and ΔG^*_{*OOH} could be calculated as follows:

$$\underline{\Delta G^*_{*O} = \mu^*_{*O} - \mu^* - (\mu_{H_2O} - \mu_{H_2})}$$

$$\underline{\Delta G^*_{*OH} = \mu^*_{*OH} - \mu^* - (\mu_{H_2O} - \frac{1}{2}\mu_{H_2})}$$

$$\underline{\Delta G^*_{*OOH} = \mu^*_{*OOH} - \mu^* - (2\mu_{H_2O} - \frac{3}{2}\mu_{H_2})}$$

The Gibbs free energy of step 1-4 reaction could be expressed as:

$$\underline{\Delta G_1 = \Delta G^*_{*OOH} - 4.92 \text{ eV} + eU}$$

$$\underline{\Delta G_2 = \Delta G^*_{*O} - \Delta G^*_{*OOH} + eU}$$

$$\underline{\Delta G_3 = \Delta G^*_{*OH} - \Delta G^*_{*O} + eU}$$

$$\underline{\Delta G_4 = -\Delta G^*_{*OH} + eU}$$

The following Supplementary Figures and detailed discussion have been supplemented in the revised **Supplementary Information** (highlighted in red, pages 23-27).

Supplementary Figure 22. Illustrations of optimized atomic configurations of (a) FeN₅, (b) CoN₄, and (c) FeN₅-CoN₄ model. (C: gray, N: blue, Fe: green, Co: pink, O: red, H: white).

Supplementary Figure 23. (a-c) Configurations of corresponding adsorbed intermediates (*OOH, *O, and *OH) on CoN₄ and (d-f) FeN₅-CoN₄ models (C: gray, N: blue, Fe: green, Co: pink, O: red, H: white).

Supplementary Figure 24. ORR free energy diagrams for the Co site in CoN₄ (green line) and in FeN₅-CoN₄ models (orange line) at U = 0 V, 0.9 V, and 1.23 V.

For Co sites, all the elementary reaction steps on both FeN₅-CoN₄ and CoN₄ models present a consistent downhill tendency at the U = 0 V, implying a spontaneous exothermal process. Upon increasing the thermodynamic equilibrium potential to 0.9 V, the ORR process on the FeN₅-CoN₄ model is still spontaneous, while the CoN₄ possesses a slight endothermic process from the desorption of *OH in the fourth step, suggesting that the external force is needed to drive this process. At U = 1.23 V, the rate-determining step (RDS) in the CoN₄ model is the desorption of adsorbed *OH with the largest Gibbs free energy change ΔG₄ of 0.42 eV. In contrast, the corresponding ΔG₄ in the FeN₅-CoN₄ model is much small (0.25 eV). The RDS for the FeN₅-CoN₄ model is the adsorption of the *OOH from the first electron transfer step with a ΔG₁ of 0.29 eV. The limiting reaction energy barrier of ORR on FeN₅-CoN₄ (0.29 eV) is lower compared with the CoN₄ model (0.42 eV). These results indicate that the introduction of the Fe site promotes the desorption of *OH on the Co site and optimizes the ORR pathway.

Supplementary Figure 25. (a-c) Configurations of corresponding adsorbed intermediates ($*\text{OOH}$, $*\text{O}$, and $*\text{OH}$) on FeN_5 and (d-f) $\text{FeN}_5\text{-CoN}_4$ models (C: gray, N: blue, Fe: green, Co: pink, O: red, H: white).

Supplementary Figure 26. ORR free energy diagrams for the Fe site in FeN₅ (orange line) and FeN₅-CoN₄ models (blue line) at U=0 V, 0.9 V, and 1.23 V.

As for Fe sites, when U = 0 V, all the elementary reaction steps on FeN₅-CoN₄ and FeN₅ models present a consistent downhill tendency, implying a spontaneous exothermal process. Upon increasing the thermodynamic equilibrium potential to 0.9 V, both the FeN₅-CoN₄ and FeN₅ present the uphill free energy in the first electron transfer step ($O_2 + H_2O + e^- \rightarrow *OOH + OH^-$), which is the RDS for the two models. It is noted that the FeN₅-CoN₄ model gives a lower energy barrier of 0.49 eV than that of the FeN₅ model (0.59 eV). Upon increasing the potential to 1.23 V, the first electron transfer step remains to be the RDS. The limiting reaction energy barrier of ORR on the FeN₅-CoN₄ (0.83 eV) model is still lower than that on the FeN₅ (0.92 eV) model, implying the ORR on the Fe site in FeN₅-CoN₄ is energetically favorable to that in the FeN₅. These results suggest that the introduction of the Co site significantly reduces the *OOH formation energy barrier on the Fe site and optimizes the ORR pathway.

Comment 6:

The author mentioned that the number of metal-N_x sites on the surface of the catalyst can be assessed by in-situ electrochemical nitrite stripping. Can the author provide the relevant fundamental principle involved in the method?

Response 6:

Thank you for the comment. The electrochemical nitrite stripping was performed to determine the surface active site density of catalysts according to the previous report by Kucernak et al. (*Nat. Commun.* **7**, 13285 (2016)). This method has been widely used to determine ASD, such as *Nat. Catal.* **2**, 259–268 (2019); *Nat. Catal.* **5**, 311–323 (2022); *J. Am. Chem. Soc.* **144**, 13487–13498 (2022).

In the presence of nitrite, the metal-nitrogen sites are poisoned due to nitrite coordination according to the equation:

Cyclic voltammetry is then performed at specific potential intervals. Due to the stripping of nitrite, the CV curve will show an increased apparent current, that is, the stripping peak. The number of active sites is then equal to the number of stripped molecules. The number of nitrite stripping can be calculated by the stripping charge (Q_{strip}), to know the number of active sites.

The corresponding description and discussion have been supplemented in the **Methods** section (highlighted in red, page 12)

“The site density (SD) was obtained according to the method presented by Kucernak et al.¹⁷ Briefly, extensive electrochemical cycling was performed on the catalyst alternatively in O₂ and N₂ in acetate

buffer (pH 5.2) to give consistent cyclic voltammetry curves in N₂. The catalyst was then poisoned by NaNO₂. The ORR performance was recorded before, during, and after the nitrite adsorption. Nitrite stripping was conducted in the potential region of 0.35 to -0.35 V vs. RHE. The excess in cathodic charge (Q_{strip}) is proportional to the SD.

$$\text{SD [mol g}^{-1}\text{]} = Q_{strip} [\text{C g}^{-1}] / n_{strip} F [\text{C mol}^{-1}]$$
$$\text{Utilization of sites (\%)} = \text{SD [mol g}^{-1}\text{]} \times M [\text{g mol}^{-1}] / W$$

Comment 7:

The detailed preparation process of the GDL for the gas diffusion layer (GDE) and the membrane-electrode-assembly (MEA) for the fuel cell should be provided in the manuscript.

Response 7:

According to your suggestion, the corresponding descriptions have been added in the revised **Methods** (highlighted in red, page 12-13).

For the preparation of GDE:

“The catalyst ink was prepared by mixing 1 mg catalysts with 40 μL 0.5% Nafion in 800 μL ethanol, followed by a 30 min ultrasonic treatment. It was then drop-casting onto a Teflon-coated carbon fiber paper with a catalyst loading of 1.0 mg cm^{-2} as the GDE. The achieved GDE was dried in a vacuum oven at 60 $^{\circ}\text{C}$ overnight.”

For the preparation of MEA:

“The PtRu/C (40 wt% Pt and 20 wt% Ru on Vulcan XC-72R, HiSpec 10000) were used as the anode catalyst. FeCo-NCH was used as the cathode catalyst with a loading of 1 mg cm^{-2} . The catalysts (with 1 mg Vulcan XC-72R per 10mg catalyst) and ionomer (PAP-TP-100, 5 wt% in ethanol for performance test, the dry weight of the ionomer accounted for 45% of the catalyst mass) were dispersed in the mixture of water and isopropanol (1:20 v/v) via sonication for 2 h in an ice water bath. The catalyst ink was then sprayed on the PAP-TP-85 membrane (Cl^{-} form, $20 \pm 5 \mu\text{m}$ in the dry state), forming a catalyst-coated membrane (CCM) with an electrode area of 5 cm^2 . The metal loading of PtRu in the anode was 0.2 mg/cm^2 . Next, the prepared CCM was soaked in 3 M KOH for 24 h at 60 $^{\circ}\text{C}$ to exchange Cl^{-} with OH^{-} , and washed with distilled water before fuel cell tests to remove the excess KOH. The resulting CCM was positioned between two pieces of Teflon-treated carbon paper (SIGRACET, 29BC) to make the membrane electrode assembly (MEA).”

Comment 8:

Some typos and spelling errors should be corrected in this manuscript, for example, the extra “32” mark in the Y-axis of Figure 3b; “0.5 μL of 0.5 Nafion solution” in the Methods section, etc.

Response 8:

Thank you very much for the careful reading. The relevant typos have been corrected.

The ordinate of Figure 3b has been corrected:

In the revised main text: (highlighted in red, page 5)

Fig 3. (b) Fourier-transform (FT) EXAFS spectra of FeCo-NCH and reference samples.

Comment 9:

Some of the recent progress in the field should be supplemented.

Response 9:

We have supplemented some of the recent progress in the field as references in the revision. For example, the recent advance of the bimetallic single-atomic structure for ORR catalysts.

“To further improve the intrinsic activity, binary dual-atomic M–N_x sites have been recently developed. They not only inherit the advantages of SACs, such as maximal atomic utilization, well-defined and adjustable coordination configuration but also offer the possibility to improve catalytic performance by harnessing synergistic effect.^{15, 20} The interaction between multiple active centers allows to regulate the electronic structure and geometric configuration, enabling to optimize the adsorption and desorption of intermediates to approach the apex of the activity volcano plot.^{18, 21}”

The corresponding references have been updated and highlighted in the **References** section:

15. Hossen M. M. *et al.* State-of-the-art and developmental trends in platinum group metal-free cathode catalyst for anion exchange membrane fuel cell (AEMFC). *Appl. Catal., B*, 121733 (2022).

18. Xiao, M., *et al.* Climbing the apex of the ORR volcano plot via binuclear site construction: Electronic and geometric engineering. *J. Am. Chem. Soc.* **141**, 17763-17770 (2019).

20. Li L., Yuan K. & Chen Y. Breaking the scaling relationship limit: From single-atom to dual-atom catalysts. *Acc. Mater. Res.* **3**, 584-596 (2022).

21. Zhang, W., *et al.* Emerging dual-atomic-site catalysts for efficient energy catalysis. *Adv. Mater.* **33**, 2102576 (2021).

Responses to the Reviewer 4's comments

Developing ORR catalysts with high intrinsic sites and dense accessible site density is critical for boosting the performance of practical energy devices but remains a great challenge. This manuscript reported a stress-induced interfacial assembly strategy for synthesizing a dual atomic catalyst with a highly accessible site density of 7.6×10^{19} sites g^{-1} . The 3-fold improvement of the accessible site density for as-obtained FeCo-NCH catalyst is very impressive. Detailed physical characterizations and electrochemical measurements were conducted and the synthesis process was fully elaborated. The practical devices employing FeCo-NCH cathode exhibited remarkably improved power densities of 569 mW cm^{-2} for AEMFC and 414.5 mW cm^{-2} for ZAB compared with the control ones, which are also among the state-of-the-art non-PGM ORR catalysts. These results prove the feasibility of the interfacial assembly strategy to improve the performance of SACs in practical energy conversion devices.

Based on these facts, this reviewer would like to recommend its publication in Nat. Commun. after addressing the following comments.

We appreciate your positive assessment and the valuable comments, which have helped us to improve the quality of our manuscript.

Reproduced specific comments and responses:

Comment 1:

The author proposed a stress-induced interfacial assembly strategy to boost the accessible site density, the stress-induced effect should be explained in the manuscript.

Response 1:

Thank you for the constructive suggestion. We accordingly carried out a series of experiments to investigate this process. With the following data, we believe that the PDA coating plays a role like a soft template during pyrolysis. Without the PDA coating layer, the precursor (ZIF-8@ZIF-67@Fe as shown in Supplementary Fig. 1c,f) tends to decompose and shrinks into concave nanoparticles with a solid structure in a smaller size (Supplementary Fig. 5). The supplemented thermogravimetric analysis (newly added Supplementary Fig. 3) shows that the PDA starts to decompose at the very early stage of the pyrolysis, and formed a relatively rigid carbon shell at the initial stage. The decomposition of the inner ZIF component begins at a temperature of around $400 \text{ }^\circ\text{C}$. Given the interaction between the partly carbonized PDA shell and the ZIF-8 phase, the outer relatively rigid carbon shell restrains the contraction of the inner ZIF component during the pyrolysis. As the decomposition of the ZIF component goes with the temperature rising, the stresses from the shell induce the accumulation of pyrolyzed species on the shell, thus resulting in the hollow structure after the pyrolysis.

Such formation process was validated by a series of TEM images taken from the various decomposing stages (newly added Supplementary Fig. 4).

Supplementary Figure 1. SEM and TEM images of (a,d) ZIF-8, (b,e) ZIF-8@ZIF-67, and (c,f) ZIF-8@ZIF-67@Fe.

Supplementary Figure 5. (a-b) SEM and (c-d) TEM images of the control sample FeCo-NC prepared without PDA coating.

The corresponding discussion has been supplemented in the revision and the detailed analyses were supplied in the **Supplementary Information**.

In the revision (highlighted in red, page 4, first paragraph).

“During the pyrolysis, the relatively rigid carbonized PDA shell (Supplementary Fig. 3) formed at the very early stage acted as a framework to restrain the contraction of the inner ZIF component and induced the accumulation of pyrolyzed species on the shell, resulting in the final hollow structure. The Fe, Co-containing ZIF-67 layer was accordingly transformed to the carbon shell with neighboring Fe-N and Co-N moieties. Such formation process was validated by a series of TEM images taken from the various decomposing stages (see details in Supplementary Fig. 4). Without the assistance of the PDA coating, the pyrolyzed product ended up with a solid nano-polyhedral morphology with a concaved surface (Supplementary Fig. 5).”

The following **Supplementary Figures 3 and 4** and the discussion have been added in the revised **Supplementary Information** (highlighted in red, page 5).

The PDA starts to decompose at the very early stage of pyrolysis and formed a relatively rigid carbon structure at the initial stage. The decomposition of the inner ZIF component begins at a temperature over 440 °C. Given the interaction between the partly carbonized PDA shell and the ZIF-8 phase, the outer relatively rigid carbon shell restrains the contraction of the inner ZIF component during the pyrolysis. As the decomposition of the ZIF component goes with the temperature rising, the stresses from the shell induce the accumulation of pyrolyzed species on the shell, thus resulting in the hollow structure after the pyrolysis.

Supplementary Figure 3. TGA plots for ZIF8, PDA, and ZIF-8@ZIF-67@Fe@PDA composites. The tests were conducted in an N₂ atmosphere with a heating rate of 10 °C min⁻¹.

Supplementary Figure 4. TEM images collected at various temperatures during the pyrolysis for the preparation of FeCo-NCH. (a) 300 °C, (b) 400 °C, (c) 450 °C, (d) 500 °C, (e) 550 °C, (f) 650 °C, (g) 750 °C, and (h) 800 °C.

The morphology evolution of the FeCo-NCH during the pyrolysis was investigated in detail. It is found the composite surface becomes rougher but the inner remains in a solid structure at the low temperature of 300 °C, consistent with the TGA results that PDA decomposes before the ZIF component. As the temperature goes up to 400 °C at which ZIF starts to decompose, a small hollow appears, which grows as the temperature rises to 450 and 500 °C. The hollow structure formed at the temperature of 550 °C. The dark spots can be attributed to Zn-containing species from ZIF-8. As the temperature keeps rising, the carbon shell becomes thinner and the Zn species evaporate from the product. The final carbonized hollow structure was maintained at 800 °C.

Comment 2:

To validate the advantages of constructing interfacial assembly active sites, the metal contents of the

single-metal-based control samples should be also included in Supplementary Table 3.

Response 2:

Thank you very much for the comment. As your suggested, we have added the single metal-based samples in the revised Supplementary Table 3 as shown below. The metal content obtained by the XPS and ICP is similar, validating the advantages of constructing interfacial assembly active sites, the related content has been updated in the revision.

In the revised main text: (highlighted in red, page 7)

“The hollow FeCo-NCH gives a much higher surface metal content than solid FeCo-NC (6.9 vs. 3.0 wt%) (Fig. 4a). The metal loadings for all catalysts were listed in Supplementary Table 3.”

In **Supplementary Information**: (highlighted in red, page 42)

“**Supplementary Table 3.** Metal contents in various catalysts, obtained by XPS and ICP.”

Samples		ICP (wt%)	XPS (wt%)
FeCo-NCH	Fe	2.4	2.1
	Co	5.5	4.6
FeCo-NC	Fe	2.1	0.6
	Co	5.0	2.4
Fe-NCH	Fe	2.7	2.7
Fe-NC		1.9	1.1
Co-NCH	Co	5.2	4.9
Co-NC		4.5	2.2

Comment 3:

Some catalysts listed in Supplementary Table 2 are not derived from MOF-based precursors. The authors should check this table again.

Response 3:

We carefully checked and updated Supplementary Table 2 as shown below. (highlighted in red, page 41)

“**Supplementary Table 2.** Metal content comparison of dual-site SACs according to the recently reported MOF-derived catalysts.”

Catalysts	Content of metal elements (wt%)	Ref.
-----------	---------------------------------	------

FeCo-NCH	Co: 5.5 Fe: 2.4	This work
(Fe,Co)/N-C	Co: 1.17 Fe: 0.93	1
FeCo-NPC	Fe: 0.33 Co: 0.14	2
Fe, Mn-N/C-900	Fe: 1.75 Mn: 0.07	3
FeCo-ISAc/NC	Co: 0.218 Fe: 0.964	4
CoFe@C	Co: 0.5 Fe: 0.37	5
Ni/Fe-N-C	Ni: 0.97 Fe: 0.34	6
Zn/CoN-C	Co: 0.37 Fe: 0.43	7
FeCoN ₃ C	Co: 1.12 Fe: 1.06	8
FeCo-IA/NC	Co: 1.06 Fe: 0.26	9
FeNi-N ₆ -C	Ni: 1.472 Fe: 1.448	10
Co ₁ -PNC/Ni ₁ -PNC	Co: 2.15 Ni: 0.68	11
Fe/Ni-N _x /OC	Fe: 1.37 Ni: 0.47	12
ZnCoNC	Zn: 0.35 Co: 0.72	13
Co ₂ /Fe-N@CHC	Co: 2.13 Fe: 0.98	14
Ni/Cu-N-C	Ni: 0.27 Cu: 0.31	15
Ni-Zn-N-C	Ni: 0.84 Zn: 1.32	16
Fe ₇ -Ni ₁ -N-C	Fe: 0.45 Ni: 0.42	17
NiFe-DASC	Ni: 4.05 Fe: 3.24	18
ZnLa-1/CN	Zn: 2.52 La: 1.16	19
Fe/Cu-N-C	Fe: 0.30 Cu: 0.19	20

Comment 4:

The as-prepared catalyst was denoted as FeCo-NCH (line 75). What does H stand for?

Response 4:

The letter “H” in the abbreviation of FeCo-NCH stands for “hollow” and is specially marked to distinguish it from the solid samples. To avoid any confusion, we have updated the related content in the revision (highlighted in red, page 3).

“we herein reported an interfacial assembly strategy for constructing binary single-atomic Fe/Co-N_x sites with a high ASD (denoted as FeCo-NCH)”

Comment 5:

The XRD pattern of ZIF-8@Fe was given in Supplementary Figure 2. However, ZIF-8@Fe was not mentioned elsewhere throughout the manuscript. The authors should revise Supplementary Figure 2 accordingly.

Response 5:

Thank you for the careful reading. As you suggested, we have revised Supplementary Figure 2 as shown below.

In the revised **Supplementary Information** (highlighted in red, page 3).

Supplementary Figure 2. XRD patterns of ZIF-8, ZIF-8@ZIF-67, and ZIF-8@ZIF-67@Fe.

Comment 6:

To prepare the ZIF-8@ZIF-67@Fe, the ZIF-8@ZIF-67 and Fe²⁺ ions were dispersed in EtOH and the mixture was stirred at RT for 10 min. In this process, Fe²⁺ ions should be adsorbed on the surface instead of entering the ZIF skeleton. Therefore, the statement of “Fe substitution” is inaccurate.

Response 6:

We have corrected the expression of “Fe substitution” to “Fe adsorption”.

In the revised main text: (highlighted in red, page 3)

“To achieve binary metal sites, Fe adsorption was further carried out on ZIF-8@ZIF-67 via a facile impregnation (ZIF-8@ZIF-67@Fe), the Co and Fe species exist in the same ZIF-67 layer.”

Comment 7:

The name of the same precursor should be unified throughout the manuscript and also the supplementary material (ZIF-8@ZIF-67, rather than ZIF-8@ZIF67).

Response 7:

Thanks for the careful reading. We have checked the precursor names in the entire manuscript and supporting materials and unified them to “ZIF-8@ZIF-67”.

Comment 8:

The authors should correct some mistakes in the manuscript, including: the ordinate of Figure 3b; the caption of Supplementary Figure 4; “adsorption” and “desorption” in Supplementary Figure 16a and 17a; “0.1256 cm⁻²” (line 407).

Response 8:

Thank you very much for the careful reading. We have checked the entire manuscript and Supplementary Information. The relevant typos have been corrected.

Responses to the Reviewer 5's comments

Manuscript: NCOMMS-22-33399

“Boosting the Accessible Site Density by Interfacial Assembly of Binary Atomic Metal - N_x for High-Performance Energy Devices” , by Zhe Jiang, Xuerui Liu, Xiao-Zhi Liu, Ying Liu, Ze-Cheng Yao, Yun Zhang, Qing-Hua Zhang, Lin Gu, Li-Rong Zheng, Youjun Fan, Tang Tang, Zhongbin Zhuang, Jin-Song Hu

In this work the electrocatalyst materials with dual-metal M-N_x active sites are prepared and tested for ORR performance in alkaline media using the rotating (ring)-disk electrode methods. These catalysts are employed as cathode materials in anion exchange membrane fuel cells and Zn-air batteries. The results obtained are important in the field of ORR electrocatalysis on non-PGM catalysts. The catalyst materials are thoroughly characterized by various physico-chemical methods. A discussion about the electrochemical properties of the studied catalysts needs further elaboration. The authors should determine the turnover frequency values and compare these with literature data. Overall. The paper is suitable for publication in this Journal. Recommendation: major revision

Brief response:

We greatly appreciate your valuable time, very careful reading, and constructive comments, which have definitely helped us to improve the quality of our manuscript. We have supplemented the corresponding data and discussion in the revision to clarify these points. Please see our responses below.

Reproduced specific comments and responses:

Comment 1:

Page 2, line 59. Refs. 17 and 18 do not contain the information presented, thus appropriate references are needed for ORR electrocatalysis on dual-atom M-N_x sites. Also, a discussion about the benefits of this type of bimetallic catalyst materials should be further elaborated.

Response 1:

Thank you for the careful reading and comment. We have accordingly updated the reference and added a detailed discussion to highlight the benefits of the atomically dispersed bimetallic catalyst.

The following discussion has been updated in the revision (highlighted in red, page 2, third paragraph).

“To further improve the intrinsic activity, binary dual-atomic M-N_x sites have been recently developed. They not only inherit the advantages of SACs, such as maximal atomic utilization, well-defined, and adjustable coordination configuration but also offer the possibility to improve catalytic performance by harnessing synergistic effects.^{15, 20} The interaction between multiple active centers

allows regulating of the electronic structure and geometric configuration, enabling to optimize of the adsorption and desorption of intermediates to approach the apex of the activity volcano plot.^{18, 21}

The corresponding references have been updated and highlighted in the **References** section:

15. Hossen M. M. *et al.* State-of-the-art and developmental trends in platinum group metal-free cathode catalyst for anion exchange membrane fuel cell (AEMFC). *Appl. Catal., B*, 121733 (2022).

18. Xiao, M., *et al.* Climbing the apex of the ORR volcano plot via binuclear site construction: Electronic and geometric engineering. *J. Am. Chem. Soc.* **141**, 17763-17770 (2019).

20. Li L., Yuan K. & Chen Y. Breaking the scaling relationship limit: From single-atom to dual-atom catalysts. *Acc. Mater. Res.* **3**, 584-596 (2022).

21. Zhang, W., *et al.* Emerging dual-atomic-site catalysts for efficient energy catalysis. *Adv. Mater.* **33**, 2102576 (2021).

Comment 2:

Page 2, line 35. “Such FeCo-NCH cathode electrocatalyst enables AEMFC or ZAB to deliver outstanding peak power densities of 569.0 or 414.5 mW cm⁻²,” This reviewer does not think that 569 mW cm⁻² is an outstanding peak power density for AEMFCs. Please re-phrase.

Response 2:

As you suggested, we have rephrased the corresponding description.

In the revised Abstract (highlighted in red, page 2):

“Such FeCo-NCH cathode electrocatalyst enables AEMFC or ZAB to deliver promising peak power densities of 569.0 or 414.5 mW cm⁻², 3.4 or 2.8 times higher than the devices with control catalyst FeCo-NC.”

Comment 3:

Page 2, line 50. Important literature is missing. See, for example, recent reviews by Sarapuu et al. DOI: 10.1039/c7ta08690c and Hossen et al. <https://doi.org/10.1016/j.apcatb.2022.121733>

Response 3:

We appreciate this useful information. We carefully reviewed the literature and updated the corresponding discussion and the references as Refs 14 and 15.

In the revision_(highlighted in red, second paragraph, page 2):

“it is necessary to develop efficient and durable non-PGM ORR electrocatalysts.^{14, 15”}

The corresponding references have been updated and highlighted in the **References** section:

14. Sarapuu, A., Kibena-Põldsepp, E., Borghei, M., Tammeveski, K. Electrocatalysis of oxygen reduction on heteroatom-doped nanocarbons and transition metal–nitrogen–carbon catalysts for alkaline membrane fuel cells. *J. Mater. Chem. A* **6**, 776-804 (2018).

15. Hossen M. M. *et al.* State-of-the-art and developmental trends in platinum group metal-free cathode catalyst for anion exchange membrane fuel cell (AEMFC). *Appl. Catal., B*, 121733 (2022).

Comment 4:

1. Page 2, lines 60-62. Reference 19 is not relevant, there are plenty of review articles available.

Response 4:

Thanks for the comment. We have updated it with the following new Refs 15 and 22.

In the revision (highlighted in red, last paragraph, page 2):

“few of them have demonstrated comparable performance to PGM catalysts in fuel cells or ZABs.^{15, 22”}

In the **References** section:

15. Hossen M. M. *et al.* State-of-the-art and developmental trends in platinum group metal-free cathode catalyst for anion exchange membrane fuel cell (AEMFC). *Appl. Catal., B*, 121733 (2022).

22. Asset, T., Atanasov, P. Iron-nitrogen-carbon catalysts for proton exchange membrane fuel cells. *Joule* **4**, 33-44 (2020).

Comment 5:

Page 2, line 62. “It is due to not only the insufficient intrinsic activity of M-N_x sites...”. These sites are considered to be the most active ones in alkaline media. This statement is not justified.

Response 5:

Thanks for the comment. As you mentioned, the M-N_x sites are considered to be the most active ones in alkaline media. However, the intrinsic ORR activity of these M-N_x sites still needs to be further enhanced. One of the strategies is to construct the binary dual-atomic M-N_x sites to optimize the adsorption and desorption of intermediates to approach the apex of the activity volcano plot.

The corresponding description has been rephrased as follows.

In the revision_(highlighted in red, last paragraph, page 2):

“It is probably due to the insufficient amount of available M–N_x sites in the catalyst layer.”

Comment 6:

XPS survey spectra of materials should be included in the manuscript as well and discussed.

Response 6:

Thank you for the suggestion. We have supplemented the XPS survey spectrum as Supplementary Figure. 10 and the discussion on page 5, paragraph 2.

In the revision_(highlighted in red, second paragraph, page 5):

“The XPS survey shows the coexists of the Fe, Co, N, O, and C in the catalyst (**Supplementary Figure. 10**)”

In the revised Supplementary Information (highlighted in red, page 11):

Supplementary Figure 10. The XPS survey spectra of FeCo-NCH.

Comment 7:

Page 6, lines 196-198. These findings should be discussed more in depth. Why in bimetallic material the overall metal and just Co loading can be higher without agglomerate formation?

Response 7:

Thanks for the insightful comment. Actually, we have optimized the synthetic conditions to maximize

metal loading in the single metal atomic catalysts (Fe-NCH and Co-NCH). The Fe loading in Fe-NCH can only reach ~2.7 wt% and the Co loading in Co-NCH can only reach ~5.2 wt%. If we further increase the amount of metal sources, the aggregated metal particles will inevitably appear in the catalyst, as shown in Figure R1. However, bimetallic FeCo-NCH SAC gives a total metal loading of 7.9 wt% (2.4 wt% for Fe and 5.5 wt% for Co). The metal loading is limited by the diffusion of active metal atoms and its interaction with the substrate and coordination with N-species during the pyrolysis. According to our experiences, the Fe atom is more active and prone to aggregate in such conditions so that we can achieve higher metal loading for Co-based SAC than Fe-based SAC. This phenomenon is also found in literature such as *Nat. Commun.* 10, 1278, 2019; *Nat Commun.* 10, 4585, 2019, etc. It seems such interactions for various metal does not interfere much with each other much in our experiments, given the total metal loading is still not high. The underlying fundamental science is needed to be further investigated.

Fig. R1. TEM images of the (a) Co-NCH and (b) Fe-NCH with ~8 wt % metal loading.

Comment 8:

Page 9. The authors should determine the turnover frequency (TOF) values of the catalysts for ORR at a specific potential (0.9 or 0.85 V vs. RHE) and compare these with literature data. The TOF value shows the intrinsic ORR activity of the M-N_x sites and is therefore highly relevant.

Response 8:

Thank you for the constructive comment. The TOF value shows the intrinsic ORR activity of the M-N_x sites and is highly related to the catalyst performance. Since the accessible site density (ASD) of the catalysts has been accessed by the nitrite stripping experiments, we estimated the TOFs from the decrease in ORR kinetic current on nitrite poisoning divided by ASD according to the literature

(Kucernak et al. In situ electrochemical quantification of active sites in Fe–N/C non-precious metal catalysts. Nat. Commun. 7, 13285 (2016)). This method has been widely used to determine ASD and TOF as can be found in Nat. Catal. 2, 259–268 (2019); Nat. Catal. 5, 311–323 (2022); J. Am. Chem. Soc. 144, 13487–13498 (2022). In this method, TOFs are commonly reported at 0.80 V. The TOF value of the FeCo-NCH catalyst is calculated to be $2.2 \text{ e}^{-1} \text{ s}^{-1} \text{ site}^{-1}$, which is higher than most of the previously reported ORR catalysts (**Supplementary Table 5**). It indicates a fast ORR kinetic on the binary single-atomic Fe/Co-N_x sites. The high TOF together with the high ASD contributes to the enhanced ORR performance for FeCo-NCH.

The following discussion has been added in the revision (highlighted in red, page 9):

“The TOF value of the FeCo-NCH is calculated to be $2.2 \text{ e}^{-1} \text{ s}^{-1} \text{ site}^{-1}$ at 0.80 V, which is higher than the most of the previously reported ORR catalysts (Supplementary Table 5), indicating a fast ORR kinetic.⁵¹ The high TOF together with the high ASD contributes to the enhanced ORR performance for FeCo-NCH.”

The corresponding TOF calculation method has been supplied in the **Methods** section (highlighted in red, page 12):

“The turnover frequency (TOF) was calculated by the following equation:

$$\text{TOF} [\text{e}^{-1} \text{ s}^{-1} \text{ site}^{-1}] = n_{\text{strip}} \times j_k (\text{mA cm}^{-2}) / Q_{\text{strip}} [\text{C g}^{-1}] \times L_c (\text{mg cm}^{-2})$$

Where n_{strip} (= 5) is the number of electrons associated with the reduction of one nitrite per site; F is the Faraday constant (96485 C mol^{-1}); j_k is the kinetic current density, L_c is the catalyst loading (0.27 mg cm^{-2}). M is the average atomic mass of Fe and Co for FeCo-NCH or FeCo-NC (based on the ratio of Fe and Co); W is the metal contents of FeCo-NCH or FeCo-NC.”

The following Table S5 has been supplemented in the revised **Supplementary Information** (highlighted in red, page 45).

“**Supplementary Table 5. Comparison of the site density (SD) and TOF performances for our electrocatalysts and reported PGM-free cathode electrocatalysts.**”

Catalysts	SD (site g ⁻¹)	TOF (e ⁻¹ s ⁻¹ site ⁻¹)	Ref.
FeCo-NCH	7.6×10^{19}	2.2 @ 0.80 V	This work
FeCo-NC	2.1×10^{19}	1.8 @ 0.80 V	This work
TPI@Z8(SiO ₂)-650-C	~	1.63 @ 0.80 V	48
Fe-N/C	7.2×10^{18}	1.6 @ 0.80 V	49
Fe-NC ^{A-DCDA}	4.69×10^{19}	0.13 @ 0.85 V	50

*Fe _{0.5} NC-800	3.99×10 ¹⁹	0.46 @ 0.9 V	51
PAJ	~2×10 ¹⁹	~0.7 @ 0.80 V	52
*PANI-Fe	~1×10 ²⁰	0.23 @ 0.90 V	53
CNRS	~6×10 ¹⁹	~0.2 @ 0.80 V	52
FeNC-CVD-750	1.92×10 ²⁰	0.8 @ 0.80 V	54

The * marks represent SD and TOF were calculated based on CO sorption and desorption while the others are based on the ASD measured by the nitrite stripping experiments.

Comment 9:

Page 9, lines 268-271. “The smaller Tafel slope means the faster ORR process.⁵¹ As expected,…” This claim is not justified. The authors should recall the meaning of Tafel slope in electrochemical kinetics (see, for example, any basic textbook of electrochemical kinetics).

Response 9:

Thanks for pointing out this unprecise description. The Tafel plots were introduced by Tafel in 1905 in the studies of HER (Tafel, J. Über die polarisation bei kathodischer wasserstoffentwicklung. *Z. Phys. Chem.* **50U**, 641-712 (1905).). A lower value of the Tafel slope means increasing the same current density requires a smaller overpotential, implying faster reaction kinetics.

We have rephrased this statement.

In the revision (highlighted in red, second paragraph, page 9):

“The smaller Tafel slope indicates enhanced ORR kinetics.^{52, 53}”

The corresponding references have been updated in the **References** section as 52 and 53.

52. Tafel, J. Über die polarisation bei kathodischer wasserstoffentwicklung. *Z. Phys. Chem.* **50U**, 641-712 (1905).

53. Wang, J., *et al.* Non-precious-metal catalysts for alkaline water electrolysis: Operando characterizations, theoretical calculations, and recent advances. *Chem. Soc. Rev.* **49**, 9154-9196 (2020).

Comment 10:

Page 9, 2nd paragraph. The mechanistic aspects of the ORR on dual-metal M-N_x sites need to be properly discussed. At present it remains unclear why these materials are highly ORR active.

Response 10:

We appreciate the constructive comment. To understand the synergistic effects between the Fe-N_x and Co-N_x species, we have performed the density functional theory (DFT) calculations to reveal the origin of the improved catalytic performance for ORR.

On the basis of the characterization results of XPS and EXAFS, we first constructed three catalyst models: FeN₅, CoN₄, and FeN₅-CoN₄ model, respectively (Supplementary Fig. 22). The ORR process on the Co sites and Fe sites were discussed respectively. The adsorption models of various intermediate states on three models with different sites during the ORR process were optimized as shown in Supplementary Fig. 23 (Co sites) and 25 (Fe sites). At the equilibrium potential of 0 V, all the elementary reaction steps on Co sites and Fe sites are exothermic, implying the ORR on the FeN₅-CoN₄, CoN₄, and FeN₅ is a spontaneous exothermal process.

For Co sites (Supplementary Fig. 24), the rate-determining step (RDS) for the CoN₄ model is the desorption of adsorbed *OH with the largest Gibbs free energy change ΔG_4 of 0.42 eV at $U = 1.23$ V. In contrast, the corresponding ΔG_4 for FeN₅-CoN₄ model is much small (0.25 eV). The RDS for the FeN₅-CoN₄ model is the adsorption of the *OOH from the first electron transfer step with a ΔG_1 of 0.29 eV. The limiting reaction energy barrier of ORR on FeN₅-CoN₄ (0.29 eV) is thus much lower than that on the CoN₄ model (0.42 eV). These results indicate the introduction of Fe sites promotes the desorption of *OH on the Co site, optimizing the ORR pathway.

For Fe sites (Supplementary Fig. 26), the RDS for both the FeN₅-CoN₄ model and FeN₅ model is the adsorption of the *OOH at $U = 1.23$ V. The RDS energy barrier on the FeN₅-CoN₄ model is 0.83 eV, also lower than that on the FeN₅ model (0.92 eV). These results suggest that the introduction of the Co site reduces the *OOH formation energy barrier in RDS on Fe sites.

These DFT calculation results clearly support that the combination of FeN₅ and CoN₄ moieties in FeN₅-CoN₄ generates a synergistic effect therebetween, justifying the enhanced ORR performance for FeCo-NCH compared with single-metal Fe-NCH or Co-NCH catalysts.

The following brief discussion and calculation details have been supplemented in the revised main text and **Methods** section, respectively (highlighted in red, page 9, first paragraph). The corresponding Supplementary Figures and detailed discussion have been provided in the Supplementary Information.

“The synergistic effect on binary Fe/Co-N_x sites for ORR was also evaluated by the density functional theory (DFT) calculations. Three models (i.e. FeN₅, CoN₄, and FeN₅-CoN₄) were constructed based on the EXAFS results (Supplementary Fig. 22). The downhill free energy pathways at $U = 0$ V on both Co and Fe sites in three models indicate the ORR is spontaneous exothermal (Supplementary Fig. 23-26).⁵⁰ The calculation results indicate that compared with Co or Fe-only site, the introduction of the Fe site promotes the desorption of *OH on the Co site (Supplementary Fig. 23, 24) while the introduction of the Co site significantly reduces the *OOH formation energy barrier on the Fe site (Supplementary Fig. 25, 26). In either case, the ORR pathway is promoted, suggesting the synergistic effect for FeN₅ and CoN₄ moieties and justifying the enhanced ORR activity for FeCo-NCH compared with single-metal Fe-NCH or Co-NCH. The detailed discussion can be found in Supplementary Information.^{21, 40} These results confirm that the constructing neighboring FeN₅ and CoN₄ moieties can generate a synergistic effect therebetween and contributes to enhanced ORR

performance.”

In the **Methods** section: (highlighted in red, pages 12-13)

“Density functional theory (DFT) calculations. All DFT calculations were performed by using Vienna ab initio Simulation Package (VASP).⁶² The generalized gradient approximation (GGA) with the Perdew-Burke-Ernzerh (PBE)⁶³ of the exchange correlation functional within the projector augmented wave method⁶⁴ was utilized to model the electron-ion interaction. An energy cutoff of 450 eV for the plane-wave basis set was used. The convergence threshold was set to 10⁻⁵ eV in energy and 0.02 eV/Å in force, respectively. To prevent the interaction between two neighboring images, the vacuum layer thickness was set to 15 Å. A semi-empirical van der Waals (vdW) correction proposed by Grimme (DFT-D3)⁶⁵ was included to account for the dispersion interactions. A 2×2×1 Monkhorst-Pack grid of k-points was used to sample the first Brillouin zones of the surfaces for structural optimizations. All atoms are fully relaxed during structural optimizations.

The free energy of each elementary step in the proton-coupled electron transfer (PECT) reactions was computed using the computational hydrogen electrode (CHE) model for oxygen reduction reaction (ORR). Considering the O₂ molecule was not broken before reduction, the associative 4e⁻ reduction pathway was evaluated to be most feasible for ORR in this work, as follows:

where * represents the active site on the corresponding surface.

The free energy of the reactants and each intermediate state at an applied electrode potential U were computed by the equations, $\mu = E + E_{\text{ZPE}} - TS$, $\Delta G = \Delta E + \Delta E_{\text{ZPE}} - T\Delta S$, then the ΔG^*_{O} , ΔG^*_{OH} and ΔG^*_{OOH} could be calculated as follows:

$$\begin{aligned} \Delta G^*_{\text{O}} &= \mu^*_{\text{O}} - \mu^* - (\mu_{\text{H}_2\text{O}} - \mu_{\text{H}_2}) \\ \Delta G^*_{\text{OH}} &= \mu^*_{\text{OH}} - \mu^* - (\mu_{\text{H}_2\text{O}} - \frac{1}{2}\mu_{\text{H}_2}) \\ \Delta G^*_{\text{OOH}} &= \mu^*_{\text{OOH}} - \mu^* - (2\mu_{\text{H}_2\text{O}} - \frac{3}{2}\mu_{\text{H}_2}) \end{aligned}$$

The Gibbs free energy of step 1-4 reaction could be expressed as:

$$\begin{aligned} \Delta G_1 &= \Delta G^*_{\text{OOH}} - 4.92 \text{ eV} + \text{eU} \\ \Delta G_2 &= \Delta G^*_{\text{O}} - \Delta G^*_{\text{OOH}} + \text{eU} \\ \Delta G_3 &= \Delta G^*_{\text{OH}} - \Delta G^*_{\text{O}} + \text{eU} \\ \Delta G_4 &= -\Delta G^*_{\text{OH}} + \text{eU} \end{aligned}$$

The following Supplementary Figures and detailed discussion have been supplemented in the revised **Supplementary Information** (highlighted in red, pages 23-27).

Supplementary Figure 22. Illustrations of optimized atomic configurations of (a) FeN₅, (b) CoN₄, and (c) FeN₅-CoN₄ model. (C: gray, N: blue, Fe: green, Co: pink, O: red, H: white).

Supplementary Figure 23. (a-c) Configurations of corresponding adsorbed intermediates (*OOH, *O, and *OH) on CoN₄ and (d-f) FeN₅-CoN₄ models (C: gray, N: blue, Fe: green, Co: pink, O: red, H: white).

Supplementary Figure 24. ORR free energy diagrams for the Co site in CoN₄ (green line) and in FeN₅-CoN₄ models (orange line) at U = 0 V, 0.9 V, and 1.23 V.

For Co sites, all the elementary reaction steps on both FeN₅-CoN₄ and CoN₄ models present a consistent downhill tendency at the U = 0 V, implying a spontaneous exothermal process. Upon increasing the thermodynamic equilibrium potential to 0.9 V, the ORR process on the FeN₅-CoN₄ model is still spontaneous, while the CoN₄ possesses a slight endothermic process from the desorption of *OH in the fourth step, suggesting that the external force is needed to drive this process. At U = 1.23 V, the rate-determining step (RDS) in the CoN₄ model is the desorption of adsorbed *OH with the largest Gibbs free energy change ΔG_4 of 0.42 eV. In contrast, the corresponding ΔG_4 in the FeN₅-CoN₄ model is much small (0.25 eV). The RDS for the FeN₅-CoN₄ model is the adsorption of the *OOH from the first electron transfer step with a ΔG_1 of 0.29 eV. The limiting reaction energy barrier of ORR on FeN₅-CoN₄ (0.29 eV) is lower compared with the CoN₄ model (0.42 eV). These results indicate that the introduction of the Fe site promotes the desorption of *OH on the Co site and optimizes the ORR pathway.

Supplementary Figure 25. (a-c) Configurations of corresponding adsorbed intermediates ($*\text{OOH}$, $*\text{O}$, and $*\text{OH}$) on FeN_5 and (d-f) $\text{FeN}_5\text{-CoN}_4$ models (C: gray, N: blue, Fe: green, Co: pink, O: red, H: white).

Supplementary Figure 26. ORR free energy diagrams for the Fe site in FeN₅ (orange line) and FeN₅-CoN₄ models (blue line) at U=0 V, 0.9 V, and 1.23 V.

As for Fe sites, when U = 0 V, all the elementary reaction steps on FeN₅-CoN₄ and FeN₅ models present a consistent downhill tendency, implying a spontaneous exothermal process. Upon increasing the thermodynamic equilibrium potential to 0.9 V, both the FeN₅-CoN₄ and FeN₅ present the uphill free energy in the first electron transfer step ($O_2 + H_2O + e^- \rightarrow *OOH + OH^-$), which is the RDS for the two models. It is noted that the FeN₅-CoN₄ model gives a lower energy barrier of 0.49 eV than that of the FeN₅ model (0.59 eV). Upon increasing the potential to 1.23 V, the first electron transfer step remains to be the RDS. The limiting reaction energy barrier of ORR on the FeN₅-CoN₄ (0.83 eV) model is still lower than that on the FeN₅ (0.92 eV) model, implying the ORR on the Fe site in FeN₅-CoN₄ is energetically favorable to that in the FeN₅. These results suggest that the introduction of the Co site significantly reduces the *OOH formation energy barrier on the Fe site and optimizes the ORR pathway.

Comment 11:

Page 9, 3rd paragraph. By comparing the AEMFC results with literature data, the author should make reference to previous studies in which FeCo-N-C materials were used as cathode catalyst.

Response 11:

Thanks for the comment. We have revisited the literature and supplemented the references using FeCo-N-C materials as cathode catalysts in **Supplementary Table 4**, as highlighted in the following table.

“Supplementary Table 4. Comparison of the RDE and AEMFC performances for our electrocatalysts and various reported PGM-free cathode electrocatalysts (H₂/O₂ and H₂/Air).”

Catalysts	Ionomer	Membrane	$E_{1/2}$ (V vs. RHE)	P_{max} (H ₂ -O ₂) (mW cm ⁻²)	P_{max} (H ₂ -Air) (mW cm ⁻²)	Operating conditions	Ref.
FeCo-NCH	PAP-TP-100	PAP-TP-85	0.89	569.0	299.3	80 °C, 100% RH, 200 kPa	This work
Pt/C	PAP-TP-100	PAP-TP-85	0.86	750.7	508.1	80 °C, 100% RH, 100 kPa	This work
Fe/N/C nanotubes	aQAPS-S ₁₄	aQAPS-S ₈	0.93	485		60 °C, 100% RH	21
Fe-NMG	AS-4	Tokuyama A201	0.83	218		60 °C, 100% RH	22
Fe-Co-N-C	ETFE-RG	RG-LDPE	0.76	420		60 °C, 52% RH	23

SNBC12	FAA-3-Br	FAA-3-50	0.85	215		60 °C, 80% RH	24
Fe-N-C	ETFE-RG	HDPE-AEM	0.84	2050	~1000	80 °C, 200 kPa	25
MnCo ₂ O ₄ /C	QAPPT	QAPPT	0.84	1200		80 °C, 100% RH, 100 kPa	26
FeCN-S-800	Fumion FAA-3	Fumapem FAA-3	0.76	125		50 °C	27
Fe _{0.5} -N-C	I2, Acta S.P.A.	A901-Tokuyama	0.85	504		50 °C, 100% RH, 150 kPa	28
MCS	aQAPS-S ₁₄	aQAPS-S ₈	0.85	1100		60 °C, 50% RH, 100 kPa	29
NFC@Fe/Fe ₃ C-9	FAA-3-SOLUT-10	Fumapem FAA-3	0.87	273		60 °C, 100% RH, 250 kPa	30
Fe/Ce-NCNW	FLN-55	m-TPN	0.915	500		80 °C, 100% RH	31
Co@G/C	I2	A201	0.8	412		60 °C, 100% RH	32
Co-N-CDC/CNT _{mel}	HMT-PMBI	HMT-PMBI	0.82	577		60 °C, 100% RH, 200 kPa	33
Co-NC	FAA-3	FAA-3	0.904	271		60 °C, 100% RH	34
Fe-N-Gra	HMT-PMBI	HMT-PMBI	0.77	243		60 °C, 100% RH	35
Fe/IL-PAN-A1000	HMT-PMBI	HMT-PMBI	0.74	289		60 °C, 100% RH, 200 kPa	36
Zn/Fe _{SA} -PC/950/HN ₃	FAA-3	FAA-3-20	0.87	352		60 °C, 100% RH	37
FeCoNC-at	HMT-PMBI	HMT-PMBI	0.829	415		60 °C, 100% RH, 200 kPa	38
Fe-N-C	HMT-PMBI	HMT-PMBI	0.89	220		60 °C, 80% RH	39
FeN-SiCDC	HMT-PMBI	HMT-PMBI	-	356		60 °C, 82% RH, 200 kPa	40
FeCoN-MWCNT	FAA-3	VMFAA-3-10-rf	0.86	692	~420	60 °C, 100 kPa	41
CoFe-N-CDC/CNT	ETEE	ETFE-BTMA	0.83	1120	800	60 °C, 100% RH	42
PBC/900/M	FAA-3	FAA-3-20	0.862	658		60 °C, 100% RH	43
Fe/Co/IL-CNF-800b	-	HMT-PMBI	0.859	195		60 °C, 80% RH, 200 kPa	44
FeN _x -CNTs	-	PVA-0.8PQVBC40%	0.94	1150		60 °C, 200 kPa	45

Co ₃ N/C	QAPPT	QAPPT	0.862	700	80 °C, 100% RH	46
Fe-N-MPC	HMT-PMBI	HMT-PMBI	0.89	473	60 °C, 50% RH, 200 kPa	47

Comment 12:

The RRDE measurements. The experimental details of RRDE measurements are lacking, e.g. (i) which ring electrode was used? (ii) at which potential the ring electrode was kept; (iii) was there any pretreatment of the ring electrode used prior to the ORR measurements? These details of the RRDE measurements need to be added. Also, the obtained ring currents are unreadable in Figure S20, which needs to be fixed. Use a proper scale for the ring current values.

Response 12:

We appreciate these suggestions. The details for the RRDE measurements have been supplemented in electrochemical measurement in the **Methods** section. The original Figure S20 has been replotted as Supplementary Figure 28 (other supplementary figures were added to respond to the reviewers' comments) to clearly show the ring current.

In the revised **Methods** section (highlighted in red, page 12):

The rotating ring-disk electrode (RRDE) (4 mm in diameter) measurements were also tested in O₂-saturated 0.1 M KOH with the studying catalysts as the working electrode. The ring electrode is Pt and the ring potential was kept at 1.26 V.

Prior to use, the RRDE was polished with 0.3 and 0.05 μm alumina slurries and then cleaned by subsequently sonicated in Milli-Q water and 2-propanol to a mirror finish state.

In the revised **Supplementary Information** (highlighted in red, page 29):

Supplementary Figure 28. RRDE measurement curves of (a) FeCo-NCH, (b) FeCo-NC, (c) Pt/C, (d) Co-NCH, and (e) Fe-NCH, recorded in O₂-saturated 0.1 M KOH under a rotating rate of 1600 rpm.

Comment 13:

The SCN⁻ anion poisoning test was conducted in acidic media. Why not in alkaline solution?

Response 13:

Thank you very much for the comment. First of all, the SCN⁻ is unstable in the alkaline media thus the ORR activity of the poisoned active sites can be easily recovered in the alkaline media (Li et al. *Angew. Chem., Int. Ed.* **56**, 6937-6941 (2017)). Second, as widely reported in the literature (for example, Shi et al. *Small*, **14**, 1704319 (2018)). Under alkaline conditions (0.1 M KOH), the ORR activity of M-N-C may only decrease slightly when the M-N_x sites are poisoned by SCN⁻, which can be attributed to other active nitrogen species (eg. pyridine nitrogen) in the catalyst since the SCN⁻ is only specifically adsorbed on the metal site (Junji Nakamura et al. *Science* **351**, 361-365 (2016)). Therefore, SCN⁻ anion poisoning test under alkaline solutions can not properly reflect the poison properties of the M-N_x sites. In contrast, under acidic conditions, the ORR activity of the M-N_x can be significantly reduced, due to the other active nitrogen species showing poor ORR activity in acid media. Therefore, for M-N_x sites such as FeCo-NCH, in order to verify the existence of the true single atomic active site, it is necessary to test under acidic media rather than alkaline media.

Comment 14:

Comparison with H₂/air AEMFC results from the literature should be added.

Response 14:

As you suggested, we have added the literature with H₂/air AEMFC results for comparison in **Supplementary Table 4**, as shown in the Response to Comments 11.

Comment 15:

The Co-NCH catalyst seems to have rather similar ORR electrocatalytic activity to FeCo-NCH, but the performance of Fe-NCH is significantly worse. What could be the reason for that?

Response 15:

We think there are two possible reasons for this phenomenon. 1) **The difference in the intrinsic activity of the M-N_x sites in Co-NCH and Fe-NCH.** It is known that the intrinsic activity of the M-N_x sites is strongly dependent on their configuration. In our case, Fe-N₅ sites dominate in the Fe-NCH and Co-N₄ sites dominate in the Co-NCH according to the XAS results. We have supplemented the DFT calculations. The results show that the ORR process is more favorable on the Co-N₄ sites than that on the Fe-N₅ sites. As shown in **Supplementary Figures 24 and 26**, the RDS of the CoN₄ model is the last electron transfer step of the desorption of *OH with a free energy of 0.42 eV. While for the FeN₅ model, the RDS is *OOH adsorption with a free energy of 0.92 eV, which is much higher than that of the CoN₄ model. This result implies that CoN₄ may possess a better ORR activity than FeN₅. 2) **The Co-NCH has more active sites than the Fe-NCH** (5.5 wt% for Co vs. 2.4 wt% for Fe). With these two effects, the Co-NCH shows higher ORR performance than the Fe-NCH.

The following description has been added to the discussion of the DFT results in the **Supplementary Information** (highlighted in red, last paragraph, page 27):

“Moreover, the DFT calculation results show that the ORR process is more favorable on the Co-N₄ sites than that on the Fe-N₅ sites. The RDS of the Co-N₄ model is the last electron transfer step of the desorption of *OH with a free energy of 0.42 eV. While for the FeN₅ model, the RDS is *OOH adsorption with a free energy of 0.92 eV, which is much higher than that of the CoN₄ model. This result implies that CoN₄ may possess a better ORR activity than FeN₅ in our case.”

Comment 16:

Figures 5a, d, g. These Figures are not informative and should be removed from the manuscript.

Response 16:

As you suggested, we have removed Figures 5a, d, g into supplementary information and reorganized Figure 5 as shown below.

Fig 5. Performance of H₂-O₂ AEMFCs and Zn-air batteries. (a) AEMFCs performance of FeCo-NCH and control samples. (b) H₂-Air (CO₂ free) AEMFCs performance of FeCo-NCH and control samples. (c) Optical image of an LED light array powered by two Zn-air batteries in series using FeCo-NCH as the air cathode. (d) Discharge polarization curves and corresponding power density curves of FeCo-NCH and control samples. (e) Discharge curves of the FeCo-NCH-based Zn-air battery at current densities of 2, 5, 10, 25, 50, and 2 mA cm⁻² for 120 min with each step being 20 min. (f) Discharge curves of FeCo-NCH-based Zn-air battery at 5 mA cm⁻² for 100 h.

Comment 17:

Page 2, line 44. A recent review by Hussain et al. dealing with ORR electrocatalysis on Pt-based catalysts is missing (<https://doi.org/10.1016/j.ijhydene.2020.08.215>). It needs to be cited.

Response 17:

Thanks for the useful information. We have added this comprehensive review in the revision as Ref 7. In the **References** section:

7. Hussain S. *et al.* Oxygen reduction reaction on nanostructured Pt-based electrocatalysts: A review. *Int. J. Hydrogen Energy* **45**, 31775-31797 (2020).

Comment 18:

In Supplementary Tables S2, S4 and S5 the references should be replaced with number and the full list of references should be given in the end of the Supplementary Information.

Response18:

Thank you for the comment. As you suggested, we have replaced the references in Supplementary Tables S2, S4, and S5 with numbers and supplemented the corresponding reference list at the end of the **Supplementary Information**.

Reproduced minor remarks and responses:

We greatly appreciate you for the very careful reading and for pointing out these minor issues/typos in writing and discussion. The related corrections have been made in the revision accordingly.

1. Page 2, line 24. “M-N-C catalyst” should be the last keyword.

Response:

Thanks for the suggestion. The “M-N-C catalyst” has been added as the last keyword.

2. Page 2, line 36. “...3.4 or 2.8 times higher than control devices.” As the control devices are not specified, then this part needs re-phrasing or removing.

Response:

We have rephrased the description as follows.

In the revision (highlighted in red, Abstract, page 2):

“..., 3.4 or 2.8 times higher than the devices with the control catalyst FeCo-NC.”

3. Page 2, line 60. “...shown excellent ORR activities on rotating disk electrodes (RDE),” should be re-phrased.

Response:

We have rephrased the description as follows.

In the revision (highlighted in red, last paragraph, page 2):

“Moreover, although a variety of M–N–C catalysts have shown excellent ORR activities in rotating disk electrode (RDE) tests, few of them have demonstrated comparable performance to PGM

catalysts in fuel cells or ZABs.”

4. In alkaline media HO_2^- forms instead of H_2O_2 . This should be corrected throughout the manuscript.

Response:

Thank you very much for the input. The relevant descriptions have been corrected in the entire manuscript and Supplementary Information including the Figures and **Methods** section.

In the **Methods** section: (highlighted in red, page 12):

“The following formula was applied to calculate the HO_2^- yield (HO_2^- %) and the electron transfers number (n), respectively:

$$(\text{HO}_2^- \%) = 200 \times (I_r/N) / (I_r/N + I_d)$$

$$n = 4 \times I_d / (I_r/N + I_d)$$

where I_d and I_r represent disk and ring current, respectively, and N is the ring collecting efficiency (0.37).”

The ordinate of **Supplementary Figure 29**. has been corrected.

In the revised **Supplementary Information**: (highlighted in red, page 30)

Supplementary Figure 29. HO_2^- yield and calculated electron transfer number during ORR for FeCo-NCH, Pt/C, and control samples.

5. Page 8, line 236. “AEMFC discharge performance” should be “AEMFC performance”

Response:

We have changed the description as suggested.

6. Page 8, line 334. “and energy devices” should be “and electrochemical energy devices”

Response:

We have changed the description as suggested.

7. Page 10. The origin and purity of used chemicals is missing, this information should be added.

Response:

Thank you very much for the comment. We have supplemented the origin and purity of used chemicals in the **Methods** section (highlighted in red, page 11).

“Chemicals and materials.

Cobalt (II) nitrate hexahydrate (98%), iron(II) sulfate heptahydrate (98%), zinc nitrate hexahydrate (99%), 2-methylimidazole (99%) zinc acetate dihydrate (98%), Pt/C (20%), dopamine hydrochloride (99%), 3-tris (hydroxymethyl) aminomethane (99.8%-100.1%), sodium thiocyanate (98%), potassium hydroxide (99.98%), sodium acetate trihydrate (99%), glacial acetic acid (99.9985%), and zinc foil (99.994%) were purchased from Alfa Aesar. Methanol and ethanol were received from Beijing Chemical Work Co. in analytic grade (AR). All chemicals were used as received without further purification. Nafion (5 wt%, DuPont) was obtained from commercial suppliers. Milli-Q ultrapure water (resistance of 18.2 MΩ·cm at 25 °C) was used for all experiments.”

8. Page 10, line 315. “high than” should be replaced with “higher than”

Response:

We have changed the description as suggested.

9. Page 11, line 407. “0.5 Nafion solution” What is meant here? Also, it should be “glassy carbon”, not “glass carbon”. The unit of the surface area is also wrong. Please correct.

Response:

We have made the corrections as suggested.

In the revision (highlighted in red, page 12).

“The working electrode was prepared by drop-casting 12 μL of catalyst ink (2 mg FeCo-NCH in 500 μL ethanol) and 0.5 μL of 0.5 wt% Nafion solution onto a glassy carbon rotating disk electrode (RDE, area: 0.1256 cm^2), giving a catalyst loading of 0.33 mg cm^{-2} .”

10. Page 11, line 413. “The K-L equations” should be “The Koutecky-Levich (K-L) equation”

Response:

We have clarified the description as suggested.

11. Experimental details of H_2 /air AEMFC are missing.

Response:

Thank you very much for the comment. The experimental details of H_2 /air AEMFC are supplemented in the **Methods** section.

In the revision (highlighted in red, page 13).

“ H_2 /Air single-cell AEMFCs were tested at 80 $^\circ\text{C}$. H_2 and air were fully humidified at 80 $^\circ\text{C}$ (100% RH) and fed at a flow rate of 1000 mL/min and a backpressure of 100 kPa symmetrically on both sides.”

12. Tables S4 and S5. There are some errors, so the authors should check the data in these Tables and correct it (e.g. Fe-N-Gra was tested at 60 $^\circ\text{C}$ not 80 $^\circ\text{C}$; CoFe-N-CDC should be CoFe-N-CDC/CNT).

Response:

Thank you very much. We have checked Table S4 and Table S5 carefully and corrected the incorrect descriptions as suggested.

In the revised **Supplementary Information** (highlighted in red, pages 43-44).

Fe-N-Gra	HMT- PMBI	HMT-PMBI	0.77	243		60 $^\circ\text{C}$, 100% RH	35
CoFe-N- CDC/CNT	ETEE	ETFE-BTMA	0.83	1120	800	60 $^\circ\text{C}$, 100% RH	42

13. The value of the electrode rotation rate used is missing from Figure S15, S20, and S22 captions.

Response:

We have supplemented the electrode rotation rate in the captions of **Supplementary Figures 15, 20, and 22**.

In the revised **Supplementary Information** (highlighted in red, page 19).

Supplementary Figure 15. (**Supplementary Figure 18**, now)

“LSV scanning experiments to determine the SD of FeCo-NCH and FeCo-NC through reversible nitrite poisoning O₂-saturated 0.5 M acetate buffer under a rotating rate of 1600 rpm.”

In the revised **Supplementary Information** (highlighted in red, page 29).

Supplementary Figure 20. (**Supplementary Figure 28**, now)

“RRDE measurement curves of (a) FeCo-NCH, (b) FeCo-NC, (c) Pt/C, (d) Co-NCH, and (e) Fe-NCH, recorded in O₂-saturated 0.1 M KOH under a rotating rate of 1600 rpm.”

In the revised **Supplementary Information** (highlighted in red, page 31).

Supplementary Figure 22. (**Supplementary Figure 30**, now)

“Poisoning experiments of FeCo-NCH in an O₂-saturated 0.1 M HClO₄ electrolyte under a rotating rate of 1600 rpm with and without NaSCN in the electrolyte. The significantly degraded ORR activity suggested that the active sites should be ascribed to metal-N_x sites.”

REVIEWER COMMENTS

Reviewer #1 (Remarks to the Author):

Hu and co-workers have addressed the major points raised by the reviewers, and the revised manuscript now appears suitable for publication in Nature Commun.

Reviewer #2 (Remarks to the Author):

Thanks to the authors for making a great effort in revising this manuscript. However, this reviewer still felt uncomfortable supporting its publication in the journal due to its insufficient impact and novelty. Developing nonprecious metal catalysts for the ORR in ALKALINE media is not challenging anymore. Many similar Fe/N/C and carbon catalysts have been reported in the last decade, showing exceptional performance and even being superior to Pt catalysts.

This manuscript presented comprehensive but routine characterization and DFT calculations to justify the importance of the Fe-N-C catalyst concerning its zinc-air battery performance. The authors claimed that one of the most significant innovations is the hollow carbon structures. However, similar carbon nanostructures have been studied before for the Fe-N-C catalysts. Also, this work did not demonstrate sufficiently important research to solve the problem the community is facing. Therefore, the reviewer still holds a high standard and does not support its publication in the journal.

Reviewer #3 (Remarks to the Author):

The comments in last report have been well addressed in the revised manuscript. I would be pleased to recommend acceptance of this work in Nature Communications.

Reviewer #4 (Remarks to the Author):

The issues raised by this reviewer have been addressed. This reviewer has no further comments.

Reviewer #5 (Remarks to the Author):

Manuscript: NCOMMS-22-33399A

“Boosting the Accessible Site Density by Interfacial Assembly of Binary Atomic Metal–Nx for High-Performance Energy Devices”, by Zhe Jiang, Xuerui Liu, Xiao-Zhi Liu, Ying Liu, Ze-Cheng Yao, Yun Zhang, Qing-Hua Zhang, Lin Gu, Li-Rong Zheng, Youjun Fan, Tang Tang, Zhongbin Zhuang, Jin-Song Hu

In my opinion the authors satisfactory answered most of the reviewers' comments and questions. However, there are still some issues that need to be addressed. Recommendation: minor revision

Further comments to the authors:

1. Page 5, lines 146-148. While the XPS survey spectra are added as suggested by the reviewer, the elemental contents are not mentioned herein. Reference to Table S3 is probably missing. The discussion about the XPS results is somewhat disjoint, some of it is on page 5, some on page 7. In Figure S10 the surface oxygen content is rather high. What could be the reason for that?
2. Page 7, lines 217-219. Why is the metal content different in Fe-NCH and Co-NCH samples? Also, the bimetallic FeCo-NCH catalyst has different iron and cobalt contents. Why is that?
3. Page 8, lines 269-270 (Figure S21). Fe-NCH is less active catalyst for ORR than Co-NCH in terms of half-wave potential and limiting current densities. What could be the reason for that? According to many studies the ORR activity of Fe-N-C should not be lower than that of Co-N-C.
4. Page 9, line 286. The TOF value is determined at 0.8 V. At this potential the ORR process on a FeCo-NCH catalyst is under diffusion control. The TOF value should be determined at 0.9 V.
5. Page 9, line 298. "The smaller Tafel slope indicates enhanced ORR kinetics.^{52, 53}" This claim is too simplistic. The sentence and refs. 52, 53 should be removed from the manuscript.
6. Page 9, lines 302-305. The n value is around 3.7, so still lower than 4, which means that there is some 2-electron ORR occurring as is also evident from peroxide yield in the RRDE studies. It would be correct to state that the ORR proceeds mostly via 4-electron pathway. Also, the claim "less than ~ 5%" is not true – the yield is ~5%, at 0.6-0.8 V slightly above 5% (Figure S29). The formation of peroxide is detrimental in fuel cells. It reduces the fuel cell efficiency and degrades the catalysts and membranes. The authors should discuss about these aspects in a short paragraph in the main manuscript.
7. A comparison of AEMFC results with literature data is still insufficient. More discussion about the results should be added to the text. For example, Kumar et al. reported a better AEMFC performance of CoFe-N-C catalysts (<https://doi.org/10.1021/acsami.1c06737>). While the H₂/air fuel cell data is included in Table S4, there is no discussion about it. For some reason the authors have decided to exclude results over 1000 mW cm⁻² from Figure S34. Also, there is no comparison brought out with PAP-TP-85 membrane, as membrane plays crucial role, then comparing results obtained with the same membrane would be more informative.

Minor remarks:

SAC is not defined.

Page 2, lines 65-67. "The interaction between multiple active centers allows regulating the electronic structure and geometric configuration, enabling to optimize of the adsorption and desorption of intermediates". This sentence is a bit confusing and should be re-phrased.

Page 5, lines 162 and 187. 'references' should be replaced with 'reference materials'

Page 8, line 261 and elsewhere. The term "mesoporous shell" etc. is a bit misleading as the nanocage structure contains plenty of micropores together with small mesopores. Please correct.

Page 10, line 362. The title should be 'Conclusions' instead of 'Discussion'.

Page 11, line 390. 'Nafion' should be 'Nafion'

Page 12, lines 502 and 504. It should be 'Vulcan' not 'Vulkan'.

Page 12. The origin of PAP-TP ionomer and membrane is missing.

Page 11, line 551. Ref. 15 must be updated Appl. Catal. B 325, 121733 (2023)

Was any electrochemical cleaning/activation applied to the Pt-ring prior to the ORR measurements?

Figure S32. The x-axis values should start from 0 mA cm⁻².

Responses to the Reviewers' Comments

Responses to Reviewer #1's comments

Reviewer #1:

Hu and co-workers have addressed the major points raised by the reviewers, and the revised manuscript now appears suitable for publication in Nature Commun.

Response:

We greatly appreciate the constructive comments from Reviewer 1, which has helped us to improve the quality of our manuscript.

Responses to Reviewer #2's comments

Reviewer #2:

Thanks to the authors for making a great effort in revising this manuscript. However, this reviewer still felt uncomfortable supporting its publication in the journal due to its insufficient impact and novelty.

Developing nonprecious metal catalysts for the ORR in ALKALINE media is not challenging anymore. Many similar Fe/N/C and carbon catalysts have been reported in the last decade, showing exceptional performance and even being superior to Pt catalysts.

This manuscript presented comprehensive but routine characterization and DFT calculations to justify the importance of the Fe-N-C catalyst concerning its zinc-air battery performance.

The authors claimed that one of the most significant innovations is the hollow carbon structures. However, similar carbon nanostructures have been studied before for the Fe-N-C catalysts. Also, this work did not demonstrate sufficiently important research to solve the problem the community is facing.

Therefore, the reviewer still holds a high standard and does not support its publication in the journal.

Response:

We appreciate your critic comments, which helped us to improve the quality of our manuscript.

We have carefully looked up the relevant information and literature and respond your concerns as below.

Regarding your concern about the non-PGM catalyst performance in alkaline media and the novelty of the manuscript. Although the ORR $E_{1/2}$ of some recently reported advanced M/N/C catalysts in RDE tests is approaching or even superior to benchmark Pt catalysts, the MEA performance of the M/N/C catalysts still falls far behind Pt-based catalysts nowadays. We have carefully looked up the recently reported literature. **Thousands of single-atomic catalysts have been reported for alkaline ORR in the last few years, however, few of them have demonstrated the high performance in alkaline fuel cell devices (Fig. R1, data obtained from Google Scholar).** It suggests the challenge still remains for their practical applications in devices.

Despite the encouraging activity improvement, no M/N/C catalysts can practically replace the Pt catalysts in AEMFCs currently. To achieve comparable fuel cell performance to the Pt catalysts, M/N/C catalysts usually required **tens of times catalyst loading** in the MEA catalyst layer, which leads to mass transfer issues. The huge gap between the superior RDE performance and poor MEA performance lies in the accessible catalytic sites in the assembled catalyst layer structure, which is now receiving ever-growing attention (Shui e et al. *Nat. Catal.* **2**, 259–268 (2019); Jaouen et al. *Nat. Mater.* **20**, 1385–1391 (2021); Kucernak et al. *Nat. Catal.* **5**, 311–323 (2022), etc.). **Therefore, the investigation of structure-performance relationship between the density and utilization of active sites of M/N/C catalysts and fuel cell devices is of great significance and critical for conveying the high RDE performance to the high MEA performance. Our manuscript is focusing on this important topic in the communities and provided new insights which are**

expected to benefit the design of efficient electrocatalysts for not only fuel cells themselves also other related devices with membrane electrodes such as water electrolyzers and CO₂ electrolyzers.

Figure R1. An approximate number of publications on the single-atomic type catalysts for alkaline ORR and AEMFC in the last few years. The data were obtained from Google Scholar.

The main highlights of our manuscript does not lie in the hollow carbon structure itself. We directly revealed the significance of site utilization and asseccible site density in achieving high-performance energy devices by designing both hollow and solid M/N/C catalysts. Although a couple of M/N/C hollow structures have been previously reported, **none of them focused on improving site density utilization for enhancing the device performance.** In our work, we focused on the relationship between fuel cell device performance and the accessible active site density (ASD) and demonstrated facile strategies to boost the device performance. By concentrating the active site at the interface to raise the ASD from 2.1×10^{19} to 7.6×10^{19} sites g^{-1} and site utilization from 2.5% to 9.3% (compared with solid ones), the catalyst FeCo-NCH exhibited significantly improved AEMFC performance, which is also applicable for Zn-air batteries.

We believe these results would be appealing to the community and a broad range of readerships for a comprehensive journal like Nat. Commun.. By addressing the comments your raised with supplemented data and evidences, we hope the reivewer could support the acceptance of this manuscript for publication.

Responses to Reviewer #3&4's comments

Reviewer #3:

The comments in last report have been well addressed in the revised manuscript. I would be pleased to recommend acceptance of this work in Nature Communications.

Response:

We greatly appreciate the constructive comments from Reviewer #3, which has helped us to improve the quality of our manuscript.

Reviewer #4:

The issues raised by this reviewer have been addressed. This reviewer has no further comments.

Response:

We greatly appreciate the constructive comments from Reviewer #4, which has helped us to improve the quality of our manuscript.

Responses to Reviewer #5's comments

Manuscript: NCOMMS-22-33399A

“Boosting the Accessible Site Density by Interfacial Assembly of Binary Atomic Metal-Nx for High-Performance Energy Devices” , by Zhe Jiang, Xuerui Liu, Xiao-Zhi Liu, Ying Liu, Ze-Cheng Yao, Yun Zhang, Qing-Hua Zhang, Lin Gu, Li-Rong Zheng, Youjun Fan, Tang Tang, Zhongbin Zhuang, Jin-Song Hu

In my opinion the authors satisfactory answered most of the reviewers' comments and questions. However, there are still some issues that need to be addressed. Recommendation: minor revision

Brief response:

We greatly appreciate your valuable time, very careful reading, and constructive comments, which have definitely helped us to improve the quality of our manuscript. For the specific comments, we have supplemented the corresponding data and discussion in the revision to clarify these points. Please see our responses below.

Reproduced specific comments and responses:

Comment 1:

Page 5, lines 146-148. While the XPS survey spectra are added as suggested by the reviewer, the elemental contents are not mentioned herein. Reference to Table S3 is probably missing. The discussion about the XPS results is somewhat disjoint, some of it is on page 5, some on page 7. In Figure S10 the surface oxygen content is rather high. What could be the reason for that?

Response 1:

Thank you for the comment. We have reorganized the discussion about the XPS results and integrated it on page 5. The elemental contents are now provided in **Supplementary Table 3** and properly referred and discussed. **Supplementary Figs. 10 and 11** were also updated. The Fe, Co, N, C, and O content is 0.50, 1.12, 9.23, 79.81, and 9.34, at.% respectively. The high surface oxygen content of FeCo-NCH mainly comes from the surface carbon-oxygen functionalities as evidenced by the high-resolution O *1s* spectrum in updated **Supplementary Fig. 11c** with the absence of lattice O at 528-529 eV. This phenomenon is commonly observed in carbon-supported SAC catalysts prepared by pyrolysis (*ChemCatChem* **12**, 4568–4581 (2020); *ChemElectroChem* **7**, 1739–1747 (2020); *J. Mater. Chem. A* **3**, 3559–3567 (2015); *ACS Catal.* **12**, 1216–1227 (2022); *Adv. Mater.* **32**, 1906905 (2020)).

The following description has been updated in the revision (highlighted in red, last paragraph, page 5-6)

“The surface Fe and Co content examined by XPS is 0.5 at.% (2.1 wt%) and 1.12 at.% (4.6 wt%), respectively, giving a total surface metal content of 6.7 wt% (Supplementary Table 3). The N, O, and C content is 9.23, 9.34, and 79.81 at.%, respectively (Supplementary Figs 10 and 11). The surface oxygen in the FeCo-NCH originates from the carbon-oxygen functionalities, which are commonly observed in carbon-supported materials.^{32, 33”}

Corresponding references have been updated in the **References** section as 32 and 33.

32. Mooste, M., *et al.* Electrospun polyacrylonitrile-derived Co or Fe containing nanofibre catalysts for oxygen reduction reaction at the alkaline membrane fuel cell cathode. *ChemCatChem* **12**, 4568-4581 (2020).

33. Sibul, R., *et al.* Iron- and nitrogen-doped graphene-based catalysts for fuel cell applications. *ChemElectroChem* **7**, 1739-1747 (2020).

Comment 2:

Page 7, lines 217-219. Why is the metal content different in Fe-NCH and Co-NCH samples? Also, the bimetallic FeCo-NCH catalyst has different iron and cobalt contents. Why is that?

Response 2:

Thank you for the good comment. Different metallic elements show different thermal-reducing behavior. It has been demonstrated that Fe is more active than Co under high-temperature pyrolysis and more easily tend to aggrate to nanoparticles under a high metal loading. Our experiments also demonstrated that in the preparation of monometallic samples, further increasing the Fe loading leads to the formation of Fe particles, while the formation of Co nanoparticles occurs at a relatively higher Co loading. It is commonly observed that the higher loading of Co than Fe in SAC catalysts can be obtained using similar strategies in literature reports (such as *J. Am. Chem. Soc.* **144**, 4913–4924 (2022); *Nat. Mater.* **21**, 681-688 (2022); *Adv. Mater.* **32**, 1906905 (2020); *Nat. Commun.* **10**, 1278 (2019); *Nat. Commun.* **10**, 3663 (2019), etc.). In addition, according to the coordination field theory, the tetrad coordination structure Co is more stable than Fe, thus the Co will be more dominant than the Fe when replacing the Zn nodes in ZIF-8. This may also result in different doping amounts of Co and Fe in the final samples.

Comment 3:

Page 8, lines 269-270 (Figure S21). Fe-NCH is less active catalyst for ORR than Co-NCH in terms

of half-wave potential and limiting current densities. What could be the reason for that? According to many studies the ORR activity of Fe-N-C should not be lower than that of Co-N-C.

Response 3:

We appreciate this comment. Indeed, some literatures reported that Fe-N-C showed the better ORR activity than Co-N-C. In our case, the inferior ORR performance of Fe-N-C than Co-N-C can be attributed to two main reasons:

(1) The difference in the intrinsic activity of the M-N_x sites in Co-NCH and Fe-NCH.

The intrinsic ORR activities of Fe-N_x are directly related to their coordination structures (Atanassov et al. *Joule* **4**, 33-44 (2020)). Most of the excellent Fe-N-C materials in the literatures have the Fe-N₄ sites (such as *Energy Environ. Sci.* **15**, 2619-2628 (2022); *J. Am. Chem. Soc.* **144**, 9280-9291 (2022); *Angew. Chem.Int. Ed.* **61**, e2022010 (2022); *Nat. Commun.* **13**, 57 (2022)). In our case, Fe-N₅ sites dominate in the Fe-NCH while Co-N₄ sites dominate in the Co-NCH according to the XAS results. We have supplemented the DFT calculations. The results show that the ORR process is more favorable on the Co-N₄ sites than that on the Fe-N₅ sites. As shown in **Supplementary Figs. 24 and 26**, the RDS of the CoN₄ model is the last electron transfer step of the desorption of *OH with a free energy of 0.42 eV while for the FeN₅ model, the RDS is *OOH adsorption with a free energy of 0.92 eV, which is much higher than that of the CoN₄ model. This result implies that CoN₄ may possess a better ORR activity than FeN₅.

Supplementary Figure 24. ORR free energy diagrams for the Co site in Co-N₄ (green line) and in FeN₅-CoN₄ models (orange line) at U=0 V, 0.9 V, and 1.23 V.

Supplementary Figure 26. ORR free energy diagrams for the Fe site in FeN₅ (orange line) and in FeN₅-CoN₄ models (blue line) at U=0 V, 0.9 V, and 1.23 V.

(2) **The Co-NCH has more active sites than the Fe-NCH** (5.5 wt% for Co vs. 2.4 wt% for Fe). The lower metal loading corresponds to the fewer functional sites to catalyze the ORR. As shown in **Supplementary Fig. 21**, the Co-NCH exhibits a large limiting current density than the Fe-NCH, indicating that more active sites participate in the ORR process than the Fe-NCH.

The following description has been added to the revision (highlighted in red, first paragraph, page 9)

“Notably, the DFT calculation also predicts the higher intrinsic ORR activity of Co-N₄ to the Fe-N₅. Together with more active site loading, the Co-NCH exhibits better performance than the Fe-NCH (detailed discussion in Supplementary Fig. 26).”

The detailed analysis has been added in the **Supplementary Information** (highlighted in red, last paragraph, page 27-28)

“Notably, the higher ORR performance of Co-NCH than Fe-NCH can be attributed to the following two reasons.

(1) Higher intrinsic activity of Co-N₄ compared to the Fe-N₅ sites. The DFT calculation results show that the ORR process is more favorable on the Co-N₄ sites than that on the Fe-N₅ sites. The RDS of the Co-N₄ model is the last electron transfer step of the desorption of *OH with a free energy of 0.42 eV. While for the FeN₅ model, the RDS is *OOH adsorption with a free energy of 0.92 eV, which is much higher than that of the CoN₄ model. This result implies that CoN₄ may possess a better ORR activity than FeN₅ in our case.

(2) Co-NCH has more active sites than Fe-NCH. The metal loading of Co-NCH is 5.5 wt% vs. 2.4 wt% of Fe in Fe-NCH. The higher metal loading corresponds to more functional sites to catalyze the ORR. As shown in **Supplementary Fig. 21**, the Co-NCH exhibits a larger limiting current density than the Fe-NCH, indicating that more active sites participate in the ORR process than the Fe-NCH.

Comment 4:

Page 9, line 286. The TOF value is determined at 0.8 V. At this potential the ORR process on a FeCo-NCH catalyst is under diffusion control. The TOF value should be determined at 0.9 V.

Response 4:

Thanks for the constructive comment. We have recalculated the TOF value at 0.9 V, The corresponding description and **Supplementary Table 5** have be updated.

In the revision (highlighted in red, first paragraph, page 9):

“The TOF value of the FeCo-NCH is calculated to be $0.6 \text{ e}^{-1} \text{ s}^{-1} \text{ site}^{-1}$ at 0.90 V, which is higher than most of previously reported ORR catalysts (Supplementary Table 5), indicating a fast ORR kinetic.⁵³”

And in **Methods** section (highlighted in red, page 12):

“The turnover frequency (TOF) was calculated by the following equation:

$$\text{TOF} [\text{e}^{-1} \text{ s}^{-1} \text{ site}^{-1}] = j_k (\text{mA cm}^{-2}) \times N_e / W \times C_{cat} \times N_A / M$$

Where n_{strip} ($= 5$) is the number of electrons associated with the reduction of one nitrite per site; F is the Faraday constant (96485 C mol^{-1}); j_k is the kinetic current density, N_e is the electronnumber per Coulomb (6.24×10^{18}), W is the metal contents of FeCo-NCH or FeCo-NC. C_{cat} is the catalyst loading on the electrode. N_A is the Avogadro constant (6.022×10^{23}), M is the average atomic mass of Fe and Co for FeCo-NCH or FeCo-NC (based on the ratio of Fe and Co).”

Comment 5:

Page 9, line 298. “The smaller Tafel slope indicates enhanced ORR kinetics.^{52, 53}” This claim is too simplistic. The sentence and refs. 52, 53 should be removed from the manuscript.

Response 5:

Thanks for pointing out this unprecise description. To avoid any confusion, we have removed the sentence and refs 52,53 in the revised manuscript as you suggested.

Comment 6:

Page 9, lines 302-305. The n value is around 3.7, so still lower than 4, which means that there is some 2-electron ORR occurring as is also evident from peroxide yield in the RRDE studies. It would be correct to state that the ORR proceeds mostly via 4-electron pathway. Also, the claim “less than ~ 5%” is not true - the yield is ~5%, at 0.6-0.8 V slightly above 5% (Figure S29). The formation of peroxide is detrimental in fuel cells. It reduces the fuel cell efficiency and degrades the catalysts and membranes. The authors should discuss about these aspects in a short paragraph in the main manuscript.

Response 6:

Thanks for pointing out this unprecise description. The corresponding description have be rephrased in the revision as follows. Besides, a short paragraph for the peroxide for the ORR and fuel cell performance has also been added thereafter.

In the revision (highlighted in red, last paragraph, page 9):

“FeCo-NCH shows an n value ranging from 3.76 to 3.96 and produces only ~ 5% HO_2^- in the potential range of 0.4 – 0.8 V. The slightly higher HO_2^- yield in FeCo-NCH should come from the Co-N_4 sites, which is common for Co SACs.^{54, 55} This demonstrates that the ORR proceeds on the FeCo-NCH mainly via the 4-electron pathway, with is essential for fuel cells. The side 2-electron pathway with the formation of HO_2^- reduces the fuel cell efficiency and degrades the catalysts and membranes, which is detrimental in fuel cells.^{55”}

References:

54. Lilloja, J., *et al.* Cathode catalysts based on cobalt- and nitrogen-doped nanocarbon composites for anion exchange membrane fuel cells. *ACS Appl. Energy Mater.* **3**, 5375-5384 (2020).

55. Lilloja, J., *et al.* Transition-metal- and nitrogen-doped carbide-derived carbon/carbon nanotube composites as cathode catalysts for anion-exchange membrane fuel cells. *ACS Catal.* **11**, 1920-1931 (2021).

Comment 7:

A comparison of AEMFC results with literature data is still insufficient. More discussion about the results should be added to the text. For example, Kumar et al. reported a better AEMFC performance of CoFe-N-C catalysts (<https://doi.org/10.1021/acsami.1c06737>). While the H_2 /air fuel cell data is included in Table S4, there is no discussion about it. For some reason the authors have decided to exclude results over 1000 mW cm^{-2} from Figure S34. Also, there is no comparison brought out with PAP-TP-85 membrane, as membrane plays crucial role, then comparing results obtained with the same membrane would be more informative.

Response 7:

Thank you for the useful information. As you suggested, we have made the following changes to response the above comments.

- 1) A brief discussion has been added in the revision to discuss the AEMFC results and highlights the importance of constructing advanced catalyst architecture for AEMFC (highlighted in red, second paragraph, page 10).

“The performance for FeCo-NCH MEA ranks among the highest for non-PGM-based AEMFC cathode catalysts (Supplementary Fig. 34 and Supplementary Table 4).^{33, 57} The importance of constructing advanced catalyst architecture was also demonstrated by Kumar et al.⁵⁸ By constructing three-dimensional (3D) carbon nanotube network with superior electron and mass transfer properties, they reported a FeCoN-MWCNT catalyst with a high P_{max} of 692 mW cm⁻².”

The following Ref. No 58 have also been added to **Supplementary Fig. 34** and **Supplementary Table 4** for comparison.

58. Kumar, Y., et al. Bifunctional oxygen electrocatalysis on mixed metal phthalocyanine-modified carbon nanotubes prepared via pyrolysis. *ACS Appl. Mater. Interfaces* **13**, 41507-41516 (2021).

- 2) A discussion has been added in the revision to highlight the importance of the catalyst architecture optimization for the H₂-air performance and introduce the H₂-air performance of the state-of-the-art M-N-C catalysts listed in **Supplementary Table 4** (highlighted in red, second paragraph, page 10).

“these values are also comparable to the reported non-PGM-based cathode catalysts, as summarized in **Supplementary Table 4**. The performance variation between H₂-air and H₂-O₂ conditions was due to the lower O₂ content in the realistic air, leading to sluggish mass transport behavior. The design of catalyst architecture to provide greater active site accessibility thus accelerating mass transfer properties is more vital under H₂-air conditions.⁵⁹ Notably, the FeCo-NCH MEA also demonstrated nearly the same current density as Pt-based MEA in the kinetic region (>0.75 V), effectively illustrating the promoted mass transport properties in FeCo-NCH by the interfacial assembly strategy. Lilloja and Kumar et al also demonstrated a high-performance H₂-air fuel cell by constructing a 3D carbon nanotube network to promote mass transfer (**Supplementary Table 4**).^{55, 58”}

References:

55. Lilloja, J., et al. Transition-metal- and nitrogen-doped carbide-derived carbon/carbon nanotube composites as cathode catalysts for anion-exchange membrane fuel cells. *ACS Catal.* **11**, 1920-1931 (2021).

58. Kumar, Y., et al. Bifunctional oxygen electrocatalysis on mixed metal phthalocyanine-modified carbon nanotubes prepared via pyrolysis. *ACS Appl. Mater. Interfaces* **13**, 41507-41516 (2021).

59. Chung, H. T., *et al.* Direct atomic-level insight into the active sites of a high-performance PGM-free ORR catalyst. *Science* **357**, 479-484 (2017).

3) The following **Supplementary Fig. 34** has been updated in the revised **Supplementary Information** (highlighted in red, page 36)

Supplementary Figure 34. Peak power density comparison of the FeCo-NCH with reported Pt-free catalysts.

4) The PAP-TP-85 membrane used in this manuscript was prepared according to the report of Professor Yan's research group. (*Nat. Energy* **4**, 392–398 (2019), *Nat. Commun.* **11**, 5651 (2020), *J. Electrochem. Soc.* **166**, F3305 (2019)). The PAP-TP-85 membrane features the following advantages: *i*) Good stability in the alkaline environment. *ii*) Highly rigid skeleton that can maintain a moderate swelling ratio for better mechanical strength and integrity of the membrane electrode assembly in the operating cells. *iii*) The ion exchange capacity and hydrophobicity of the membrane can be adjusted. With this membrane, we can achieve efficient and stable operation of AEMFC under our operating conditions, while the anode and cathode heat rejection can be also alleviated. However, such an exchange membrane was developed in 2019 (US Patent No. US10290890B2) and has not yet been commercialized on a large scale. We currently lack comparable data on different non-PGM cathode materials. We will compare them in our future works.

Reproduced minor remarks and responses:

We greatly appreciate you for the very careful reading and for pointing out these minor issues/typos

in writing and discussion. The related corrections have been made in the revision accordingly.

1. SAC is not defined.

Response:

The definition of “single-atom catalysts (SACs)” has been added in the revision.

2. Page 2, lines 65-67. “The interaction between multiple active centers allows regulating the electronic structure and geometric configuration, enabling to optimize of the adsorption and desorption of intermediates”. This sentence is a bit confusing and should be re-phrased.

Response:

We have rephrased the description as follows.

In the revision (highlighted in red, lines 65-67, page 2):

“The interaction between multiple active centers modifies the electronic structure and geometric configuration of the active site, which optimizes the adsorption and desorption of intermediates on the active site to approach the apex of the activity volcano plot.”

3. Page 5, lines 162 and 187. ‘references’ should be replaced with ‘reference materials’.

Response:

We have clarified the description as suggested.

4. Page 8, line 261 and elsewhere. The term “mesoporous shell” etc. is a bit misleading as the nanocage structure contains plenty of micropores together with small mesopores. Please correct.

Response:

We have made the corrections as suggested. The term “mesoporous shell” has been rephrased to “porous shell” throughout the revised manuscript and **Supplementary Information**.

5. Page 10, line 362. The title should be ‘Conclusions’ instead of ‘Discussion’.

Response:

Thank you very much for the comment. We have changed the title to ‘Conclusions’ as suggested.

6. Page 11, line 390. 'Nafion' should be 'Nafion'.

Response:

We have corrected this typo in the revision.

7. Page 12, lines 502 and 504. It should be 'Vulcan' not 'Vulkan'.

Response:

Thank you very much. We have corrected this typo.

8. Page 12. The origin of PAP-TP ionomer and membrane is missing.

Response:

Thank you very much for the comment. The PAP-TP ionomer and membrane used in the manuscript are synthesized according to Yan et al's reports (*Nat. Energy* **4**, 392-398 (2019)). The following content and corresponding references were added in the **Methods** and **References** section as follows.

In the **Methods** section (highlighted in red, page 13):

"The hydrogen exchange membrane and ionomer (PAP-TP- x , x is the molar ratio between N-methyl-4-piperidone and terphenyl monomers) was synthesized as reported by Yan et al.^{10, 65"}

In the **References** section (highlighted in red, page 14-15):

10. Xue, Y., et al. A highly-active, stable and low-cost platinum-free anode catalyst based on RuNi for hydroxide exchange membrane fuel cells. *Nat. Commun.* **11**, 5651 (2020).

69. Wang, J., et al. Poly(aryl piperidinium) membranes and ionomers for hydroxide exchange membrane fuel cells. *Nat. Energy* **4**, 392-398 (2019).

9. Page 11, line 551. Ref. 15 must be updated Appl. Catal. B 325, 121733 (2023).

Response:

We have updated the corresponding reference as follows.

In the **References** section (highlighted in red, page 14)

15. Hossen, M. M., et al. State-of-the-art and developmental trends in platinum group metal-free cathode catalyst for anion exchange membrane fuel cell (AEMFC). *Appl. Catal., B* **325**, 121733

(2023).

10. Was any electrochemical cleaning/activation applied to the Pt-ring prior to the ORR measurements?

Response:

Prior to the ORR measurements, the Pt ring was electrochemically cleaned via the CV scans. The detailed procedure was added in the **Methods** section as follows.

In the revision (highlighted in red, page 12)

“The Pt ring was cleaned with CV scans from 0.05 to 1.2 V (vs. RHE) for 100 cycles at a scan rate of 50 mV s⁻¹.”

11. Figure S32. The x-axis values should start from 0 mA cm⁻².

Response:

Thank you for the comment. **Supplementary Figure 32** has been updated in the revised **Supplementary Information** as follows (highlighted in red, page 34).

Supplementary Figure 32. ORR LSV curves of FeCo-NCH before and after 10,000 cycles between 0.6 and 1.0 V at a scan rate of 100 mV s⁻¹.

REVIEWERS' COMMENTS

Reviewer #5 (Remarks to the Author):

Manuscript: NCOMMS-22-33399B

“Boosting the Accessible Site Density by Interfacial Assembly of Binary Atomic Metal–Nx for High-Performance Energy Devices”, by Zhe Jiang, Xuerui Liu, Xiao-Zhi Liu, Shuang Huang, Ying Liu, Ze-Cheng Yao, Yun Zhang, Qing-Hua Zhang, Lin Gu, Li-Rong Zheng, Li Li, Yianan Zhang, Youjun Fan, Tang Tang, Zhongbin Zhuang, Jin-Song Hu

The paper could be accepted for publication in this Journal after making minor corrections.

Minor comments:

Page 2, line 24. A review on ZABs should be cited (<https://doi.org/10.1016/j.coelec.2023.101229>). Important literature on AEMFC performance is missing (<https://doi.org/10.1016/j.jcis.2020.09.114>; <https://doi.org/10.1016/j.electacta.2022.141676>; <https://doi.org/10.1016/j.cej.2023.141468>).

Page 9, line 21. The unit of TOF needs to be corrected (e- s-1 site-1). See also page 12, line 61.

Page 10, line 25. “in the realistic air” should be replaced with “in air”

Page 11, line 35. “Nafion” should be replaced with “Nafion solution”

Responses to the Reviewers' Comments

Responses to Reviewer #5's comments

Manuscript: NCOMMS-22-33399B

“Boosting the Accessible Site Density by Interfacial Assembly of Binary Atomic Metal–Nx for High-Performance Energy Devices”, by Zhe Jiang, Xuerui Liu, Xiao-Zhi Liu, Shuang Huang, Ying Liu, Ze-Cheng Yao, Yun Zhang, Qing-Hua Zhang, Lin Gu, Li-Rong Zheng, Li Li, Yianan Zhang, Youjun Fan, Tang Tang, Zhongbin Zhuang, Jin-Song Hu

The paper could be accepted for publication in this Journal after making minor corrections.

Brief response:

We greatly appreciate Reviewer #5's valuable time, very careful reading, and constructive comments, which have helped us to improve the quality of our manuscript. We have addressed the comments point-by-point as follows.

Reproduced specific comments and responses:

Minor comment 1:

Page 2, line 24. A review on ZABs should be cited (<https://doi.org/10.1016/j.coelec.2023.101229>).

Response 1:

Thank you for the comment. We have read the literature carefully and cited it appropriately in the revision as **Ref. 13** (highlighted in red, page 12).

13. Kumar Y., Mooste M. & Tammeveski K. Recent progress of transition metal-based bifunctional electrocatalysts for rechargeable zinc–air battery application. *Curr. Opin. Electrochem.* **38**, 101229 (2023).

Minor comment 2:

Important literature on AEMFC performance is missing (<https://doi.org/10.1016/j.jcis.2020.09.114>; <https://doi.org/10.1016/j.electacta.2022.141676>; <https://doi.org/10.1016/j.cej.2023.141468>).

Response 2:

Thank you for the comment. We have read the literature carefully and added the AEMFC performances of the catalysts reported therein to **Supplementary Table 4**. The corresponding references were added to the references list of the revision and revised **Supplementary Information**.

As Refs. 59, 60, and 61 in revision (highlighted in red, page 13-14).

59. Lilloja J. *et al.* Cobalt-, iron- and nitrogen-containing ordered mesoporous carbon-based catalysts for anion-exchange membrane fuel cell cathode. *Electrochim. Acta* **439**, 141676 (2023).

60. Akula S. *et al.* Transition metal (Fe, Co, Mn, Cu) containing nitrogen-doped porous carbon as efficient oxygen reduction electrocatalysts for anion exchange membrane fuel cells. *Chem. Eng. J.* **458**, 141468 (2023).

61. Kisand K. *et al.* Transition metal-containing nitrogen-doped nanocarbon catalysts derived from 5-methylresorcinol for anion exchange membrane fuel cell application. *J. Colloid Interface Sci.* **584**, 263-274 (2021).

And as Refs. 38, 48, and 49 in the **Supplementary Information** (highlighted in red, pages S48-49)

Minor comment 3:

Page 9, line 21. The unit of TOF needs to be corrected ($e^- s^{-1} site^{-1}$). See also page 12, line 61.

Response 3:

Thanks for pointing out this unprecise description. We have corrected it in the revision (highlighted in red, pages 6 and 10)

Minor comment 4:

Page 10, line 25. “in the realistic air” should be replaced with “in air”

Response 4:

We appreciate for pointing out this unprecise description. We have replaced “in the realistic air” with “in air” in the revision (highlighted in red, page 7).

Minor comment 5:

Page 11, line 35. “Nafion” should be replaced with “Nafion solution”

Response 5:

Thanks for pointing out this unprecise description. We have amended this statement in the revision (highlighted in red, pages 8 and 11).